# A Survey of Reinforcement Learning from Human Feedback

**Timo Kaufmann**  *timo.kaufmann@ifi.lmu.de*
*LMU Munich, MCML Munich*

**Paul Weng**  *paul.weng@dukekunshan.edu.cn*
*Digital Innovation Research Center, Duke Kunshan University*

**Viktor Bengs**  *viktor.bengs@dfki.de*
*German Research Center for Artificial Intelligence (DFKI)*

**Eyke Hüllermeier**  *eyke@lmu.de*
*LMU Munich, MCML Munich, DFKI Kaiserslautern*

**Reviewed on OpenReview:** *https://openreview.net/forum?id=f7OkIurx4b*

## Abstract

Reinforcement learning from human feedback (RLHF) is a variant of reinforcement learning (RL) that learns from human feedback instead of relying on an engineered reward function. Building on prior work on the related setting of preference-based reinforcement learning (PbRL), it stands at the intersection of artificial intelligence and human-computer interaction. This positioning provides a promising approach to enhance the performance and adaptability of intelligent systems while also improving the alignment of their objectives with human values. The success in training large language models (LLMs) has impressively demonstrated this potential in recent years, where RLHF has played a decisive role in directing the model's capabilities towards human objectives. This article provides an overview of the fundamentals of RLHF, exploring how RL agents interact with human feedback. While recent focus has been on RLHF for LLMs, our survey covers the technique across multiple domains. We provide our most comprehensive coverage in control and robotics, where many fundamental techniques originate, alongside a dedicated LLM section. We examine the core principles that underpin RLHF, how algorithms and human feedback work together, and discuss the main research trends in the field. Our goal is to give researchers and practitioners a clear understanding of this rapidly growing field.

## Contents

# 1 Introduction

In reinforcement learning (RL), an agent navigates an environment and learns through trial and error. The optimality of these decisions is determined solely by reward signals, which must be carefully designed based on performance measurements to ensure the agent learns the desired behavior. However, designing effective reward functions presents significant challenges. In real-world applications, success is often difficult to formally define and measure. Moreover, sparse success signals are not well suited for agent learning, often necessitating *reward shaping* (Ng et al., 1999), where the reward signal is transformed into one that is more suitable for learning. Unfortunately, this transformation can introduce spurious correlations – rewarding behaviors that correlate with desired outcomes without directly achieving the true objective. This can lead to *reward hacking* (Skalse et al., 2022), where agents exploit such correlations to maximize rewards while producing undesired outcomes.

Reinforcement learning from human feedback (RLHF) has emerged as a practical solution to these challenges by incorporating human feedback directly into the learning process. Unlike traditional RL, where objectives are fixed a priori, RLHF allows humans to define and iteratively refine objectives during training. This approach addresses the limitations of classical RL while promoting better alignment between agent behavior and human values and objectives, enabling the development of more ethically sound, socially responsible, and practically useful AI systems.

RLHF has seen a number of successful applications, methodological advances, and theoretical insights since the last comparable survey (Wirth et al., 2017). The applications span various domains, including large language model (LLM) fine-tuning (OpenAI, 2022), image generation (Lee et al., 2023), music generation (Cideron et al., 2024), continuous control (Christiano et al., 2017), games (Ibarz et al., 2018), and robotics (Hejna & Sadigh, 2022). Alongside these applications, significant methodological advances have emerged. These include fusing multiple feedback types to leverage their relative strengths (Section 3.5), enhancing query efficiency through active learning and query synthesis (Section 4.1.1), incorporating psychological insights to improve feedback quality (Section 4.2.1), using techniques such as meta-learning to quickly adapt learned preferences to new tasks using prior data (Section 5.5.1), and using available preference data more efficiently through approaches such as data augmentation and semi-supervised learning (Section 5.5.2). Finally, novel theoretical results (Section 7) have provided insights as well as new questions about the fundamental mathematical problems underlying RLHF.

While we focus on the core principles underlying RLHF, LLM fine-tuning requires additional considerations due to the unique characteristics of LLMs and their operational scale. We address these considerations without delving deeply into LLM-specific details. Notably, applying RLHF to LLMs is not our primary focus. While RLHF for LLMs remains in scope, we survey RLHF broadly, examining techniques and applications across domains with particular attention to control applications. Nonetheless, many techniques and insights discussed herein apply to LLMs, and we address some unique aspects of LLM fine-tuning. For readers primarily interested in LLM fine-tuning, Section A provides an LLM-focused introduction that complements this survey and guides the application of the insights presented here to LLM fine-tuning. Note that the literature focused on RLHF for LLMs in particular is vast and rapidly growing, and we cannot provide a comprehensive overview of this area.

This survey therefore provides an overview of current RLHF research, classifies existing approaches, describes their main characteristics, and briefly surveys application areas. In the remainder of this section, we begin by examining the motivation (Section 1.1) and origins (Section 1.2) of RLHF, defining the scope of this survey (Section 1.3), and outlining the structure of subsequent sections (Section 1.4).

## 1.1 Why Human Feedback?

In conventional RL, the agent's objective is defined by a reward function that it aims to maximize (Sutton & Barto, 2018). Specifying this reward function can be challenging, particularly in complex domains: What would be a suitable reward function for a robot assisting humans in a household environment or for autonomous vehicles navigating through a busy urban environment? Moreover, even reward functions that initially seem well-defined can lead to unexpected behaviors due to distributional shifts or over-optimization,

raising practical and safety concerns. Learning the agent's objective from human feedback circumvents reward engineering challenges and fosters robust training, with the reward function being dynamically refined and adjusted to distributional shifts as the agent learns.

**Interactive Feedback vs. Demonstrations**  The field of inverse RL aims to infer reward functions from human demonstrations (Arora & Doshi, 2021). While this can partially resolve reward engineering challenges, it faces inherent difficulties: (i) it is generally not possible to robustly identify rewards from demonstrations (Cao et al., 2021a; Mindermann & Armstrong, 2018), (ii) it is only applicable in scenarios where good demonstrations can be obtained, (iii) it struggles to outperform the demonstrator, and (iv) humans often do not demonstrate the behavior they would prefer a machine to adopt (Basu et al., 2017). Interactive feedback, by contrast, can use active queries to differentiate between aspects relevant or irrelevant to the human preference, is much easier to provide than demonstrations, does not require near-optimal performance from human evaluators, and elicits preferences about the behavior that a human would prefer from the machine. Interactive feedback also complements demonstrations effectively, refining capabilities learned through initial training methods like behavioral cloning and preventing overfitting to demonstrated behavior (Abramson et al., 2022).

**Avoiding Reward Engineering**  Reward engineering in RL presents significant challenges, as accurately specifying reward functions is notoriously difficult (Amodei et al., 2016; Knox et al., 2023). These challenges can be mitigated by using human feedback, which enables training agents for tasks that are hard to define manually and represents a step towards addressing safety issues arising from misaligned rewards (Skalse et al., 2022). Safety issues related to a misalignment between the agent's and the human's objectives are studied as the AI alignment problem (Gabriel, 2020), in particular agent alignment and value alignment (Kirchner et al., 2022).

Excessive optimization of poorly specified rewards frequently produces unintended behaviors. Examples of such behaviors include exploiting flaws in the environment simulation for higher rewards (Lehman et al., 2020; Baker et al., 2020) or engaging in more general *reward hacking* (Skalse et al., 2022), where the behavior maximizes the specified reward but deviates from the intended objective. This is evident in cases where agents focus on intermediate rewards without achieving the actual goal (Clark & Amodei, 2016) or prematurely exit games to avoid negative rewards (Saunders et al., 2018). The root of these issues is that the reward function does not properly reflect the actual learning task. While such problems appear trivial in game environments, they become critical in safety-critical contexts such as healthcare and autonomous driving, where misaligned rewards could lead to care robots causing injuries or self-driving cars compromising road safety.

RLHF presents a promising approach to enhance alignment (Leike et al., 2018) by enabling agents to learn from human feedback, which is often more closely aligned with the true objective than manually specified rewards. Nonetheless, the effectiveness of RLHF in resolving these alignment issues is debated (Christiano, 2023). Examples of possible pitfalls raised in this debate are that an agent may be incentivized to manipulate the human teacher to provide feedback that is easier to optimize (Armstrong et al., 2020; Carroll et al., 2023) or that the agent may learn to exploit errors in human judgment (Ngo et al., 2024). We refer the interested reader to the survey by Casper et al. (2023) for a more detailed discussion of these issues. Despite these concerns, RLHF represents an important early step towards aligning agents with human values and serves as a foundation to build on to further improve agent alignment.

## 1.2  The Origins of RLHF

Learning behavior from human feedback has long been studied as a subfield of RL, but methods and terminology have evolved over time. Early methods focused on learning directly from human rewards (Knox, 2012; Isbell et al., 2001; Knox & Stone, 2009), from action advice (Maclin et al., 2005), or from action critique (Judah et al., 2010). Notable examples include TAMER (Knox & Stone, 2009; Warnell et al., 2018), which interprets human feedback as samples of the optimal action-value function, and COACH (MacGlashan et al., 2017; Arumugam et al., 2019), which interprets human feedback in a policy-dependent way, i.e., as samples of the advantage function. This survey, however, focuses on more indirect approaches to inferring the objective from human feedback.

Table 1: Feedback types classified as belonging to PbRL, SSRL, and RLHF as defined in this survey.

| Feedback Type | PbRL | SSRL | RLHF |
|---|:---:|:---:|:---:|
| Binary trajectory comparisons | ✓ | ✗ | ✓ |
| Trajectory rankings | ✓ | ✗ | ✓ |
| State preferences | ✓ | ✗ | ✓ |
| Action preferences | ✓ | ✗ | ✓ |
| Binary critique | ✗ | ✓ | ✓ |
| Scalar feedback | ✗ | ✓ | ✓ |
| Corrections | ✗ | ✗ | ✓ |
| Action advice | ✗ | ✗ | ✓ |
| Implicit feedback | ✗ | ✗ | ✓ |
| Natural language | ✗ | ✗ | ✓ |

Modern reinforcement learning from human feedback (RLHF) originates from preference-based reinforcement learning (PbRL), independently introduced by Akrour et al. (2011) and Cheng et al. (2011). PbRL infers the objective from qualitative feedback, such as pairwise preferences between behaviors or between actions given states, instead of quantitative feedback in the form of numerical rewards. The term RLHF was later coined as an alternative (Askell et al., 2021; Ouyang et al., 2022; OpenAI, 2022), competing with the short-lived term reinforcement learning from human preferences (RLHP) (Menick et al., 2022), though initially referring to the same concept of learning behavior from relative feedback.

Disentangling PbRL and RLHF is challenging due to their overlapping usage in the literature. For instance, Christiano et al. (2017) themselves are using the term PbRL, yet are often cited as a seminal reference for RLHF (Daniels-Koch & Freedman, 2022; Ouyang et al., 2022). This demonstrates that the terms are often used interchangeably. Practically, RLHF is often associated with reward modeling and deep RL, while PbRL is often linked to direct policy optimization in traditional RL settings. This is underlined by Jeon et al. (2020), who characterize PbRL as limited to direct policy learning from preferences. This is in contrast with other sources, however, who include reward learning within the scope of PbRL (Christiano et al., 2017; Wirth et al., 2017). Additionally, PbRL is rarely used in the language modeling domain, while RLHF is common in both language modeling and classical RL domains such as continuous control.

Despite overlapping and sometimes conflicting usage, in this work we view RLHF as a generalization of PbRL. While both involve human feedback to define RL objectives, PbRL primarily focuses on relative feedback, such as binary comparisons and rankings. RLHF not only includes these aspects but also extends to a wider range of feedback types (Metz et al., 2023; Yuan et al., 2024). Table 1 gives an exemplary overview of our interpretation of these terms.

Another concept, semi-supervised reinforcement learning (SSRL), introduced by Christiano (2016) and discussed by Amodei et al. (2016), refers to an RL setting where an agent receives feedback on a subset of its experiences. The initial discussions of SSRL focused on absolute feedback on subsets of the agent's experiences, making the concept complementary to PbRL. In contrast to PbRL and RLHF, the term SSRL seems to be used less in the recent literature.

We adopt the view that RLHF broadly encompasses all approaches using human feedback to define RL objectives, including both PbRL and SSRL. As the definitions and distinctions between these terms are not universally agreed upon, these distinctions are based on our interpretation of the current predominant usage of these terms in the literature.

## 1.3 Scope of the Survey

This section outlines the criteria guiding our selection of RLHF approaches we cover. We concentrate on methods where a learned reward model serves as the sole source of objective information, learned through interactive, online, scalable, and asynchronous human feedback. We describe each of these criteria in more detail below.

**Reward Modeling** We focus on approaches that learn a reward model from human feedback and then use this model to train a policy. Although it is possible to directly optimize a policy from human feedback (Wirth et al., 2017), thereby performing RLHF without reward learning, this approach was rarely practiced for a long time and has only recently gained renewed interest, especially in the domain of language model fine-tuning (see Section 6.3). The decomposition into reward learning and policy training offers many conceptual and practical benefits. These include the direct applicability of supervised learning techniques for the reward model and the ability to evaluate the reward model in isolation. Additionally, the decomposition naturally leads to a form of semi-supervised learning, enabling the agent to use labeled episodes for reward model training while leveraging unlabeled episodes to refine its behavior and explore the environment.

**Human Defined** While many approaches include humans in the RL loop, we focus on approaches where human feedback is the only source of truth about the objective. This excludes approaches to reward shaping, feature engineering, and other forms of human guidance that are supplementary to a given objective.

**Interactive and Online** Our scope encompasses methods that collect human feedback interactively during the learning process, rather than relying solely on pre-collected demonstrations. This excludes pure imitation learning, learning from demonstration, and standalone inverse RL approaches. However, hybrid methods that combine inverse RL with interactive reward refinement fall within our scope, as discussed in Sections 3.3 and 5.5.1.

**Scalable and Asynchronous** We consider methods that incorporate human feedback without requiring synchronous human-agent interaction. The agent must be capable of continuing its learning process while awaiting human input, and humans need not provide continuous supervision. This distinguishes RLHF from more direct methods of incorporating a human into the RL loop, and we believe that this is key for practicality and efficiency.

Beyond these methodological criteria, we primarily cover works from 2017 through 2024[1], building upon the comprehensive treatment of earlier works by Wirth et al. (2017). We selectively revisit earlier foundational contributions that continue to inform current practice or have fundamentally shaped the field.

While RLHF has recently gained prominence through LLM applications, this survey adopts a broader perspective, examining RLHF across multiple domains with particular depth in control and robotics. We dedicate Section A to LLMs, addressing domain-specific challenges including large action spaces, distribution shift, and KL regularization. Many techniques from control domains transfer naturally to the LLM setting, which is frequently constrained to a single turn, though not all techniques are equally applicable. Given the rapidly evolving nature of LLM research, our coverage of techniques specific to this domain is less exhaustive than our treatment of control and robotics applications.

This survey can be seen as the canonical continuation of Wirth et al. (2017), examining the evolution from PbRL to RLHF and the accompanying methodological advances. We provide both a thorough description of the basics and an in-depth discussion of current advances and trends. We refer to Section B for an overview of prior and related surveys and their relation to this work.

## 1.4 Outline

In the next section, we begin with an introduction to the basics by revisiting the most important concepts from the standard RL setting, which are also naturally important in RLHF (Section 2). We then dive into the RLHF topic by outlining the most studied scenario of reward model learning from pairwise preferences. Using this introductory and illustrative example scenario, we explain the basic framework of RLHF alongside its three main components of (human) feedback, label collection (feedback acquisition), and reward model learning. These three main components will essentially form the structure of our survey. In Section 3, we turn our attention to the human feedback component and provide an overview of the different types of feedback as well as their key attributes. The important concepts in terms of label collection are then

---

[1]We updated preprints to their peer-reviewed versions where applicable, leading to references outside this window.

explained in Section 4, followed by learning the reward model in Section 5. We also discuss policy learning in Section 6, since some RLHF methods adapt standard RL training methods to the RLHF setting. Section 7 is devoted to an overview of recent progress on the theoretical side of RLHF, including approaches involving a theoretical guarantee, as well as theoretical insights into the relationship between standard RL and RLHF. Finally, Section 8 highlights some interesting practical applications of RLHF and the existing benchmarks before Section 9 concludes the survey by pointing out some possible avenues for future work.

## 2 Preliminaries

In this section, we review the basic setting and the most important concepts of RL and RLHF. In the course of this review, we will establish the notation that will be used throughout the survey. We first introduce what is probably the most studied RLHF scenario, i.e., learning a reward model from binary trajectory comparisons. Based on this introductory and illustrative example scenario, we explain the basic framework of RLHF with its main components and briefly discuss the respective roles of these components in the learning process. We will also briefly touch on active learning, which strongly connects to the feedback collection component.

**Notation**   For any integer $n \in \mathbb{N}$, we denote by $[n]$ the set $\{1, 2, \ldots, n\}$. For any set $S$, $\Delta(S)$ denotes the set of probability distributions over $S$. We use $\mathbb{P}(E)$ for denoting the probability of some event $E$, while $\mathbb{E}[X]$ is used to denote the expected value of a random variable $X$. In some cases, we will write $\mathbb{E}_P[\cdot]$ or similar variants to emphasize that the distribution for the expected value is governed by the probability distribution $P \in \Delta(S)$. Moreover, we will write $X \sim P$ if a random variable $X$ is distributed according to a probability distribution $P$.

### 2.1 Reinforcement Learning

Reinforcement learning (RL) (Sutton & Barto, 2018) is the setting of learning behavior from rewarded interaction with an environment. Such a learning environment is formalized as an Markov decision process (MDP), which is a model for sequential decision-making. In an MDP, an agent iteratively observes its current state, takes an action that causes the transition to a new state, and finally receives a reward that depends on the action's effectiveness. Formally, an MDP is defined as a tuple $(\mathcal{S}, \mathcal{A}, P, R, d_0, \gamma)$ where

- $\mathcal{S}$ is a set of states (the *state space*),

- $\mathcal{A}$ is a set of actions (the *action space*),

- $P : \mathcal{S} \times \mathcal{A} \to \Delta(\mathcal{S})$ is a transition function (the *transition dynamics*),

- $R : \mathcal{S} \times \mathcal{A} \to \mathbb{R}$ is a reward function,

- $d_0 \in \Delta(\mathcal{S})$ is a distribution over initial states,

- and $\gamma \in [0, 1]$ is a discount factor.

The transition function $P$ defines the dynamics of the environment: For any state $s$ and action $a$, the value $P(s, a)(s')$, also sometimes denoted $P(s' \mid s, a)$, is the probability of reaching the state $s'$ after executing $a$ in $s$. In light of this, we will also sometimes refer to the transition function simply as the *transition dynamics*. For a given state and action, the transition is conditionally independent of all previous states and actions, which is known as the *Markov property* and the reason for the naming as an MDP. The value $R(s, a) \in \mathbb{R}$ provides an immediate evaluation after performing action $a$ in state $s$, which is also called the (instantaneous) reward. It is also possible that the instantaneous reward is 0 for some states, and one only receives a reward in specific states, for example, in so-called *terminal* states for which the transition function is zero. When both the state space $\mathcal{S}$ and the action space $\mathcal{A}$ are finite, we call the MDP a *tabular* MDP.

In an MDP, an *H-step trajectory* $\tau$ is a sequence of $H \in \mathbb{N} \setminus \{0\}$ state-action pairs ending in a terminal state. Formally, it is given by $\tau = (s_0, a_0, s_1, a_1, \ldots, s_H)$. A trajectory $\tau$'s *return* $R(\tau)$ is the accumulated

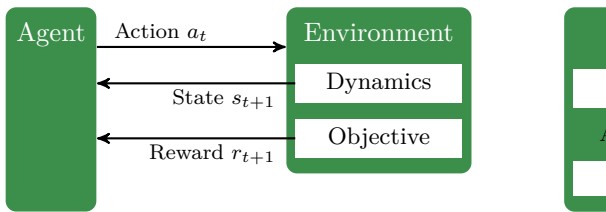
(a) The standard RL setting.

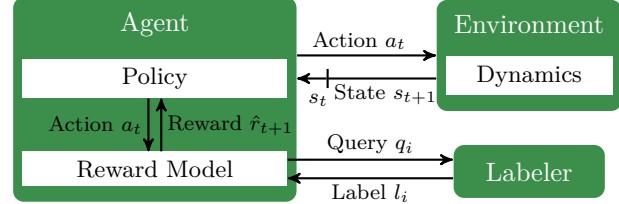
(b) RLHF with reward modeling.

Figure 1: Contrasting the standard RL setting with RLHF in its most common formulation, using a reward model. In each step, the policy commits to an action $a_t$ and receives the next state $s_{t+1}$ and either the true reward $r_{t+1}$ or an estimate $\hat{r}_{t+1}$ in return. In contrast to the standard RL setting, the true reward function is not known in the RLHF setting but instead learned from human feedback. This reward learning process is decoupled from policy learning and can happen fully asynchronously. The dataset consists of a set of queries $q_i$ (e.g., pairs of trajectory fragments) and their labels $l_i$ (e.g., a preference for one of the fragments).

(discounted) rewards collected along this trajectory:

$$R(\tau) = \sum_{h=0}^{H-1} \gamma^h R(s_h, a_h). \tag{1}$$

Note that we here use the same notation for the return and the reward function; however, both have different signatures (trajectory vs. state-action pair). We can also define the return $R(\sigma)$ of a segment $\sigma$ in a similar manner. The return is well defined even if the horizon $H$ is infinite as long as $\gamma < 1$. If the MDP is a tabular MDP and any trajectory has finite length, i.e., $H$ is necessarily finite, we call the MDP *finite* and otherwise *infinite*.

A *policy* specifies how to select actions in a state, either deterministically or stochastically. In the former case, a policy is simply a mapping $\pi : \mathcal{S} \to \mathcal{A}$ from states to actions, while in the latter, it is a mapping $\pi : \mathcal{S} \to \Delta(\mathcal{A})$ from states to probability distributions over actions. Since the deterministic case is a special case of the stochastic one, we assume the latter case in the following.

The basic interaction loop is depicted in Fig. 1a: The agent chooses an action $a_t \sim \pi(s_t)$ based on its policy and the current state. As a consequence, the environment transitions to the new state $s_{t+1} \sim P(s_t, a_t)$, governed by the transition dynamics. The agent observes this new state and the reward $r_{t+1} = R(s_t, a_t)$, after which the interaction cycle repeats.

In this setting, the RL agent aims at learning a policy that maximizes the expected return

$$J(\pi) = \mathbb{E}_{d_0, P, \pi}[R(\tau)],$$

where the expectation is with respect to policy $\pi$, transition function $P$, and initial distribution $d_0$. To solve this problem, two families of RL approaches have been considered: *model-based* RL and *model-free* RL. The methods in the first family learn a model (i.e., $P, R$) of the underlying MDP to help solve the RL problem, while the methods in the second directly try to obtain a good policy without learning an MDP model. The second family can be further decomposed into two main categories: *value-based* methods and *policy search* methods. In deep RL, both value functions and policies are approximated with neural networks.

Value-based methods (e.g., DQN and its variants (Mnih et al., 2015; Hessel et al., 2018)) aim at learning the $Q$-function $Q^*$ of an optimal policy. The $Q$-function of a policy $\pi$ is defined by:

$$Q_\pi(s, a) = \mathbb{E}_{P, \pi} \left[ \sum_{h=0}^{H-1} \gamma^h R(s_h, a_h) \right],$$

where $s_0 = s$ and $a_0 = a$, and in the expectation, $a_h \sim \pi(\cdot \mid s_h)$ as well as $s_h \sim P(\cdot \mid s_{h-1}, a_{h-1})$ for $h \in [H-1]$. A policy can be naturally designed from a $Q$-function by choosing an action in a greedy

manner in each state: $\pi(s) = \arg\max_a Q(s, a)$. Note that for a deterministic optimal policy $\pi^*$ it holds that $J(\pi^*) = \mathbb{E}_{d_0}[Q^*(s, \pi^*(s))]$.

Similar to the action-value function $Q$, we can also define the state-value function

$$V_\pi(s) = \mathbb{E}_{P,\pi}\left[\sum_{h=0}^{H-1} \gamma^h R(s_h, a_h) \,|\, s_0 = s\right].$$

Its value for some state $s$ is the expected return when starting in that state and then always using the policy $\pi$. It is related to the $Q$-function by means of

$$V_\pi(s) = \mathbb{E}_{a\sim\pi(s)}[Q_\pi(s, a)]$$

for any state $s \in \mathcal{S}$.

In contrast, policy search methods directly aim at finding a good policy in some parametrized policy space. The most data-efficient algorithms in this class of methods follow an actor-critic scheme where both an actor (i.e., a policy) and a critic (i.e., usually its $Q$-value function) are learned at the same time. Typical representative methods here are PPO (Schulman et al., 2017), TD3 (Fujimoto et al., 2018), or SAC (Haarnoja et al., 2018).

RL algorithms can further be classified as either *on-policy* or *off-policy*. In an on-policy algorithm, such as PPO, only the recently generated transitions are used for training. In contrast, in an off-policy algorithm, such as DQN (or its variants), TD3, or SAC, the agent can be updated with transitions not necessarily generated by its current policy. While on-policy training is usually more stable, off-policy training enables more data-efficient learning by reusing samples from a replay buffer that stores past transitions.

## 2.2 Preference-Based MDPs

In contrast to standard RL as described in the previous section, RLHF does not assume that a reward signal is available. It instead assumes the existence of an *oracle* (e.g., *human labeler*) that can provide information about the reward in a specific indirect manner. More precisely, in RLHF, the agent can make queries $q_i$ to the oracle, which in practice means asking for human feedback, and in response, the agent receives a label $l_i$, which in general gives a hint about the reward. In principle, the query can be made asynchronously to the actual conventional RL cycle. See Fig. 1b for an illustration.

In the most common setting, the oracle can compare two (segments of) trajectories, but various other cases have been considered, as we shall see later on. For the former case, RLHF is based on the setting of preference-based MDPs (Gilbert et al., 2017; Wirth et al., 2017), which can be defined as an MDP model without reward function, but where comparisons of trajectories are available.

## 2.3 Reward Learning

RLHF approaches can be divided into two categories, depending on whether a utility-based approach is used for reward modeling or whether an alternative criterion that is detached from a utility concept is used (Gilbert et al., 2016; Gilbert & Weng, 2016; Wirth et al., 2017). Most works fall into the first category, on which this overview focuses. Such approaches assume a human-dependent utility function that can be used as a reward function in order to apply standard RL methods. Next, we will describe the commonly used approach for reward learning for the common setting of binary trajectory comparisons.

The prevalent approach to learning a utility function from observations of pairwise comparisons is based on the Bradley-Terry model (Bradley & Terry, 1952), which stipulates a probabilistic model for the oracle (human labeler):

$$\mathbb{P}(\tau_1 \succ \tau_2) = \frac{1}{1 + \exp(R(\tau_2) - R(\tau_1))},$$

where $\succ$ means "preferred to" and $R(\tau)$ corresponds to the utility (i.e., return in the context of RL) of $\tau$. Note that this utility function is a kind of surrogate function for the true reward function, which is

(tacitly) assumed to induce the same optimal policy as the true reward function. For a given data set $\mathcal{D} = \{\tau_1^i \succ \tau_2^i \mid i \in [N]\}$, a utility function $R_\psi$ parameterized by $\psi$ can then be learned by the maximum likelihood principle (or equivalently using a cross-entropy loss):

$$\max_\psi \prod_{i=1}^{N} \frac{1}{1 + \exp(R_\psi(\tau_2^i) - R_\psi(\tau_1^i))} \,. \tag{2}$$

In the context of RL, since $R_\psi(\tau) = \sum_{h=0}^{H-1} \gamma^h R_\psi(s_h, a_h)$, (2) can then directly be used to train a function approximator (e.g., single or ensemble of neural networks) to approximate $R$.

This approach accommodates the case of a noisy or unreliable oracle, in which case the Bradley-Terry model can be understood as the generative model of the oracle's answers (or labels provided by the human labeler). When the oracle is reliable, more direct methods based on preference elicitation for recovering the reward function have been studied (Regan & Boutilier, 2009; 2011; Weng & Zanuttini, 2013; Gilbert et al., 2015; Sadigh et al., 2017; Wilde et al., 2018). In this survey, we will focus on the general case where the oracle may be noisy.

Note that, in contrast to the typical way of preference learning, the learned reward function is used to train an RL agent and not directly to compare trajectories. This discrepancy between the objective function used in reward learning and how the learned rewards are actually used may lead to suboptimal policies (Lindner et al., 2021b).

## 2.4 Reinforcement Learning from Human Feedback

In the RLHF setting as illustrated in Fig. 1b, the learning agent needs to solve an RL task without having access to a reward function. To this end, the agent usually simultaneously learns an approximation of the reward function (via the assumed utility function) and an RL policy. Therefore, a generic RLHF algorithm consists of repeating two phases: (1) reward learning and (2) RL training. The first phase can itself be decomposed into two main steps: (i) generate queries to ask the oracle, (ii) train a reward function approximator with the answers provided by the oracle. The RL training part is more conventional and is usually directly based on running a deep RL algorithm using the currently trained reward function approximator.

---

**Algorithm 1** Generic RLHF Algorithm in an Actor-Critic Scheme.

---

1: Initialize parameters $\theta$ (policy), $\phi$ (critic), and $\psi$ (reward)
2: Initialize replay buffer $\mathcal{B}$ with randomly-generated trajectories
3: **for** $i = 1, \ldots, N$ **do**
4:     // Reward learning
5:     Generate queries from $\mathcal{B}$
6:     Update $\mathcal{D}$ with answers to queries from the oracle
7:     Update $\psi$ using $\mathcal{D}$ (e.g., to maximize Eq. (2))
8:     // RL training
9:     Update $\mathcal{B}$ with new trajectories generated with $\pi_\theta$
10:    Update $\theta$ (actor) using $\mathcal{B}$ and $R_\psi$
11:    Update $\phi$ (critic) using $\mathcal{B}$ and $R_\psi$
12: **end for**

---

This basic generic algorithm is summarized in Algorithm 1, where an off-policy actor-critic scheme is assumed to be used for the RL training part, but other RL policy learning approaches can, of course, also be used here. For an on-policy algorithm, such as PPO (Schulman et al., 2017), only the recently generated transitions are used for training. For a DQN-like algorithm, lines 9 to 11 would be replaced by a loop where transitions are generated by a behavior policy based on the current estimate of the $Q$-function (e.g., $\varepsilon$-greedy algorithm) and the $Q$ network is updated using mini-batches of transitions sampled from the replay buffer $\mathcal{B}$.

An efficient RLHF algorithm needs to overcome several difficulties which are specific to this setting:

- The oracle may provide various types of feedback (see Section 3). The questions of what information some given feedback provides and how observed feedback can be exploited need to be answered (see Section 5).

- Informative queries need to be generated to minimize the efforts of the oracle, which is crucial when it is a human (see Section 4). Active learning techniques (see next subsection) can be adapted to face this challenge.

- The RL agent is actually trained in a non-stationary environment since the reward approximator is concurrently updated. The RL training part therefore needs to account for this factor, e.g., using non-vanishing learning rates (see Section 6).

- There is also the question of how the agent's performance can be meaningfully evaluated, especially if the reward function is not known (see Section 8).

- Collecting feedback directly from humans introduces its own challenges, such as the question of a suitable user interface and the associated issues of delay between query and feedback observation, or the feedback variability and reliability (see Section 4). This may explain why many studies evaluate novel RLHF algorithms with simulated feedback.

A standard RL algorithm can be run in the RL training part, as done in most previous work in RLHF (although this may not be the best approach). This suggests that any improvements in a standard deep RL method (e.g., auxiliary losses (Gelada et al., 2019), planning in learned model (Hafner et al., 2020), curriculum learning (Narvekar et al., 2020), or data augmentation (Laskin et al., 2020; Lee et al., 2020; Lin et al., 2020b)) may potentially be transferred to the RLHF setting. In addition, most previous work in RLHF directly uses trajectories stored in replay buffer $\mathcal{B}$ to synthesize queries. An interesting research direction to explore in RLHF would be to specifically generate trajectories in order to be able to synthesize more informative queries (instead of only generating trajectories that are beneficial for RL training). This would lead to tackling a novel exploration-exploitation dilemma: Shall we visit state-action pairs that may be bad but may help better learn the reward function, or shall we visit state-action pairs that we currently think are good? This is further discussed in Section 5.5.3.

In RLHF, since the oracle is a human or a group of humans, reducing the number of queries is crucial to limit the labeling cost. Therefore, the reward learning part requires techniques similar to those proposed in active learning, which we recall next.

## 2.5 Active Learning

In active learning (Settles, 2012), the task is to strategically select data points for labeling to minimize the amount of labeled data required to achieve a desired level of performance, which is particularly valuable in scenarios like RLHF where labeling is costly. Unlike batch learning, where labeled data is predetermined, active learning empowers the learner to actively select the most informative unlabeled instances for labeling, maximizing the learning process with limited labeled data. We will only briefly introduce the active learning task here and then discuss the strategies for creating informative queries considered thus far in Section 4.

For RLHF with pairwise comparisons, this setting can be formally described as follows. Suppose there is a set of $N$ pairs of trajectories $\{(\tau_1^i, \tau_2^i) \mid i = 1, \ldots, N\}$, where each pair $(\tau_1^i, \tau_2^i)$ can be interpreted as an unlabeled instance. To efficiently learn a reward function to explain observed pairwise comparisons, an agent can select a set of unlabeled pairs (possibly a singleton) to query an oracle to obtain their labels.

At a high level, the main idea in active learning is to query data points to quickly reduce the epistemic (i.e., reducible) uncertainty about the predictions of the learned model, although other aspects can be important, such as the representativeness of the queried data points (see Wang et al. (2024b) for a survey). Two main representations are considered to describe this epistemic uncertainty: either using an ensemble of models or using a Bayesian representation. In both cases, a first basic approach selects queries using uncertainty-based criteria in order to focus on instances with high prediction uncertainty as measured by, e.g., variance or entropy computed over predictions. In contrast to the first approach, where the selection criteria are

instance-based, a second approach considers criteria that may depend on all instances. Possible options are, for instance, expected model change, expected error reduction, or density-weighted uncertainty-based criteria. Here, the weights in the expectation or density allow us to take into account the distributional information about the instances and, therefore, to focus on the higher-density regions.

# 3 Feedback

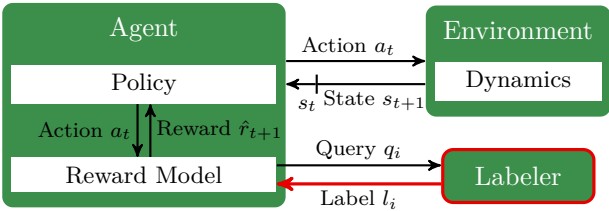

Figure 2: RLHF diagram highlighting components discussed in this section.

Feedback mechanisms are fundamental to the success of any RL system. In the standard setting as described in Section 2.1, RL agents expect feedback in the form of scalar immediate rewards. These rewards are most commonly determined by a hand-engineered reward function, which can be used to evaluate any state-action combination. As discussed in Section 1.1, it is desirable to allow humans to refine behavior interactively through feedback instead of requiring them to pre-specify a reward function.

While a human could, in principle, assign rewards to each of the agent's actions directly, thereby taking the role of the reward function, this is usually impractical for multiple reasons. The main challenge is the human effort required to provide rewards on a sufficiently regular basis, i.e., at least once per episode. In addition to that, directly integrating human rewards into the RL loop would require these rewards immediately, which would impede the learning pace while waiting for human feedback. Finally, the standard RL setting expects numeric state-action rewards, which are challenging to provide in a consistent manner.

In contrast to directly rewarding each of the agent's actions, RLHF as discussed in this survey (see Section 1.3) harnesses indirect and asynchronous feedback methods. Such methods avoid the challenges of immediate numeric rewards and are also better aligned with human interaction patterns, resulting in improved learning progress and human user experience.

A feedback type is a kind of interaction in which a human conveys some information about their preferences. Examples include pairwise comparisons and direct critiques. This section is concerned with the many ways in which human feedback can be expressed and used. Several previous works have already studied and sorted feedback types by listing the most common ones (Jeon et al., 2020; Yuan et al., 2024) and discussing their attributes and dimensions (Metz et al., 2023; Lindner & El-Assady, 2022). The attributes and classes described in this section build upon this prior work and can be considered a synthesis and extension of it.

The remainder of this section will start by discussing relevant attributes of feedback types that can be used to classify them (Section 3.1). We will then discuss common classes and examples of interactive feedback types (Section 3.2) as well as some non-interactive types that can serve as initializations (Section 3.3).

## 3.1 Attributes of Feedback Types

Feedback types may differ in many dimensions, some of which relate to the way feedback is given (arity, involvement), others to the form of the query instance it is given on (granularity, abstraction), and yet others to features of the human interaction (intent, explicitness). The attributes we discuss in this section are based on the framework proposed by Metz et al. (2023). We have adjusted and added terminology where it aids clarity, generalized the distinction between relative and absolute feedback to 'arity', and added the categories of co-generative involvement and literal intent. Furthermore, we systematically analyze a set of exemplary

feedback types in the next section, expanding on the initial examination of a smaller set of abstract classes by Metz et al. (2023). In the following, we will introduce each of the six attributes in more detail.

**Arity** This attribute describes whether a single instance is evaluated in isolation (*unary*) or relative to other instances (*binary*, *n-ary*). Unary feedback, e.g., an absolute score for an observed behavior, is often convenient for detailed and descriptive feedback but lacks any grounding and therefore puts a great burden on the human to provide consistent feedback. Non-unary feedback always has an implicit grounding but requires the instances to be comparable. While *n*-ary feedback, such as a ranking over multiple observed behaviors, can provide more information than binary feedback, it also puts a higher cognitive burden on the labeler.

**Involvement** The labeler may either passively *observe* an instance, actively *generate* it, or coactively participate in its generation (*co-generation*). Passive involvement, exemplified by an evaluator that observes, but does not intervene in a robot's behavior, poses the smallest challenge to the labelers since it does not require the ability to demonstrate the task. It can also easily be directed at the most informative examples with active learning techniques. Unfortunately, passive feedback often cannot match the information density of active feedback such as a human steering a robot to generate a demonstration. It is, therefore, common to combine both types to first initialize the reward model from (possibly very suboptimal) active feedback and then refine it from passive feedback. Between these two extremes is co-generative feedback, in which a human can share control with the agent. This can be less demanding than active feedback and makes it possible to direct the human's attention to the most informative samples, but it is still more taxing than purely passive involvement.

**Granularity** Feedback may also differ on the granularity of the instances being evaluated. This ranges from feedback over whole *episodes* over partial *segments* to feedback on individual *steps* (i.e., states, actions, or state-action pairs). A more coarse-grained granularity has the advantage of giving humans more context and getting feedback for larger sections of behavior but also poses credit assignment problems. Finer-grained feedback is much easier to learn from and simplifies credit assignment, but is often impractical or tedious for humans to provide. It is also possible to strike a compromise, as demonstrated by Guan et al. (2021) who propose to use queries on step granularity, but batch queries within an episode to provide additional context and make it easier to give feedback. Note that we only classify a type of feedback as "episode" granularity if it *requires* entire episodes. If it is compatible with partial segments as well, we classify it as "segment" even if the discussed source paper uses entire episodes.

**Abstraction** This describes whether feedback is given directly on raw *instances*, e.g., behavior recordings (see granularity), or on abstract *features* of the instances, such as an autonomous car's speed and the forces experienced by the passengers. While feature-level information can be easier to learn from, extracting useful features is challenging. In some contexts, it may also be harder for a human to make abstract judgments rather than more intuitive instance-level judgments. Note that this always refers to the level of abstraction that the user sees, which may differ from the features used as inputs for the reward model. Types of feedback that depend on active generation (see involvement), such as improvements, generally work on a raw instance level.

**Explicitness** Humans may communicate *explicitly* for the purposes of feedback, e.g., by responding to a pairwise comparison query, or *implicitly* as a side-effect of actions directed at other purposes such as interventions in the agent's behavior. While explicit information is easiest to learn from, implicit information is often much more readily available and can possibly communicate more detailed preferences.

**Intent** The assumed human intent can be important for feedback processing. A human may be *evaluative*, *instructive*, or *descriptive* in their explicit feedback, while they are generally *literal* in their implicit feedback. Evaluative, instructive, and descriptive feedback is pedagogical in nature, aiming to teach a reward function, whereas literal feedback is a byproduct of a human actor's efforts to optimize the reward function directly. Descriptive feedback is often given on the level of the task, e.g., through

Table 2: An overview of the common classes and their defining attributes. The feedback classes are grouped into primary classes, which can be used on their own to learn a reward model, supplementary (sup.) classes, which can be used in conjunction with a primary class, and representation-focused (rep.) classes, which can be used to learn a representation prior to learning a reward model. When ✚ is specified for an attribute, this indicates that it is not a defining feature of the class and may vary in different instantiations.

|  | Class | Granularity | Involvement | Arity | Abstr. | Intent | Expl. |
|---|---|---|---|---|---|---|---|
| Primary | Critique | ✚ | Observed | Unary | ✚ | Evaluative | Explicit |
| | Comparisons | ✚ | Observed | 2+ | ✚ | Evaluative | Explicit |
| | Inter-Temporal | Segment | Observed | Unary | ✚ | Evaluative | Explicit |
| | Proxy Rewards | Episode | Observed | ✚ | Feature | Descriptive | Explicit |
| | Social Behavior | Segment | Observed | Unary | Instance | Literal | Implicit |
| | Improvements | Episode | Co-generative | Unary | Instance | ✚ | ✚ |
| | Natural Language | ✚ | Observed | Unary | ✚ | Descriptive | Explicit |
| Sup. | E-Stops | Episode | Observed | Unary | Instance | Literal | Implicit |
| | Importance | ✚ | Observed | ✚ | ✚ | Descriptive | Explicit |
| Rep. | Feature Traces | Segment | Active | Unary | Instance | Descriptive | Explicit |
| | Similarity Queries | ✚ | Observed | Ternary | ✚ | Descriptive | Explicit |

a partial reward function. The other types of intent, in contrast, are generally given within the context of a particular instance of behavior.

The distinction between literal and pedagogical feedback was introduced by Milli & Dragan (2020). They argue that humans with pedagogical intent (i.e., evaluative, instructive, or descriptive) may act differently compared to humans with literal intent. Even though they find that assuming the (wrong) literal intent can still lead to better reward inference, it still indicates that it can be important to know this intent to choose the right human model (see Section 5.1.5).

## 3.2 Common Classes

Even though the concrete types of feedback used in the literature are rarely exactly the same, they can generally be sorted into a set of common classes. We will describe a selection of those classes, their defining attributes, and examples of concrete instances in the literature in the following. Table 2 gives an overview of the classes and their attributes as described in Section 3.1.

### 3.2.1 Primary Feedback Classes

This section introduces common feedback classes which can be used on their own to learn a reward model. The classes are critique, comparisons, inter-temporal feedback, proxy rewards, social behavior, improvements, and natural language.

**Critique** Critique is arguably the most direct type of feedback. In this setting, the human expresses their preference by directly critiquing an instance of agent behavior, often in the form of binary feedback. Note that critique, as considered in this survey, is distinct from directly supplying a reward signal since it is given in an asynchronous and indirect manner (see Section 1.3). In the critique setting, a labeler may for example observe recordings of agent behavior and give their approval or disapproval. The defining features of critique are that the human passively observes the behavior (**involvement**), gives feedback on a single instance (**arity**), and does so explicitly (**explicitness**) with an evaluative **intent**. The feedback may be given for any **granularity** and on any level of **abstraction**.

There are many examples of critique feedback in the literature. Xiao et al. (2020) employ binary feedback on individual state and action pairs. Although they learn a shaping reward that complements an environment reward signal, the same technique could be used without environment reward. Huang et al. (2023) extend

this to multi-label feedback, allowing the user to distinguish between a regular good or bad action and a terminal action that achieves the goal or fails to do so. They map these classes to scalar values and then learn a reward model by regression. Wang et al. (2020) present an approach to learning a reward model from noisy critiques in the form of human physiological signals (brain signals) using active querying. In contrast to this action-level feedback, Fu et al. (2018b); Singh et al. (2019) rely on binary outcome success labels. Fu et al. (2018b) introduce the basic approach, which Singh et al. (2019) extend by moving to an off-policy setting and including online queries, thereby reducing the reliance on many positive examples by interactively correcting false positives.

In addition to learning the main reward function, critique can also be used for safety evaluation. Cosner et al. (2022) train a secondary reward model focused on safety from binary action critiques. This is in addition to the main reward model, which is trained from comparisons in their approach. Note that this secondary safety model could, in principle, be trained with any of the feedback types discussed here, using methods identical to the ones used for reward learning.

**Comparisons**   Binary comparisons and rankings are among the most common types of feedback. The defining features of comparisons are that the human passively observes the behavior (**involvement**), gives relative feedback on multiple instances (**arity**), and does so explicitly (**explicitness**) with an evaluative **intent**. It is most commonly given on a segment (**granularity**), but other granularities such as individual actions are also possible (Cosner et al., 2022). Similarly, comparisons are commonly requested on an instance level (**abstraction**), but this is not a requirement.

Comparisons were first used for direct policy learning (Akrour et al., 2011; Cheng et al., 2011), but were later extended to the reward-learning setting (Wirth et al., 2016; Christiano et al., 2017). The most common setting (Christiano et al., 2017) relies on pairwise comparisons of trajectory segments, but comparisons of individual states or actions were also considered in early PbRL works (Fürnkranz et al., 2012) and comparisons can even be extended to more abstract trajectory features (Pinsler et al., 2018).

This basic setting has been extended and modified in various ways. To reduce noise in the labels, it is common to extend the binary choice by giving the labelers the option to avoid the hard choice and instead indicate incomparability, uncertainty, or perceived similarity (Holladay et al., 2016). This option is commonly interpreted as "equally preferable", e.g., as if each trajectory had an equal probability of being preferred in the preference predictor. It is also common, however, to additionally provide an "incomparable" option which results in simply omitting the query from the data set (Christiano et al., 2017; Ibarz et al., 2018). In contrast to this, Verma et al. (2023) explicitly state that they do not allow for the equally preferred option, arguing that these cases are rare enough not to matter very much. Another line of research suggests that more precision in the expression of pairwise preferences, such as softening the hard binary choice to scalar feedback indicating the strength of a preference (Wilde et al., 2021; Touvron et al., 2023), can be beneficial for preference learning. Other extensions change the pairwise setting to choices among larger choice sets (Ziegler et al., 2020) or even full rankings (Myers et al., 2021; Ouyang et al., 2022). Jain et al. (2015) compare different kinds of re-ranking feedback (select first which is better than top, choose best from top-5, choose best from random set). Their study suggests that the first two are roughly equivalent, while the latter is much less informative. Askell et al. (2021) evaluate binary comparisons as well as ranking, but find that binary comparisons (even if extracted from a ranking) perform better. Ziegler et al. (2020) note that for language tasks, a larger choice set can amortize the time needed for a labeler to get acquainted with the context necessary to understand a query. Basu et al. (2019) propose to use hierarchical queries, i.e., a sequence of pairwise comparisons that build up on each other.

Pairwise comparisons are generally easier to supply than ratings, even when the feedback is given by a foundation model (Wang et al., 2024c). While absolute feedback is generally dependent on a policy serving as an implicit baseline (MacGlashan et al., 2017), the explicit baseline in pairwise comparisons can be more easily controlled. Therefore, comparisons provide many benefits over absolute ratings, such as reduced bias, inconsistencies, and subjectivity (Yannakakis & Martínez, 2015). On the flip-side, comparisons convey relatively little information per label. Tien et al. (2023) study the weaknesses of pairwise-comparison-based reward learning. Since the amount of information provided for each label is small, these models are prone to causal confusion, i.e., misattributing reward to noise and misidentifying the reward function.

**Inter-Temporal Feedback**  One limitation of trajectory comparisons is that they require a set of roughly comparable trajectories. In many real-world environments, starting conditions or even the agent's current task may vary between episodes. In these cases, it is hard for human labelers to compare trajectories from these episodes, limiting the usefulness of comparison feedback. One way to remedy this limitation is to provide feedback within a single trajectory. Instead of comparing a set of instances with each other as in regular comparative feedback, inter-temporal feedback conveys relative judgments over different states in time within a single instance, e.g., by comparing a robot's behavior at the start and the end of a trajectory. The defining features of inter-temporal feedback are that it is given explicitly (**explicitness**) on a segment (**granularity**) while passively observing (**involvement**) a single instance (**arity**) of the agent's behavior with evaluative **intent**. It is most commonly given on raw instances, but any level of **abstraction** is possible in principle. There are two main ways to convey this feedback: *Reward sketching* and *inter-temporal preferences*.

Reward sketching, as introduced by Cabi et al. (2020), involves users sketching a visual representation of the reward function over time. This type of feedback, which can be given by sketching a graph with the mouse while watching a behavior recording, provides intuitive reward annotations for each time step. Rahtz et al. (2022) also adopted this approach, referring to it as "one of the highest-bandwidth feedback mechanisms currently available".

Inter-temporal preferences were introduced by Abramson et al. (2022). In this setting, humans give feedback on multiple points of a trajectory, indicating whether an agent makes progress towards or regresses from a goal. This is then interpreted as preferences relative to the other labeled and unlabelled points. The authors note that one potential downside of this feedback type is that labelers may tend to give preferences on short-term actions that are easy to judge, failing to communicate long-horizon preferences. Cui & Niekum (2018) propose a similar type of feedback, in which humans segment a trajectory into good and bad parts. This makes it possible to derive many state-action labels from a few segmentation points.

**Proxy Rewards**  Proxy rewards are partial or inaccurate reward functions that convey information about the task the agent is supposed to complete but may not induce optimal behavior. An example of this is a reward function that only rewards the agent for reaching the goal state but does not penalize negative side-effects. This form of feedback does not generally refer to any particular behavior instance but instead gives global direction for the entire task. However, in line with our selection criteria (Section 1.3), we only consider proxy reward feedback that is interactive and online within the context of one or multiple observations to fill holes in the initial description. The defining features of proxy reward feedback is that the labeler passively observes the agent's behavior (**involvement**) and gives feedback explicitly (**explicitness**) on a feature-level (**abstraction**) with descriptive intent (**intent**). Proxy reward feedback may be given with respect to a single or multiple instances (**arity**), although it generally refers to multiple instances. It is most commonly given on an episode (**granularity**), but other granularities are possible in principle.

The work by He & Dragan (2021) exemplifies this form of feedback through an iterative process where designers provide proxy reward functions on training environments. The method maintains a belief distribution over reward functions consistent with behaviors that achieve high proxy reward on training environments, then actively proposes new environments that maximize information gain about the true reward function, helping designers refine their reward specifications. Alternatively, Mindermann et al. (2018) suggest querying about the reward function, extending the inverse reward design framework of Hadfield-Menell et al. (2017b), which treats proxy reward functions as observations to infer the designer's true intent. The active approach by Mindermann et al. (2018) allows users to choose from a set of understandable, linear proxy rewards or to specify which features are more critical in the linear reward structure. In a related setting, Guan et al. (2023) lets the user specify *changes* to a current symbolic reward function in the form of changes to target attributes (e.g., walking speed).

**Social Behavior**  Humans give rich implicit social feedback in the form of facial reactions and gestures when interacting with agents. The defining attributes of this type of feedback are that it is given implicitly (**explicitness**) on passively observed (**involvement**) segments (**granularity**) with respect to a single instance (**arity**) and literal **intent**.

Cui et al. (2021) propose a framework to learn reward functions from such social behavior. They suggest a two-phase training setup. In the first phase, they ground the implicit feedback by use of incentives, i.e., they incentivize humans to have a known objective. After learning a mapping from feedback to reward, they use regular RL techniques to learn a policy. Note that the learned reward function can be seen as conditional on human implicit feedback and, therefore, they require a human in the loop throughout training.

**Improvements**   Improvements are a form of feedback in which the human improves on the agent's behavior, either by intervening as the agent acts or by providing a corrected behavior after the agent acts. To improve an episode, it is usually necessary to observe the entire episode (**granularity**) at the instance level (**abstraction**). In this type of feedback, the human both observes and demonstrates behavior, resulting in co-generative **involvement**. Improvements generally relate to a single reference trajectory being improved (unary **arity**), although an improvement could also be interpreted as a binary comparison between the improved and the non-improved trajectory. Improvements are most commonly provided explicitly with instructive **intent**.

We distinguish between post-facto improvements, calling them *corrections*, and improvements made while the agent is acting, calling them *interventions*. The key difference is that the uncorrected trajectory is available in the case of corrections, while it can only be estimated in the case of interventions.

Interventions can be considered to be an instance of the shared autonomy setting since the agent and the user share autonomy to reach a common goal. There are two main ways to leverage interventions: One is to learn from the occurrence of an intervention itself that the prior behavior was suboptimal, as done, e.g., by Luo et al. (2024) by directly inferring negative rewards from interventions without learning a reward model, and the other is to learn from the corrected trajectory. The latter setting is studied by Abramson et al. (2022), who ask humans to intercede on agent failure. They use this to collect targeted demonstrations where the agent is the weakest and to identify challenging situations for their evaluations. The gathered data is then used for behavior cloning and reward model training. The fact that a correction occurred is not directly used as feedback. In addition to the above distinction, interventions can come in two main formats: Either by overtaking control from the agent to correct its behavior or by physically correcting the agent's movements. Losey et al. (2022) proposes to learn a reward model from such physical corrections, a method that was later extended (Losey & O'Malley, 2018) to incorporate uncertainty for active learning and risk-sensitive deployment. Li et al. (2021) further extend this setting to learn from a sequence of correlated physical corrections without needing to wait until the trajectory is completed.

The correction case is closely related to the setting of coactive learning (Shivaswamy & Joachims, 2015), in which a learning system repeatedly proposes a solution which a user may correct to reach a common goal. Jain et al. (2015) treat corrections as a demonstration while Jeon et al. (2020) propose (but do not evaluate) an alternative interpretation of inferring implicit preferences from comparisons by assuming the corrected trajectory is preferred over the original one. Corrections are also commonly used in the LLM fine-tuning setting to incorporate AI feedback: By prompting a language model with a set of principles, it can critique and revise a generated response according to those principles. This can then be used to generate demonstrations (Bai et al., 2022b) or targeted pairwise preferences (Castricato et al., 2024) and to internalize these principles with further training on this data.

**Natural Language**   Natural language is a versatile form of feedback that can be used to convey a wide range of information. It cannot only be used to express preferences, but also to suggest concrete changes. Natural language feedback may be given on any **granularity**, at any level of **abstraction**. Its defining features are that it is given explicitly (**explicitness**) in the context of a single (**arity**) observed behavior or policy (**involvement**) with descriptive **intent**.

While much work in language for RL focuses on the problem of learning a policy that follows natural-language instructions in a non-interactive setting (Hermann et al., 2017; Nair et al., 2021), we focus on works where language is used as feedback, not for task definition. Note, however, that RLHF can be a useful tool for learning a language-conditioned policy, regardless of whether or not the feedback itself is in the form of natural language. As an example of this, Abramson et al. (2022) use RLHF in an interactive agents setting, where one agent (responsible for solving problems) is controlled by the learned policy and another agent

(responsible for setting tasks) is controlled by the human. RLHF for LLMs can also be viewed as language-conditioned RL, although Shen et al. (2024) find that reward models in this context often fail to distinguish nuances in task descriptions.

Language may be interpreted as a form of feedback in the reward-rational choice setting (Jeon et al., 2020), where an utterance implies a set of trajectories compatible with the utterance (grounding) and the human can choose an utterance that maximizes the probability of a desired action (i.e., the human is assumed to be specific which leads to pragmatic reasoning). This can be used to infer a reward which may have caused the human to provide the utterances they did provide. A similar approach is followed, e.g., by (Sumers et al., 2022b).

An alternative way of incorporating language feedback is by extracting more structured forms of feedback from the language. An example is to use sentiment analysis to extract information about the reward function from language feedback, as proposed by Sumers et al. (2021).

Language feedback can also be interpreted by an LLM directly, without learning a reward model. This is demonstrated in the concurrent works by Ma et al. (2024b) and Xie et al. (2024) who propose to learn a symbolic reward model in the form of a language-model written piece of (Python) code. They use natural language feedback to improve the reward model based on observations of the agent's behavior induced by the previous version of the reward model.

In addition to the way language feedback is used, the way it is elicited can also have an impact on the learning process. Sumers et al. (2022a) study the relative merits of natural language instructions and descriptions of the desired outcome. They find that instructions tend to work better for low-autonomy settings, while descriptions are more effective in high-autonomy settings.

### 3.2.2 Supplementary Classes

This section introduces two feedback classes, e-stops and importance, that can be used in conjunction with a primary class to learn a reward model. These classes are not sufficient to learn a reward model on their own but can supplement a primary feedback type.

**E-Stops** Emergency stops (e-stops) (Ghosal et al., 2023) are an active type of feedback. In this type of feedback, the human may intervene with the agent's behavior by stopping it, i.e., they may choose to stop the agent's current trajectory at any point, for example to prevent the agent from causing harm. This is closely related to interventions but, in contrast to those, e-stops do not suggest an alternative action. The defining features of e-stops are that the human passively observes the agent's behavior (**involvement**), gives absolute feedback on a single instance (**arity**) on the instance level (**abstraction**), and does so implicitly (**explicitness**) as a side-effect of regular interaction. The **intent** is literal due to the implicit nature. For the purposes of intervention, the human usually observes the full episode (**granularity**). Due to the small amount of infrequent information they provide, e-stops should only be considered as a supplementary feedback type.

E-stops are primarily intended to prevent bad behavior and only implicitly convey information about the correct behavior. This interaction and the arising incentives have been formalized in the form of the "off-switch game" by Hadfield-Menell et al. (2017a). Jeon et al. (2020) propose to interpret this as a form of reward-rational feedback, where the 'off' choice maps to the trajectory with the robot remaining still after the off switch has been triggered (see Section 5.1.1). Kahn et al. (2021) demonstrate that a robot can learn to navigate using such feedback.

**Importance** Another form of supplementary feedback may come in the form of importance labels, communicating which parts of the observation are important for the objective, e.g., by marking salient regions in an image representing the agent's final state or important words in the language model's response. Its defining features are that the importance information itself does not contribute towards generating behavior samples (observed **involvement**), is of descriptive **intent**, and is given explicitly (**explicitness**). **Granularity**, **arity**, and **abstraction** may vary depending on the primary feedback type. Since importance feedback

needs a base task with respect to which the importance is defined, it cannot be used on its own but is rather a supplementary type of feedback.

One way to convey this information is by labeling salient parts of a visual input. This is explored by Guan et al. (2021), who augment pairwise comparisons with manually annotated visual saliency maps, informing the algorithm which parts of the visual input contributed to the decision. They leverage these annotations for data augmentation by assuming that random perturbations to irrelevant (non-salient) regions do not impact the human preferences. Basu et al. (2018) take an even more direct approach by combining comparative feedback with direct feature queries, i.e., asking the user which feature is important for inferring the reward.

### 3.2.3   Representation-Specific Classes

While the previous classes of feedback types are all aimed at directly learning a reward function, there are also classes of feedback types that do not directly learn a reward function but rather help to learn a better representation.

**Feature Traces**   While many approaches either rely on hard-coded features or learn a model entirely end-to-end, it is also possible to actively elicit new features from human feedback. Feature traces were proposed by Bobu et al. (2022) as an approach to actively learn new relevant features. This type of feedback relies on a human operator to demonstrate a behavior in which a certain feature of interest, such as the distance to a sensitive object, monotonically increases or decreases, e.g., by moving a robot arm closer to the object. They make it possible to extend the set of features once the current set can no longer adequately explain the human feedback supplied through another type of feedback. The defining characteristics of feature traces are that they are of descriptive (**intent**) and explicitly (**explicitness**) given in an active manner (**involvement**) for a single (**arity**) segment (**granularity**) on an instance-level **abstraction**.

Feature traces are strongly related to inter-temporal preferences (Section 3.2.1) since both types rely on changes in feature or reward values in the course of a single trajectory. Bobu et al. (2022) propose to learn from feature traces by leveraging a Bradley-Terry model to learn the feature values, similar to other approaches that use such a model to learn reward values. Similar to importance feedback, feature traces rely on another type of feedback to actually make use of the learned features and is, therefore, a purely supplementary form of feedback. For instance, Bobu et al. (2022) use intervention feedback to train a reward model on the set of features derived using feature traces.

**Similarity Queries**   Similarity queries are a feedback type aimed at learning a representation conforming to a notion of similarity and difference in the trajectory space. That aim is closely aligned with that of feature queries, though the actual queries are more similar to comparisons. The queries consist of triples of trajectories, with one anchor and two alternatives, for which the human has to decide which pair is more similar. Responses to similarity queries are given on observed behavior (**involvement**) with ternary **arity**, descriptive **intent**, and explicit feedback (**explicitness**), while the **granularity** and **abstraction** may vary. This type of feedback was first introduced by Bobu et al. (2023), who used it to learn representations for reward learning.

### 3.3   Initializations

Some modes of communicating reward functions are not interactive nor online and, therefore, do not directly fit within the scope of this survey (Section 1.3). However, since these are often used to initialize a reward function for later refinement with some of the previously discussed interactive feedback types, they are still worth mentioning.

Initializations are most commonly given by examples of successful task completions, either in the form of terminal or goal states (Xie et al., 2018), expert demonstrations (Ibarz et al., 2018; Fu et al., 2018a; Palan et al., 2019; Lee et al., 2021b; Bıyık et al., 2022a; Abramson et al., 2022; Huang et al., 2023), demonstrations with success labels (Du et al., 2023), or ranked demonstrations (Brown et al., 2019). It is even possible to infer some human preferences by the state of the environment alone, e.g., assuming the parts of the initial environment that are under the user's control are largely set up in accordance with the user's preferences

(Shah et al., 2019; Lindner et al., 2021a). Since offline initialization is not the main focus of our survey, we do not cover these works in detail. We refer the interested reader to literature on inverse RL (Arora & Doshi, 2021) for further details on learning from demonstrations in particular.

## 3.4 Choice of Feedback Type

The best feedback type is not always clear and may depend on the task, the user, or the agent. It may also change over time as the agent learns more about the task. Table 2 may serve as a starting point to select a set of feedback types which may be applicable based on the possible user interaction and expertise, e.g., by the desired granularity, level of abstraction or involvement. This choice may be informed, among other features, by the task complexity or time horizon: Jain et al. (2015) compare purely passive (re-ranking based) with active (slight trajectory improvements) feedback and conclude that the former is usually preferred by humans when the action space is large while the latter is preferred for complex tasks. Similarly, Sumers et al. (2023) empirically demonstrate that language is a more effective teaching modality than demonstrations for complex concepts. The relevance of the time horizon is highlighted in the work by Sumers et al. (2022b), which suggests that instructive feedback is more efficient for short time horizons and descriptive feedback for longer horizons.

In addition to these static choices, Section 4.1.2 will discuss how to choose a feedback type adaptively. It is also possible to let the user choose the feedback type, as demonstrated by Jain et al. (2015) who let the user choose between re-ranking and improvement. Jeon et al. (2020) propose to use this choice of feedback type itself as a source of information ("meta-choice").

## 3.5 Combination of Feedback Types

In addition to using any of the previously described feedback types in isolation, combining multiple feedback types is both possible and often advantageous. There are three main ways to combine feedback types: (a) a two-phase setup, consisting of initialization and refinement, (b) integrating a primary feedback type with a supplementary one; and (c) merging multiple primary feedback types.

The two-phase setup can be used to either initialize the reward model from offline data or to learn a representation that improves later reward learning. A common approach involves using one feedback type, typically demonstrations (see Section 3.3), to initialize the reward model, subsequently fine-tuning this model with another type, such as comparisons. This method is exemplified by Ibarz et al. (2018), who combined demonstrations and pairwise comparisons. For a more detailed discussion on feedback types suitable for initialization, we refer to Section 3.3. Alternatively, a representation-focused type of feedback might be employed initially (Section 3.2.3) to cultivate a superior representation, followed by the application of a primary feedback type for reward model learning.

Combining a primary feedback type with a supplementary one can be beneficial to make the most of the available user interactions. Supplementary feedback, while not sufficient for reward model training by itself, can often be collected cheaply or as a side-effect of other interactions, making it a valuable addition to the primary feedback type. We refer to Section 3.2.2 for a discussion on supplementary feedback types.

Finally, combining multiple primary feedback types can be beneficial to capitalize on the strengths of each. For instance, Koppol et al. (2020) combine informative and demanding queries with less-informative and less-demanding ones to achieve a balance between cognitive load and informativeness. Similarly, to enhance expressivity, Mehta & Losey (2023) combines demonstrations, pairwise comparisons, and corrections, allowing users to select their preferred type of feedback.

Table 3: An overview of query selection strategies used in RLHF approaches.

| References | Uncertainty | On-policy Data | Query Simplicity | Trajectory Quality | Query Diversity | Query Cost |
|---|---|---|---|---|---|---|
| Daniel et al. (2014) | Probability of improvement | ✗ | ✗ | ✗ | ✗ | ✗ |
| | Expected improvement | ✗ | ✗ | ✗ | ✗ | ✗ |
| | Upper confidence bound | ✗ | ✗ | ✗ | ✗ | ✗ |
| Christiano et al. (2017) | Ensemble variance | ✗ | ✗ | ✗ | ✗ | ✗ |
| Sadigh et al. (2017) | Volume removal | ✗ | ✗ | ✗ | ✗ | ✗ |
| Bıyık et al. (2024) | Volume removal | ✗ | ✗ | ✗ | ✓ | ✓ |
| Wilde et al. (2018) | Feasible space reduction | ✗ | ✗ | ✓ | ✗ | ✗ |
| Ibarz et al. (2018) | Random | ✗ | ✗ | ✗ | ✗ | ✗ |
| Mindermann et al. (2018) | Information gain | ✗ | ✗ | ✗ | ✗ | ✗ |
| Cui & Niekum (2018) | Information gain | ✓ | ✗ | ✓ | ✗ | ✗ |
| Racca et al. (2019) | Entropy | ✗ | ✓ | ✗ | ✗ | ✗ |
| Bıyık et al. (2019) | Mutual information | ✗ | ✓ | ✗ | ✗ | ✗ |
| Bıyık et al. (2020) | Information gain | ✗ | ✗ | ✗ | ✗ | ✗ |
| Reddy et al. (2020) | Ensemble-averaged KL-divergence to mean output | ✗ | ✗ | ✓ | ✓ | ✗ |
| Novoseller et al. (2020) | Dueling posterior sampling | ✗ | ✗ | ✓ | ✗ | ✗ |
| Wilde et al. (2020; 2021) | Maximum regret | ✗ | ✗ | ✗ | ✗ | ✗ |
| Lee et al. (2021b) | Ensemble-averaged entropy | ✗ | ✗ | ✗ | ✗ | ✗ |
| Lindner et al. (2021b) | Information gain | ✓ | ✗ | ✗ | ✗ | ✗ |
| Katz et al. (2021) | Posterior sampling | ✗ | ✗ | ✓ | ✗ | ✗ |
| Myers et al. (2023) | Expected value of information | ✗ | ✗ | ✗ | ✗ | ✗ |
| Bıyık et al. (2024) | Mutual information | ✗ | ✗ | ✗ | ✓ | ✗ |
| Dwaracherla et al. (2024) | Double Thompson sampling | ✗ | ✗ | ✓ | ✗ | ✗ |
| Hu et al. (2024b) | Random | ✓ | ✗ | ✗ | ✗ | ✗ |

# 4 Label Collection

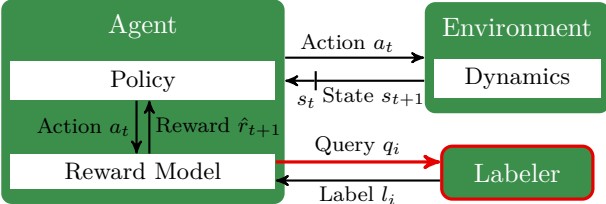

Figure 3: RLHF diagram highlighting components discussed in this section.

In this section, we explain how preference data can be collected for training a reward model, independent of the specific type of feedback used. We start by overviewing the active learning problem posed by query selection, e.g., how queries can be generated for a given query type and how even the type of feedback itself can be selected. Following this, we discuss issues arising in such human-computer interaction.

## 4.1 Active Learning

We first discuss query generation and selection for a given feedback type, then the extension of these techniques to the choice of feedback type.

### 4.1.1 Query Generation and Selection

One core problem that needs to be tackled in RLHF is that of learning about the human's preferences. This problem shares some similarities with the active learning setting since the agent can actively query a human teacher about those preferences. However, in contrast to standard active learning, which usually assumes a supervised learning setting, in RLHF, the agent needs to solve this problem in the context of RL. This means that it can both influence the distribution of the data (i.e., the transitions) and decide which data should be labeled.

As the RL agent is trained and its learned policy changes, the trajectories it generates will naturally evolve. Most work directly uses the trajectories obtained during RL training for preference learning. Using such trajectories, candidate queries are often generated randomly. However, for pairwise comparisons, a more efficient approach is to generate queries by exploiting preference transitivity (Hwang et al., 2023). Alternatively, the agent can also generate trajectories specifically to be used for querying (not necessarily for RL training), possibly with a learned transition model to limit the sampling cost (e.g., Reddy et al. (2020); Liu et al. (2023b)). This kind of active generation of queries can possibly lead to more informative ones.

In order to efficiently learn a suitable reward model, the agent must generate and select queries (Line 5 from Algorithm 1) so that it can quickly learn a good strategy using those queries. This selection is performed via a criterion, usually called *acquisition function*[2], which allows the queries to be compared. Although most existing work in RLHF uses an acquisition function that provides some measure of *uncertainty* (e.g., about the learned rewards), which is arguably one of the most important factors in active learning, an efficient acquisition function (see Table 3 for works dedicated to improving query selection) may need to include various additional other aspects, such as: *on-policy data*, *query simplicity*, *trajectory quality*, *query diversity*, or *query cost*, which will be discussed one by one in the following. As a side note, interestingly, as highlighted by Habibian et al. (2022), the queries asked by an RL agent may also reveal its current reward learning stage.

**Uncertainty** This factor usually corresponds to epistemic uncertainty (Hüllermeier & Waegeman, 2021), which represents how uncertain the agent is about the ground-truth reward function. Epistemic uncertainty can be contrasted to aleatoric uncertainty, which describes inherent stochasticity in the system. While the latter cannot be fully eliminated, the former can be reduced with additional queries. Uncertainty is usually one of the most important aspects to consider when deciding which query to ask. It is usually represented either as a probability distribution (i.e., belief) in Bayesian approaches or using an ensemble of reward networks to approximate this belief. However, other representations are also possible, e.g., using the recently-proposed epistemic neural network (Osband et al., 2023).

With a belief representation, various classic acquisition functions have been considered in the Bayesian framework, such as the *probability of improvement*, the *expected improvement*, or the *upper confidence bound*. For instance, Daniel et al. (2014) compare those three criteria in a robotic domain and observe that the latter yield the best performance, but at a high cost in terms of number of queries, while the first asks the fewest number of queries, but with a slightly lower asymptotic performance. Other alternative criteria have been considered, such as *volume removal* (Sadigh et al., 2017; Basu et al., 2018; 2019) or *information gain* (Mindermann et al., 2018; Bıyık et al., 2019; 2020). The *volume removal* criterion uses the minimum volume of the hypothesis set removed by an answer to a query as the acquisition function and has been shown to be effective in practice. However, Bıyık et al. (2019) show that volume removal has uninformative global optima and argue that the practical effectiveness is due to non-convexity leading to local optima that are informative. They also show that optimizing for information gain has the same computational complexity while avoiding this flaw. One drawback of these Bayesian approaches is that they require maintaining a distribution over reward functions (e.g., using Gaussian processes (Daniel et al., 2014) or simpler probability distributions, such as Gaussian distribution, but with a linear reward model) and, therefore, may not be suitable for more complex domains due to the computational complexity or the strong assumption about the reward model.

---

[2]We follow the terminology used in Bayesian active learning.

When using an ensemble instead of a direct belief representation, these criteria for epistemic uncertainty reduction correspond to measures of disagreement within the ensemble. Previous criteria could possibly be applied, but one popular candidate is the *variance* of the ensemble outputs (Lee et al., 2021b; Metcalf et al., 2023; Gleave & Irving, 2022) or equivalently its standard deviation (Eberhard et al., 2022).

The *average entropy* of the ensemble outputs (e.g., when assuming a Bradley-Terry model for the human answers) has also been used (Lee et al., 2021b;a; Park et al., 2022). However, note that it does not quantify epistemic uncertainty but rather the aleatoric uncertainty in the human's answers, as provided by the response model of the human. Therefore, this criterion may not be suitable in the RLHF setting since it amounts to focusing on the queries for which an answer is expected to be the most random (according to the Bradley-Terry model). By definition of this model, the segments in the pairwise comparisons are the most similar in terms of returns and are, therefore, the hardest to answer for the human.

In contrast to those acquisition functions that lead to deterministic query selection, sampling-based approaches have also been studied, from pure random selection (Ibarz et al., 2018) to Thompson sampling (Katz et al., 2021) and its variants (Novoseller et al., 2020; Dwaracherla et al., 2024). Recently, in the context of fine-tuning LLMs with pairwise comparison queries, Dwaracherla et al. (2024) shows that using the relatively novel epistemic neural network (Osband et al., 2023), double Thompson sampling (Wu & Liu, 2016), which naturally favors better elements to be compared, performs well experimentally.

In addition to epistemic uncertainty, one may also take the outcomes into account to select queries, that is consider utilities (i.e., returns or expected returns in RL). In a Bayesian setting, this leads to acquisition functions such as *expected value of information* (Myers et al., 2023) or *information gain over return differences* (Lindner et al., 2021b), while in a non-Bayesian setting, the notion of *regret*, which measures the difference of performance between a policy optimal for the ground-truth reward function and a policy optimal for a learned reward function, can be used (Wilde et al., 2020).

Finally, it should be mentioned that naturally approaches with theoretical guarantees usually also use uncertainty as the main criterion for label collection (e.g., Novoseller et al. (2023), see Section 7 for more details).

**On-Policy Data** Only focusing on uncertainty is likely insufficient or inefficient in RL because the previous methods may focus on choosing queries to identify the reward function as precisely as possible uniformly on the whole state-action space. However, it may be important to favor more on-policy trajectories to guarantee the relevance of the generated queries for the current policy, assuming that the behavior policy is stochastic (i.e., the current stochastic policy in SAC or the current deterministic policy with some noise, such as in DDPG). Indeed, improving reward learning in state-action regions that may never be visited with the current policy would lead to wasteful queries (Lindner et al., 2021b). One simple approach to ensure that the data is more on-policy is by simply sampling from the current policy (Cui & Niekum, 2018) or favoring more recently-generated trajectories (Hu et al., 2024b).

**Query Simplicity** Selecting queries only based on their informativeness may lead to queries that are hard for a human to answer, which is, for example, the case for the average entropy. The ease of answering a query is important to alleviate the cognitive load of the human oracle. Some work specifically takes this aspect into account, for instance, by considering the similarity of consecutive queries (Racca et al., 2019) or the information gain. For this latter criterion, Bıyık et al. (2019) show that in contrast to volume removal, it naturally leads to queries that are easier to answer for a human because information gain can be increased when the uncertainty in the human answer is lower.

**Trajectory Quality** Most approaches directly use the trajectories generated during RL training. Especially early in training, these can be very bad with respect to the ground-truth reward function. In addition to that, they can be irrelevant or even contradictory for a given task (Katz et al., 2021). Building queries on such trajectories may lead to unnatural queries for a human to respond to, such as comparing a very bad trajectory with an irrelevant one. Katz et al. (2021) measure trajectory

quality by optimizing over sampled reward functions. Similarly, Cui & Niekum (2018) generate trajectories using optimal policies for reward functions sampled from the current Bayesian belief.

**Query Diversity** When asking many queries (in batch, in sequence), the diversity of the queries becomes especially crucial to avoid asking redundant queries. Most work (Christiano et al., 2017; Lee et al., 2021b; Verma & Metcalf, 2024) follows a very myopic approach: Queries are often selected from a usually randomly-generated set of potential queries, and sequences of queries are not really coordinated. While some work exists that specifically tackles the selection of a batch of diverse queries (Bıyık & Sadigh, 2018; Bıyık et al., 2024), the latter is rarely considered due to its computational intractability. Indeed, planning ahead a sequence of queries would amount to solving a sequential decision-making problem under uncertainty over a combinatorial action space (i.e., the set of possible queries). For diverse batch querying, previous work considered using clustering methods such as k-medoids (Bıyık & Sadigh, 2018) or more recently determinantal point processes, which define probability distributions that promote diversity (Bıyık et al., 2024). In addition to approaches focused on selecting diverse queries, another promising research direction involves generating inherently diverse and informative behaviors to select from. This can be achieved with techniques such as entropy-regularized RL (Ziebart et al., 2008; Haarnoja et al., 2017) or generative methods based on flow networks (Bengio et al., 2021). The latter is particularly relevant for RLHF for generative models such as LLMs (Hu et al., 2024a). To our knowledge, this direction has not yet been explored in the context of RLHF.

**Query Cost** The cost of generating queries may also be an important factor if the interaction of the human is live since it may not be practical to let the human wait before showing any queries (Bıyık et al., 2024). In that case, it may be more important to quickly show some relatively good queries instead of computing the most informative ones. Although this factor may not translate directly into an acquisition function, it may influence the choice of the acquisition function and its implementation in a given problem.

Since various different acquisition functions have been considered, some effort (Lee et al., 2021b;a) has been made to compare them. Generally speaking, uncertainty-based criteria (e.g., variance or average entropy) seem to often perform better empirically compared to random selection, a query diversity-based criterion alone or combined with an uncertainty-based criterion. Surprisingly, random selection has been shown to perform competitively in some cases (Christiano et al., 2017; Ibarz et al., 2018). Thus, a better understanding of which acquisition function should be preferred in which situation or domain is still an open question.

In addition, combinations of different criteria have naturally also been evaluated. For instance, Reddy et al. (2020) use four acquisition functions (high uncertainty, high novelty, high reward, low reward) in parallel. This approach has also been validated in a 3D environment (Rahtz et al., 2022). A more sophisticated approach consists of considering a portfolio of acquisition functions and learning to select them using a multi-armed bandit approach (Hoffman et al., 2011).

Various extensions to the basic setting have also been investigated. In the context of multiple human labelers, the issue of selecting reliable teachers to query arises (Daniels-Koch & Freedman, 2022). Assuming all teachers have the same preferences, this can be modeled for pairwise comparisons by incorporating a rationality coefficient $\beta$ into a Bradley-Terry model and estimating this factor:

$$\max_{\psi} \prod_{i=1}^{N} \frac{1}{1 + \exp(\beta(R_{\psi}(\tau_2^i) - R_{\psi}(\tau_1^i)))} \,, \tag{3}$$

where a higher $\beta$ corresponds to a more reliable human (see Section 5.1.2). The setting in which this assumption does not hold, i.e., the labeler's reward functions differ (a setting already considered in inverse RL (Choi & Kim, 2012)), has also been studied recently (Siththaranjan et al., 2024; Xue et al., 2024; Dong et al., 2024b; Myers et al., 2021; Bakker et al., 2022). Interestingly, a noisy oracle may sometimes provide more information than a completely reliable oracle. For instance, in Eq. (3), the probability of erroneous answers given by the noisy oracle is related to how much a segment is preferred to the other one, and Chan et al. (2021) show that such Boltzmann-rational behavior can be more informative for reward inference than

perfectly rational behavior. In contrast, only a binary preorder over segments can be inferred from the answers of a reliable and deterministic oracle, which may not be enough to recover the true reward function.

Another notable recent work is that by Ellis et al. (2024) who raise the issue of identifiability of the ground-truth reward function: many reward functions result in the same optimal behaviors. The authors propose a framework that enables the generation of acquisition function for various definitions of reward similarity, such as the one discussed in Section 5.3.

### 4.1.2 Adaptive Choice of Feedback Type

In addition to selecting queries within a given feedback type, it is also possible to actively select the feedback type itself (Fitzgerald et al., 2022). The best choice of feedback type can depend on many factors, such as human rationality as well as task-dependent factors, some of which may change during the labeling process. Ghosal et al. (2023) formalize this setting as one in which we try to select a feedback design (or feedback type) $x$ out of the space of possible designs $\mathcal{X}$ such that the expected information gain over the distribution of reward functions is maximized for the next human response. Concretely, the goal is to choose a feedback design by means of

$$x = \underset{x \in \mathcal{X}}{\arg\max} \, \mathbb{E}_{c_h \sim P(c_h | x)} \big[ D_{KL} \big( \mathbb{P}(\theta \mid c_h, x) \mid \mathbb{P}(\theta) \big) \big],$$

where $c_h$ is the human response to a query defined by $x$ and $\mathbb{P}(\theta)$ is the prior distribution over reward functions.

Ghosal et al. (2023) find that the most informative feedback type depends on the (type-dependent) rationality of the human labeler (see Section 4.2.1). More precisely, it is shown that the most informative feedback depends on the rationality factor, e.g., while demonstrations are more informative than comparisons when the human is highly rational, comparisons should be preferred in less-rational settings. Given that this rationality might change due to factors such as fatigue or an individual labeler's capabilities, this suggests that adaptively adjusting the feedback type during the labeling process may be worthwhile. Further study of this relationship is a promising area for future work.

## 4.2 Challenges of Human Labeling

This section explores the label collection process, which follows after query selection. This task intersects with several related disciplines, especially within the social sciences, as it encompasses the design of interactions to facilitate informative query responses. A prominent field in this area is psychometrics (Furr, 2021), which focuses on measuring psychological attributes, including preferences. Similarly, survey research (Fowler, 2013) is dedicated to developing techniques for gathering information from individuals via surveys. Human-computer interaction plays a significant role as well, investigating the design of user interfaces tailored for preference elicitation (Pommeranz et al., 2012). Moreover, preference label collection is also necessary for discrete choice experiments within health economics (Ryan et al., 2008), where it is used for the assessment of service values.

### 4.2.1 Psychology-Aware Preference Elicitation

Understanding human psychology is essential for effective preference elicitation in RLHF systems. Human decision-making is complex, often diverging from traditional rational choice models due to cognitive, social, and emotional factors. This complexity is exemplified by phenomena like fatigue, which can affect the reliability of choices based on the order of queries. This section overviews these phenomena, exploring how constructive preferences, biases, framing effects, and social interactions shape the observed choices. Recognizing and addressing these psychological underpinnings is key to developing more accurate and reliable systems. In this section, we will discuss various psychological phenomena, such as cognitive biases and response biases, and related effects (fallacies, biases, heuristics, psychological phenomena impacting decision-making processes), which may falsify labels by adding systematic bias or noise.

Preference learning methods typically assume the existence of inherent, stable preferences that can be elicited through querying. Contrary to this assumption, psychological research, such as the work by Lichtenstein

& Slovic (2006), indicates that preferences are often constructed during the elicitation process and may vary with the method of elicitation or over time. This suggests that the feedback type not only affects elicitation's effectiveness but also shapes preferences. Systematic biases, noise, and other psychological factors may influence observed choices, challenging the traditional models of human choice used to infer latent utilities (see Section 5.1). The elicitation method, query presentation, and context thus play a critical role in shaping measured preferences, compounded by cognitive biases and irrationalities.

The influence of psychological phenomena on preference learning has been well-documented in the literature, especially within the context of explicit preference elicitation for recommender systems. For instance, Tran et al. (2021) provide a thorough discussion of the relationship between psychology and recommender systems. Similarly, Atas et al. (2021) review how preference construction is influenced by cognitive biases, personality traits, and emotional states in recommender systems, discussing effects like serial position, framing, anchoring, choice overload, and preference visibility. In a more specialized discussion, Mandl et al. (2011) focus on cognitive biases in the context of consumer decision-making and its interaction with recommender systems. Finally, Kaufmann et al. (2023) link these psychological aspects to RLHF, discussing the common practice of using synthetic instead of real human feedback for algorithm evaluation and highlighting the limitations of that approach. They further discuss challenges posed by real human feedback, many of which are related to the concepts discussed in the following paragraphs, as well as the opportunities provided by integrating psychological insights into RLHF systems.

Constructive preferences are closely related to *framing effects*, which refer to changes in elicited preferences based on how tasks or alternatives are described, even when these descriptions are essentially equivalent. For example, presenting a choice as a loss versus a gain can lead to different decisions despite identical outcomes. Moreover, *serial position effects*, commonly known as primacy and recency effects, also play a significant role. These effects describe the tendency for the beginning and end of an experience to influence subjective experience disproportionately. This phenomenon becomes particularly relevant in scenarios like video choices, where the initial or concluding segments might disproportionately affect preferences. Atas et al. (2021) discuss both of these effects in the context of recommender systems.

*Ordering effects* pose another challenge in preference elicitation, where the sequence of queries can affect responses. Day et al. (2012) outline several factors contributing to these effects: institutional learning, changing preferences, and varying levels of cognitive effort. Institutional learning involves gaining familiarity with the task and feedback type, which can enhance labelers' expertise and, consequently, the accuracy of their responses. However, due to the constructive nature of preferences, these preferences may evolve during the elicitation process, leading to changing preferences. This evolution might also be influenced by *anchoring effects*, where previously seen instances bias responses. Furthermore, cognitive effort levels can fluctuate due to factors like fatigue or boredom. This is closely related to *choice overload*, a form of fatigue from excessive choices, as discussed by Atas et al. (2021) and bounded rationality, as explored by Chen et al. (2013). In such scenarios, labelers might opt out of making a choice when overwhelmed by options. Bounded rationality refers to the limitations in human decision-making capabilities, particularly when processing large amounts of information, which aligns with the concept of choice overload. To address these challenges, studies like Bıyık et al. (2019) and Zhang et al. (2022) propose methods to reduce cognitive effort in responding to queries. Bıyık et al. (2019) focus on posing queries that are straightforward for humans to answer, while Zhang et al. (2022) enhance the human evaluation process by presenting queries in a user-friendly format.

Multiple labelers often collaborate on the same task in preference elicitation, as studied, e.g., by Barnett et al. (2023) and Daniels-Koch & Freedman (2022). This collaboration may lead to another set of biases if they have the opportunity to exchange information. The exchange may either be direct or indirect through observing the system's predictions, which are based on the other labeler's feedback. Such interactions can affect their preferences through several mechanisms, as identified by Atas et al. (2021): anchoring effects, transfer of emotional states, and conflict avoidance. *Anchoring effects*, for instance, occur when a labeler's choices are influenced by the knowledge of others' preferences or system predictions, a phenomenon also discussed under the term *preference visibility*. This bias can lead labelers to align their preferences with the anchors they are exposed to, which is a significant consideration in recommender systems. Understanding these biases is crucial for designing RLHF systems that mitigate the influence of labeler interactions on preference construction.

The effects previously discussed stem from systemic biases in preference expression. In addition to these biases, choices may also be affected by noise. This is commonly discussed under the term stochastic rationality, where an agent's behavior is rational with respect to an unobserved random state. The reward-rational implicit choice framework, as introduced by Jeon et al. (2020), addresses this by integrating a rationality factor $\beta$ into the human choice model (see Eq. (3)). This factor's impact has been further examined by Ghosal et al. (2023) through synthetic experiments and user studies, demonstrating that accurately estimating this type-dependent rationality coefficient can enhance learning performance and guide feedback type selection (see Section 4.1.2). However, a practical method for estimating this factor remains a challenge. While Ghosal et al. (2023) use calibration feedback with a known latent utility function for estimation, such an approach is not feasible for most tasks. In a related study, Daniels-Koch & Freedman (2022) investigate a scenario with multiple teachers, focusing on the agent's ability to select the most knowledgeable or rational teacher. Therefore, developing more advanced methods to estimate this factor, along with understanding its variability due to factors like fatigue or other ordering effects, presents a vital area for future research in preference elicitation.

Finally, the quality of human feedback is biased towards factors that are easy to judge. Hosking et al. (2024) demonstrates that in the case of LLM fine-tuning, humans tend to favor assertiveness over factuality, since the latter is hard to judge without external assistance or resources. A similar phenomenon was observed in the control setting by Amodei et al. (2017), where the agent learned a behavior that looked good only from the camera angle that the human labelers had access to.

Incorporating psychological insights into the preference-learning components of RLHF systems is essential for optimizing their efficacy. A key area of focus should be research aimed at mitigating biases and harnessing cognitive aspects of preference formation. For instance, designing user interfaces that minimize framing effects and developing algorithms that account for ordering and serial positioning are crucial steps. In this realm, Metz et al. (2023) and Yuan et al. (2024) each propose a configurable user interface for studying various feedback types and their combinations. Additionally, the study by Krening & Feigh (2018) on the impact of feedback type, such as binary critiques versus action advice, on task performance and labeler satisfaction highlights the significant role of feedback type in preference elicitation. Furthermore, the work of Pommeranz et al. (2012) in user-interaction design underlines the importance of having an expressive feedback type to increase user engagement.

The integration of these research findings into RLHF systems points to a clear need for a more multidisciplinary approach. Drawing insights from related fields like behavioral economics and psychology can provide valuable methodologies and perspectives. Addressing irrational choice patterns and enhancing the quality of human feedback remain critical challenges. As we continue to develop and refine these systems, the focus should be on creating robust frameworks that align learning processes with human behavior, effectively managing the inherent complexity and variability of human feedback.

### 4.2.2 Importance of Researcher-Labeler Agreement

High-quality labels are important for the final policy in an RLHF process. Early work on fine-tuning language models using RLHF noticed a mismatch between the researcher's goals and the (paid) labeler's actual labels (researcher-labeler disagreement). Ziegler et al. (2020) note that researchers agreed with each other about 60% of the time (on 4-way comparisons, where random choice would result in 25% agreement), while agreeing with labelers only 38% or 46% of the time (depending on the task). Stiennon et al. (2020) attempt to reduce these disagreements by maintaining a hands-on relationship with the labelers, thereby ensuring high researcher-labeler agreement. Concretely, they provide on-boarding with detailed instructions, keep an open channel of communication between researchers and labelers, and give feedback to the labelers. They evaluate the researcher-labeler agreement and reach an agreement rate of $77\% \pm 2\%$.

Perfect labels are often impossible due to the inherently subjective nature of the task. Returning to the example given by Stiennon et al. (2020), different researchers agreed with each other in only $73\% \pm 4\%$ of the cases. Ouyang et al. (2022) also report the agreement rates on a different task (instruction fine-tuning instead of summarization) and find that labelers agree with each other in $72.6 \pm 1.5\%$ of the time, after a screening procedure that, amongst others, selects labelers that agree with researcher labels. Preferences can

additionally be inconsistent between feedback types, as demonstrated by the findings of Bansal et al. (2024), which show that preferences inferred from ratings and rankings significantly disagree for both human and AI annotators.

The importance of quality does not trump the importance of quantity, however. Indeed, Stiennon et al. (2020) note that excluding low-confidence samples from the data set generally did not help with reward model training. This indicates that even though quality is important, a larger quantity is still generally better.

The scale of the labeled data set required for effective training and refinement varies widely, impacting the quality of the resulting models. In practice, the required amount of data depends on many factors including domain, task complexity, and level of pretraining. For example, while LLM fine-tuning generally often requires tens of thousands of samples due to the complexity of the task, at the same time it can leverage the extensive prior knowledge of the pretrained LLM to reduce sample complexity compared to other tasks of similar complexity. Studies have shown a broad range in data set sizes, from tens of labels in smaller studies (Jain et al., 2015) to hundreds in more complex scenarios (Christiano et al., 2017). Larger-scale applications may require thousands (Guan et al., 2021; Ouyang et al., 2022) or even millions of labels (Abramson et al., 2022), each bringing its own challenges in ensuring label accuracy and consistency.

This variability in data set size underscores the need for rigorous label quality control measures across different scales. In smaller data sets, each label carries more weight, making accuracy and precision critical. Conversely, in larger data sets, the challenge lies in maintaining consistency and mitigating systematic biases that might emerge from the sheer volume of data. The labeling setting varies in the surveyed works, from author-provided feedback (Kim et al., 2023), over small in-person studies (Katz et al., 2021), to larger remote studies (Kim et al., 2023). Each setting provides unique challenges to ensure high-quality labels.

Various works have suggested measures to improve label quality. Hagendorff & Fabi (2022) discuss the possible failure modes of the labeling task in more detail, for example, discussing systematic biases and conflicting motivation, and propose concrete changes to the training and evaluation methodology to alleviate these. Glaese et al. (2022) suggest providing labelers with multiple natural language rules and collecting preference labels for each rule individually to improve label quality. This is related to Bai et al. (2022b), who propose to generate feedback automatically based on such a set of rules and a language model.

## 5 Reward Model Training

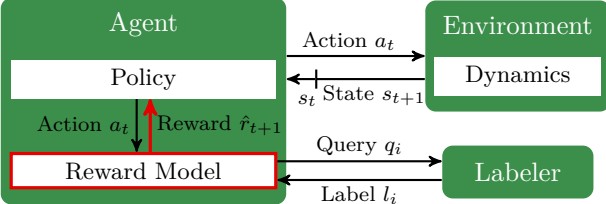

Figure 4: RLHF diagram highlighting components discussed in this section.

In this section, we delve deeper into the process of reward model learning, which we briefly touched on in Section 2.3. We will discuss various aspects associated with this topic, namely the various human feedback models, utility (i.e., reward) learning per se, different reward model inputs, and how to increase feedback efficiency.

### 5.1 Human Feedback Model

The basic premise underlying the majority of approaches in RLHF is that human feedback is directly related to the reward function to be learned. To this end, the human feedback must first be captured in a sound mathematical framework that establishes the connection to the reward function. On a high level, one can

break down almost all feedback types in Section 3.2 to a choice scenario: The human chooses one specific feedback option (label) from an available pool of possible feedback options (choice sets), which may be infinite[3]. The query determines the explicit contents of the choice set. For example, if the query is to compare two trajectories, then the choice set consists of all possible outcomes for these two trajectories.

Assuming that human choices are not always optimal, one obtains a fruitful mathematical framework when focusing on the probability

$$\mathbb{P}\left(c \text{ is chosen} \,|\, \mathcal{C}\right), \tag{4}$$

where $\mathcal{C}$ is the set of possible choices and $c \in \mathcal{C}$ one explicit choice. For the RLHF scenario, where the agent asks queries $q_i$ and the human gives labels $l_i$ as feedback (see Section 2.4), the choice set is specified by a function of the query. Formally, $\mathcal{C} = m(q)$ for some mapping $m$ that maps a query $q$ to the set of all possible candidate labels extractable from $q$ for the specific feedback type. For example, if the query is to rank a finite number of trajectories, then the choice set is the set of all possible rankings that can occur for the trajectories involved.

With this view, we can therefore place (4) in the RLHF context and write

$$\mathbb{P}\left(\text{label } l \text{ is provided} \,|\, m(q)\right) \tag{5}$$

for the probability that a human labeler returns a label $l$ from all possible candidate labels that can be extracted from a given query $q$. We explain next how this probability can be modeled and discuss various related modeling questions (e.g., human rationality, multiple humans, or Markov assumption). One could also recover the noiseless scenario if the latter probability distribution is degenerated for all possible candidate label sets.

### 5.1.1 Boltzmann Distribution

Human choice models as in (4) have been studied for a long time in various scientific fields such as psychology (Thurstone, 1927), economics (Train, 2009), or behavioral science (Cattelan, 2012). Accordingly, there are many different choice models to resort to for (5), which, in some cases, are the same models, just under different names. A popular class of such human choice models assumes every choice option $c$ to be equipped with a (latent) utility $u_c$, which the human perceives in a perturbed way. This perturbation is modeled by means of perturbation random variables $\epsilon_c$ that perturb the utility in an additive way, so that (4) becomes

$$\mathbb{P}\left(c \text{ is chosen} \,|\, \mathcal{C}\right) = \mathbb{P}\left(c = \underset{c \in \mathcal{C}}{\operatorname{argmax}} \; u_c + \epsilon_c\right). \tag{6}$$

The translation for the RLHF setting for (5) is then accordingly

$$\mathbb{P}\left(\text{label } l \text{ is provided} \,|\, m(q)\right) = \mathbb{P}\left(l = \underset{l \in m(q)}{\operatorname{argmax}} \; u_l + \epsilon_l\right), \tag{7}$$

and we shall now stick to the RLHF translation from now on. These probabilities depend on the specific distributional assumptions that are made on the perturbation variables that only for specific cases lead to a closed-form of the right-hand sides of the latter equations. When stipulating a standard Gumbel distribution for the perturbations, one always obtains a closed form that is proportional to the exponential utility of the provided label:

$$\mathbb{P}\left(\text{label } l \text{ is provided} \,|\, m(q)\right) \propto \exp(u_l). \tag{8}$$

This is known as the *Boltzmann distribution* that also appears in a perhaps slightly modified version in various different subfields of machine learning (ML) and statistics. When restricting to discrete (choice) sets for $m(q)$, this distribution is also known as the multinomial logit model (Train, 2009) or Gibbs measure (Georgii, 2011), and as the Bradley-Terry model (Bradley & Terry, 1952) when the choice sets consist of pairs. All of these

---

[3]This point of view goes back to the work of Jeon et al. (2020).

also have a link to the Plackett-Luce model (Luce, 1959; Plackett, 1975), which is a probability distribution on the space of total orders or rankings (see Alvo & Yu (2014) for details).

This model is often used for various reasons. A particularly compelling reason is the closed analytic form, which in turn makes it possible to obtain a closed form for the gradient with respect to the utilities. Another reason is that this model satisfies Luce's axiom of choice (Plackett, 1975), which requires the probability of choosing an option from a pool of choice options not being affected by the presence or absence of other options in the pool. In this way, coherent decision-making is ensured, which, however, might be challenged as humans likely do not make fully rational decisions (see Section 4.2.1). Jeon et al. (2020) show that the usage of the Boltzmann distribution is justified by the principle of maximum entropy. More precisely, they show that it is the maximum entropy distribution over choices for a so-called satisficing human decision maker, i.e., one who is making a choice with an optimal reward in expectation up to some slack $\epsilon > 0$.

To build the bridge between reward learning and the modeling of human feedback, the Boltzmann distribution can be used by assuming that the utilities can be represented as a function of the reward function, usually featuring the return of a trajectory. More specifically, one assumes a *grounding function $G$* that maps choice options (or labels) to the set of distributions over trajectories and sets the utility of a label $l$ as

$$u_l := \mathbb{E}_{\tau \sim G(l)}[R(\tau)]. \tag{9}$$

Note that $u_l$ depends on the return $R$, so that we also may use $u_l(R)$ to emphasize this dependency. For the common case of pairwise trajectory comparisons, where for two trajectories $\tau_1, \tau_2$ we obtain for the possible labels $l \in \{\tau_1 \succ \tau_2, \tau_1 \prec \tau_2\}$ the respective utility by using the projection onto the preferred trajectory as the grounding function $G$. Accordingly, the utility of the label represents essentially the utility of the preferred trajectory of that label, i.e., $\tau_1$ or $\tau_2$ in this case. As another example consider the case of e-stops feedback (see Section 3.2.2). Here, the possible labels $l$ provided by the user are $\texttt{STOP}_t$ and $\texttt{CONT}$ encoding the stopping at time $t$ or the continuation of a trajectory. For the grounding function, one can define

$$G(l) = \begin{cases} \tau, & l = \texttt{CONT}, \\ \tau_{0:t}\tau_t \ldots \tau_t, & l = \texttt{STOP}_t, \end{cases}$$

where $\tau_{0:t}$ is the trajectory of $\tau$ trimmed to the stopping time $t$, and $\tau_t$ is the action-state pair at time $t$. Table 1 in Jeon et al. (2020) provides an overview of the different grounding functions that lead to a specific feedback type. It is worth noting that one can also easily find a grounding function for the feedback type of a (partial) order over trajectories as considered, for instance, by Myers et al. (2021). Moreover, one can generalize this modeling approach by using (partial) segments instead of trajectories.

Although this general human feedback model has been much in use and shown to be useful for the sake of human alignment, it is not without its critics (see Lindner & El-Assady (2022) or Section 3.2.1 in Casper et al. (2023)). This has led to different adaptions of the general model based on the Boltzmann distribution that will be discussed in the following. Moreover, we will also concisely review other human feedback models that have been in use besides the Boltzmann distribution, discuss relevant work on the consequences or robustness of human feedback model misspecification, and highlight contributions on varying the standard assumptions on the nature of the human feedback.

### 5.1.2 Human-Specific Rationality Coefficient

The Boltzmann distribution in (8) can be extended by a rationality coefficient $\beta \in [0, \infty)$ that reflects the precision of the human labeler:

$$\mathbb{P}\left(\text{label } l \text{ is provided} \mid m(q)\right) = \mathbb{P}\left(l = \underset{l \in m(q)}{\text{argmax}} \ \beta\, u_l + \epsilon_l\right) \propto \exp(\beta \cdot u_l). \tag{10}$$

The higher $\beta$, the more (10) resembles a pointmass distribution modeling a highly rational human labeler that is always able to identify the option with highest utility. The lower $\beta$, the more it resembles a uniform distribution modeling a highly irrational human labeler acting purely at random. Without this extension,

the commonly considered Boltzmann distribution (or Bradley-Terry model in the common case of pairwise comparisons) in (8) assumes a rationality coefficient of 1. Ghosal et al. (2023) show in their experiments that the estimation of this coefficient can indeed positively influence reward learning. However, estimation requires a calibration reward function, as the rationality coefficient is otherwise not identifiable (Bengs & Hüllermeier, 2020). Similar findings are shown by Daniels-Koch & Freedman (2022), who model the rationality coefficient as a query-dependent function that might differ for the human labelers (see Section 5.1.6).

Another alternative to the rationality coefficient for representing irrational humans is achieved by introducing a query-independent error probability (Christiano et al., 2017). To be more precise, it is assumed that the human labeler only adheres to the Boltzmann distribution in (8) in 90% of cases and otherwise makes completely random decisions. This formulation is similar to Huber's contaminated model (Mu & Xiong, 2023).

### 5.1.3 Alternative Utility Notions

Knox et al. (2024) show that the Boltzmann model does not generally lead to an identifiable reward function using (9) by presenting three concrete scenarios for which identification is not possible. The root cause of the non-identifiability is the usage of a trajectory's return as the utility in (9). They therefore suggest using a trajectory's regret as an alternative, which provably leads to identifiable rewards.

A trajectory's regret is the negated sum of the optimal policy's advantage over each state-action pair in the trajectory. Empirically, it has been shown that this modification improves the alignment of the learned strategy with human preferences. The downside of this alternative is that regret depends on the unknown optimal policy. Recently, it has also been suggested to consider $Q$-values of a human policy as the utilities (Myers et al., 2023), while Holladay et al. (2016) used differences of cost functions that depend on the available choice set and the human's uncertainty.

### 5.1.4 Human Feedback Models Beyond Boltzmann

While the human feedback model based on the Boltzmann distribution is the most widely used model nowadays, other models have also been considered in the literature. In particular, for the probability in (4) other models such as the Thurstone model[4] (Wilson et al., 2012; Kupcsik et al., 2018; Bıyık et al., 2020), the ridge-noise model (Schoenauer et al., 2014), the binary model (Sugiyama et al., 2012) or mixed forms thereof (Wirth et al., 2016) have been considered. Of these models, only the Thurstone model (Thurstone, 1927) has a similar interpretation as the Boltzmann distribution based on perturbed utilities, only differing in the distribution of the perturbance random variables.

**Link functions** Another possibility, which is particularly popular in theoretical work on RLHF (see Section 7), is the use of other functions on the right-hand sides of Eq. (8) than the exponential function. The concept is primarily used for pairwise comparisons of trajectories. It essentially states that the probability of the result of a pairwise comparison between two trajectories is the difference of their utility values under a so-called *link function*. More specifically, let $q = \{\tau_1, \tau_2\}$ be the query to compare the trajectories $\tau_1$ and $\tau_2$, then, assuming a link function $\Phi : \mathbb{R} \to [0, 1]$, one models the probability in (5) for $l$ representing a preference for $\tau_1$ as

$$\mathbb{P}\left(\text{label } l \text{ is provided} \mid m(q)\right) = \mathbb{P}\left(\tau_1 \succ \tau_2 \mid m(\{\tau_1, \tau_2\})\right) = \Phi(u_{\tau_1} - u_{\tau_2}). \tag{11}$$

For $l$ representing a preference for $\tau_2$, one proceeds similarly. The minimal assumptions on the link functions are that

(i) it is (strictly) monotonically increasing to take into account that trajectories with higher utilities will also have a higher chance to be picked;

(ii) $\Phi(x) = 1 - \Phi(-x)$ to ensure that $\mathbb{P}\left(\tau_1 \succ \tau_2 \mid m(\{\tau_1, \tau_2\})\right) = 1 - \mathbb{P}\left(\tau_1 \prec \tau_2 \mid m(\{\tau_1, \tau_2\})\right)$.

---

[4]The probit model, as used in Bıyık et al. (2020), corresponds to Thurstone's Case V.

Note that the second property implies $\Phi(0) = 1/2$ so that trajectories with the same utility also have the same chance of being selected. Any cumulative distribution function of a symmetric continuous random variable fulfills these two conditions. The two most common link functions that both fulfill the conditions are the linear link function given by

$$\Phi(x) = \max\left\{0, \min\left\{1, 1/2 \cdot (1 + x)\right\}\right\}$$

and the logistic link function given by

$$\Phi(x) = \frac{1}{1 + \exp(-x)}\,.$$

Both are cumulative distribution functions: The linear link function is the cumulative distribution function of a continuous uniform distribution on $[0, 1]$. In contrast, the logistic link function is the cumulative distribution function of a logistic distribution with location parameter 0 and scale parameter 1. Moreover, both are intensively studied in theoretical approaches (see Section 7.1), and the latter leads to (8) (when restricted on pairwise comparisons) and is a special case of the softmax function.

**Two-Staged Choice Model**  Bobu et al. (2020b) propose the Limiting Errors due to Similar Selections (LESS) model that is inspired by the attribute rule model suggested by Gul et al. (2014). It assumes a feature map for trajectories and a (similarity) function mapping trajectory features and trajectories to integers and uses a two-stage process for modeling the human feedback (or choice): First, choosing a trajectory feature according to the Boltzmann distribution and then a trajectory with the (logarithmic) similarity functions as the utilities within the Boltzmann distribution. Their experiments show that this model can capture human feedback more appropriately than the standard Boltzmann distribution.

**Generative Model**  Abramson et al. (2022) evaluate the usage of a generative model for learning from human preferences. More specifically, instead of assuming some underlying utility as in the Bradley-Terry model, they attempt to train a model to generate the human feedback (inter-temporal preferences in this case, see Section 3.2.1) and directly interpret this feedback as reward. However, they found that this empirically does not perform as well as the inter-temporal Bradley-Terry model.

### 5.1.5  Misspecification

The human feedback model may be misspecified in various ways. Milli & Dragan (2020) investigate the problem of misspecifying the nature of human feedback that can be either literal or pedagogical. The former means that the human gives targeted feedback for solving the actual RL problem, while the latter means that the human gives targeted feedback that is deemed helpful for the learner. They show theoretically and empirically that the case of a learner assuming a pedagogical feedback with an actual literal human always performs worse than the reversed case, i.e., a learner assuming a literal feedback with an actual pedagogical human.

A related question is studied by Freedman et al. (2021), namely, what if the learner makes incorrect assumptions about the choices from which the human selects its feedback? They consider different types of such choice set misspecification and show that depending on the type of misspecification, the performances might vary drastically, even leading to no losses at all in some specific cases.

In the field of inverse RL, the general question of the robustness of reward learning in terms of a misspecified human feedback model is theoretically investigated by Skalse & Abate (2023). It turns out that the optimality model is not robust with respect to any misspecification, the Boltzmann model is robust for quite a range of specific misspecifications, and the degree of robustness of the maximal causal entropy model lies between the latter two. Even though these results are primarily derived for inverse RL, they also have similar immediate implications for RLHF.

### 5.1.6  Diverse Preferences

One potential issue with the RLHF framework is that it does not specify whose preferences to align to. It is common to request feedback from multiple labelers in a crowd-sourcing setting, in which case the different

labelers may disagree. There are two main ways to deal with this challenge: Either trying to learn each labeler's preference separately, or trying to learn a model of the group's mean preference.

Bakker et al. (2022) investigate the first option by proposing to learn multiple reward functions, which can then be aggregated in arbitrary manners and even be used to find consensus among people with different preferences. The second is more commonly used, however: Xue et al. (2024) learn a single reward function from multiple humans who may give diverse and inconsistent feedback, aiming to stabilize learning in spite of these inconsistencies using regularization, a consistency constraint, and ensembling. Similarly, Chhan et al. (2024) try to estimate the correct preference for pairwise trajectories directly by combining the users' expressed preference labels instead of assuming individual reward functions. As a middle-ground between the single reward function and multiple ones, Myers et al. (2021) propose to learn a multimodal reward function that captures multiple people's preferences and use a mixture model of Plackett-Luce models to represent the feedback more accurately.

With a stronger focus on the active retrieval of human feedback, Freedman et al. (2023) model the problem of selecting a suitable human labeler as a variant of the multi-armed bandit problem (Lattimore & Szepesvári, 2020), which they call hidden-utility bandit. In this variant, the agent has in each decision round the choice between two options: (i) drawing a bandit arm, then receiving a hidden arm-dependent utility, and finally observing an item, or (ii) querying a human to observe a preference between two sampled items and incurring a human-specific query cost. The feedback mechanism of all human teachers is modeled via a same Boltzmann distribution, differing only in their known individual rationality coefficients. The same modeling of human feedback is also considered by Barnett et al. (2023), who, however, use a Bayesian approach to determine which person should be queried in order to obtain the most informative feedback in expectation. Daniels-Koch & Freedman (2022) investigate the rationality coefficient already considered in the previously mentioned work and model it as a query-dependent function that might differ for the human labelers.

When dealing with diverse preferences, aggregating them via the mean may not always be suitable. Instead, after learning the individual preferences, one needs to address the important question of preference aggregation, which is studied in social choice theory (Brandt et al., 2016). Thus, a recent rapidly-developing research trend has started to investigate the tools developed in social choice in AI alignment (Conitzer et al., 2024; Sorensen et al., 2024; Mishra, 2023) in general and in RLHF (Dai & Fleisig, 2024; Zhong et al., 2024a; Park et al., 2024; Chakraborty et al., 2024b; Swamy et al., 2024; Lambert et al., 2023) in particular.

### 5.1.7 Relaxation of the Markov Assumption

Most works assume that the human feedback is given based on a latent Markovian reward model, i.e., the return of a trajectory $\tau$ decomposes into a sum of independent rewards over state-action pairs (see (1)). Early et al. (2022) relax this assumption by dropping the need for the Markov property, such that the instantaneous reward might depend on hidden states. Similarly, Kim et al. (2023) avoid the Markov assumption by using a transformer as the preference model. A similar effect may be achieved by learning a state representation with a recurrent network in which the rewards are Markov, similar to the approach taken by Hafner et al. (2023), but we are not aware of any work exploring this. Abramson et al. (2022) work in a non-Markovian setting as well by using memory-augmented networks for both the policy and the reward model.

## 5.2 Utility Learning

After choosing a human model to relate feedback to utilities, we can use the observed feedback to recover the latent utilities. This utility learning can be reduced to a standard supervised learning problem and, therefore, is commonly solved with the techniques of empirical risk minimization or Bayesian approaches, both of which will be discussed in the following.

### 5.2.1 Empirical Risk Minimization

The most prevalent variant for learning the reward function, already presented in Section 2.3, is a special case of empirical risk minimization. The general approach of empirical risk minimization for reward function

learning, assuming an underlying human feedback model with utilities as in (9) is to find the minimizer of

$$\mathcal{L}(R; \mathcal{D}) = \sum_{i=1}^{N} \ell\left(u_{\cdot}(R), l_i, m(q_i)\right),\tag{12}$$

where $\mathcal{D} = \{(l_i, q_i)\}_{i=1}^{N}$ is the given data set of observed label and query pairs, $\ell : \mathbb{R} \times \mathcal{Q} \times \mathcal{C}$ is a suitable loss function with $\mathcal{Q}$ being the set of all possible label sets, and $u_{\cdot}(R)$ denoting the utility (depending on the return $R$) of the possible labels for the given label-query pair $l_i, m(q_i)$. As an illustration, consider the common case of pairwise trajectory comparisons where queries are pairs of trajectories $q_i = \{\tau_1^i, \tau_2^i\}$, and labels $l_i \in \{\tau_1^i \succ \tau_2^i, \tau_1^i \prec \tau_2^i\} = m(q_i)$ are the human's preference over the two trajectories. For a given query $q_i$, we then obtain (2) as a special case of (12) by using the loss function:

$$
\begin{aligned}
\ell(u_{\cdot}(R), l_i, m(q_i)) &= -\log\left(\frac{1}{1 + \exp(u_{m(q_i)\backslash l_i}(R) - u_{l_i}(R))}\right) \\
&= -\log\left(\frac{1}{1 + \exp(\mathbb{E}_{\tau \sim G(m(q_i)\backslash l_i)}[R(\tau)] - \mathbb{E}_{\tau \sim G(l_i)}[R(\tau)])}\right),
\end{aligned}
$$

where in the case of pairwise comparison, $u_{\cdot}(R) = (u_{l_i}(R), u_{m(q_i)\backslash l_i}(R))$ and the grounding function $G$ is the projection onto the preferred trajectory. This is the negative log-likelihood for the Boltzmann distribution for the observational pair $(l_i, m(q_i))$.

For the entire learning process, a model class $\mathcal{R}$ is assumed for the reward function $R$. This model class is usually a parameterized class of functions, such as, for example, the class of linear reward functions (Katz et al., 2021)

$$\mathcal{R} = \{R_{\boldsymbol{\psi}}(s, a) = \boldsymbol{\psi}^{\top}\boldsymbol{\phi}(s, a) \,|\, (s, a) \in \mathcal{S} \times \mathcal{A}, \boldsymbol{\psi} \in \mathbb{R}^d\},$$

where $\boldsymbol{\phi} : \mathcal{S} \times \mathcal{A} \to \mathbb{R}^d$ is some state-action feature mapping. This entails that good features are known in advance such that rewards can be expressed as a linear combination of those features. Using such linear models may lead to reward model misspecification. Studying this setting, Bobu et al. (2020a) propose to adapt the hyperparameter $\beta$ in (3) to account for this issue.

Since the assumption of a linear reward model may be too strong in practice, most recent work is based on non-linear models, especially using differentiable models, but other cases have been investigated as well. In the latter case, for instance, decision trees have been considered to learn an interpretable reward model (see Section 5.2.4). In the former case, simple multilayer perceptron (MLP) has naturally been considered, but more recent deep learning architectures are more commonly used in the recent literature. Thus, especially with partially observable domains, the reward network may be composed of a state-action encoder followed by fully connected layers. For instance, Abramson et al. (2022) combine ResNet blocks for image processing, a learnable embedding table, a multi-modal transformer, LSTMs, and MLPs. Besides, Kim et al. (2023) use a Transformer-based architecture (Vaswani et al., 2017), motivated by the observation that rewards are often non-Markovian.

In addition to the usual empirical risk in (12), it is also typical, as in supervised ML, to add a regularization function to prevent overfitting:

$$\mathcal{L}(R_{\boldsymbol{\psi}}; \mathcal{D}) = \sum_{i=1}^{N} \ell(u_{l_i}(R_{\boldsymbol{\psi}}), l_i, m(q_i)) + \lambda_r(\boldsymbol{\psi}),\tag{13}$$

where $\lambda_r : \Psi \to \mathbb{R}_+$ is a regularization function defined on the parameter space $\Psi$ of the underlying reward model class. For instance, Christiano et al. (2017) simply use L2 regularization and also consider dropout in some domains. Recently, Verma & Metcalf (2024) propose to define a more complex regularization term, which consists in biasing the learned rewards to be proportional to an approximate state importance provided by a trained Transformer-based forward model.

The main supervised loss to train the reward model can also be augmented with additional losses corresponding to auxiliary tasks to avoid overfitting and improve generalizability. For instance, Abramson et al.

(2022) use a behavior cloning loss and add a policy head to the reward network, thereby preventing that the reward model drifts too far from its initialization from the policy. Metcalf et al. (2023) design a reward model using state-action representations trained to be temporally consistent via self-supervised learning. On a related note, the scalar preference optimization problem has been extended to a multidimensional one by Zhong et al. (2024b) to represent diverse human preferences and Marta et al. (2023) for query efficiency.

### 5.2.2 Bayesian Approach

As is generally the case in supervised ML, there is also the variant of using Bayesian modeling for learning a target object instead of the (empirical) minimization of a loss function. To this end, one starts with a prior distribution $\rho$ over the parameter space of the reward function that is updated in light of the data set $\mathcal{D}$ by means of Bayes theorem:

$$\mathbb{P}(\boldsymbol{\psi} \,|\, \mathcal{D}) \propto L_{\boldsymbol{\psi}}(\mathcal{D}) \cdot \rho(\boldsymbol{\psi}) \,,$$

where $L_{\boldsymbol{\psi}}(\mathcal{D}) = \prod_{i=1}^{N} \mathbb{P}\left(l_i \text{ is provided} \,|\, m(q_i), \boldsymbol{\psi}\right)$ is the likelihood of the data under the assumed human feedback model with reward function $R_{\boldsymbol{\psi}}$. Such an approach is used for pairwise trajectory comparisons, for instance, by Schoenauer et al. (2014) for the noisy-ridge model or by Sadigh et al. (2017) for the Boltzmann distribution as the human feedback model. In inverse RL, such Bayesian approaches have been considered as well (see Section 4.3 in Arora & Doshi (2021)).

Instead of assuming that the reward functions are parameterized, one can use the reward functions directly as a parameter class and use a prior distribution over them. This could, for example, be a Gaussian process as initially considered by Kupcsik et al. (2018) for pairwise trajectory comparisons and adapted in later works (Bıyık et al., 2020; Cosner et al., 2022). Here, again, it is worth mentioning that such considerations have also been made in inverse RL before (see Section 4.3 in Arora & Doshi (2021)).

### 5.2.3 Partial Identifiability

A crucial question when it comes to learning the reward function is whether the reward function can be identified at all. If two reward functions induce exactly the same human feedback model, the reward function is called partially identifiable or ambiguous. Skalse et al. (2023) study this topic for the Boltzmann distribution as the underlying human feedback model when demonstrations (inverse RL) or pairwise trajectory preferences are given as feedback. For demonstrations, this question has also been examined in other works (Ng & Russell, 2000; Dvijotham & Todorov, 2010; Kim et al., 2021; Cao et al., 2021a). On a related note, Ellis et al. (2024) tackle this identifiability issue by considering suitable acquisition functions (see Section 4.1.1).

### 5.2.4 Interpretability

The field of explainable artificial intelligence (XAI) has emerged in recent years to improve the transparency and explainability of models or even to enable them in the first place. Roughly speaking, the aim is to resort to more interpretable methods or provide explanations for both experts and non-experts, shedding light on why a certain input in a (black box) model leads to a certain result. Explanations can take different forms, as can the ways to ensure the transparency of models, and for a detailed overview, we refer to Barredo Arrieta et al. (2020). It is worth noting that the field has grown so extensively over the years that even dedicated overviews for the field of interpretable and explainable RL are by now available (Puiutta & Veith, 2020; Qing et al., 2023; Milani et al., 2023; Glanois et al., 2024).

For the branch of RLHF, the existing works are quite sparse and mostly limited to using tree models as transparent and explainable models for learning the reward function (Bewley & Lécué, 2022; Bewley et al., 2022; Kalra & Brown, 2022; Bewley et al., 2024; Kalra & Brown, 2024). Another way to realize explainability within RLHF suggested by Zhang & Kashima (2023) is to learn simultaneously the reward function and the importance of states using a weight network. Assuming that for (long) trajectories, only a few states are important for the preference outcome, their framework can be used to select samples for explainability purposes. Moreover, a perturbation analysis is suggested to evaluate explanations in a quantitative manner using the learned state importance weights.

### 5.2.5 Online Improvements

Christiano et al. (2017) demonstrate that it is important to improve the reward model online, a finding that has been confirmed by subsequent works such as the one by Gao et al. (2023), which empirically demonstrates that overoptimization of a reward model trained offline leads to performance degradation. Without online improvements, issues of overoptimization of an imperfect reward model may occur. Abramson et al. (2022) give an example of this: They attempt to fine-tune an agent initialized with behavioral cloning with an engineered reward function and find that it fails to generalize and actually worsens the performance. They also compare RLHF with a reward model trained offline with iterative improvement and find that iterative improvement leads to better performance, even sometimes exceeding human performance.

This is related to issues posed by the approximate nature of the reward model in general, discussed in further detail in Section 6.1, but improving reward model accuracy, in general, is not sufficient: McKinney et al. (2022) further show the interdependence of the reward model and the policy, demonstrating that reward models trained online together with a policy may not be effective when a completely new policy is trained.

Solutions to the problems of overoptimization and interdependence can take different forms: One is to update the reward model online with sufficient frequency using notably more on-policy queries (see Section 4.1.1), another is to improve the reward model, e.g., by leveraging ensembles (Eisenstein et al., 2024; Coste et al., 2024), combining multiple reward models with weight averaging(Rame et al., 2024), or by modifying the training procedure to place additional emphasis on challenging examples (Zheng et al., 2024), and a third, discussed in Section 6.1, is to add constraints to the policy training.

### 5.2.6 Learning from Multiple Feedback Types

As discussed in Section 3.5, it is often desirable to combine several feedback types. This requires extensions of the learning process to incorporate different sources of feedback. Learning from multiple feedback types can be achieved by pre-processing the feedback, assuming common latent factors, or by using the feedback types for distinct purposes.

The first approach is demonstrated by (Novoseller et al., 2023), who infer preferences from demonstrations, allowing them to treat both types of feedback equally in the learning pipeline. In the style of the second approach, Jeon et al. (2020) proposes the unified framework of reward-rational choice, which allows for interpreting many forms of human feedback as Boltzmann-rational choices and, through this common framework, enables combination and adaptive selection of feedback types. Finally, different types of feedback can be used for entirely different purposes, such as one for objective learning and another for safety-constraints (Cosner et al., 2022) or for representation learning (see Section 3.2.3).

Since multiple sources of reward information may conflict, it is important to consider how to combine them. Krasheninnikov et al. (2021) study several possible strategies of combining several reward functions in this setting, relating it to multi-task inverse RL. Note that this challenge of conflicting sources of reward relates tightly to challenges posed when receiving diverse preferences from different labelers, as discussed in Section 5.1.6.

### 5.2.7 Offline Reward Learning

There is a recent trend towards offline RLHF, where both the reward model and the policy are trained offline. The offline setting is also frequently considered in RLHF theory (Section 7). Early approaches in this area (Kim et al., 2023; Shin et al., 2023) first generate queries from an offline dataset of behaviors, gather human responses, train a reward model from the resulting preference data, and then leverage offline RL algorithms to derive a policy. We do not cover these works in detail, since this survey primarily focuses on the interactive and online setting (see Section 1.3). Nonetheless, the offline setting is particularly useful for evaluating novel approaches, e.g., for active query selection, using offline datasets. We refer the interested readers to Section 8.4 for a discussion of available datasets.

Note that offline reward learning does not necessarily need to be combined with offline policy learning, as the reward model can be used to train a policy online as well. This can be problematic in practice, as

the distribution of experiences will generally shift during online policy learning, which can lead to poor performance of the reward model. Nonetheless, this setup is commonly used in the LLM finetuning setting, since distributional shift is less of a concern due to the large and diverse training data (often collected off-policy) and the bandit-like nature of the task (compare Section 6.2). In this setting, it is also common to directly integrate preference and policy learning through direct policy optimization (DPO) techniques further discussed in Section 6.3.

## 5.3 Evaluating Learned Reward Functions

A central question when it comes to learning the reward function is how to evaluate the learned reward function and how reward functions can be compared with each other. For this purpose, different approaches are available, e.g.:

**Rollout Method** In inverse RL, a common method for evaluation is the rollout method (Ng et al., 1999; Fu et al., 2018a; Ibarz et al., 2018; Brown et al., 2019). In this approach, one first learns an optimal policy for the learned reward function and then estimates the value of this policy for online trajectory rollouts using the known ground-truth reward function. This approach can be transferred to RLHF as well. In many cases, however, especially in safety-critical areas such as medicine or autonomous driving, such online rollouts cannot be executed.

**Off-policy Evaluations** When online rollouts are not possible, so-called off-policy evaluations, which estimate the value of the optimal policy based on an available data set, may be considered. For coping with biases or large variances due to policy mismatch, approaches using importance sampling (Precup et al., 2000), regression- or classification-based methods (Paduraru, 2013; Le et al., 2019; Irpan et al., 2019), or combinations of these (Jiang & Li, 2016) have been proposed. Off-policy evaluation is particularly well-suited for evaluating LLMs reward models. Lambert et al. (2024) propose a particularly large-scale dataset for this purpose, focusing on examples that are difficult to judge. They then use accuracy-based metrics across multiple categories to evaluate reward models. The problem with these approaches, however, is that the traces of the explicit sources of error through policy learning or reward learning are blurred, and that some require access to the ground-truth rewards.

**Distance Functions** Yet another alternative, which has been advanced in the seminal paper by Gleave et al. (2022a), is using a suitable distance function for reward functions. Suitable here means that two reward functions, which differ only by certain transformations such as potential shaping (Ng & Russell, 2000) or positive scaling, should have zero distance if these transformations do not change the policy ranking with regard to the expected return. For this purpose, Gleave et al. (2022a) present a pseudometric, called Equivalent-Policy Invariant Comparison (EPIC) distance, that is determined in three steps: First, mapping two reward functions to a so-called *canonicalization form* that is invariant to transformations of the latter kind. Second, normalizing these canonicalization forms by means of a specific weighted $L_2$-norm whose weights are determined by a distribution over the transitions. Finally, the EPIC distance is the weighted $L_2$-norm distance of the normalized canonicalization forms.

Even if some attractive properties, above all a Lipschitz continuity in terms of the EPIC distance of two reward functions for the difference of the value functions of the induced optimal policies is shown, this distance has its shortcomings. One shortcoming is that the canonicalization form used by EPIC distance does not encode sufficient knowledge about the underlying transition dynamics, which might lead to unreliable distances when evaluating reward functions on physically non-realizable transitions. To this end, Wulfe et al. (2022) propose the Dynamics-Aware Reward Distance (DARD), which uses a slightly different form of canonicalization but restricts the evaluation of the reward functions to transitions that are approximately physically feasible.

Recently, EPIC-like distances (Jenner et al., 2022) and STAndardised Reward Comparison (STARC) metrics (Skalse et al., 2024), which are entire classes of pseudometrics on the space of all reward functions were proposed that generalize the three-step approach underlying the EPIC distance (and

DARD) by parameterizing each of the steps. Specifically, the canonicalization function in the first step, the normalization in the second, and the metric in the third step are kept variable. If these three functional parameters fulfill certain requirements, then the resulting distance has some appealing properties, e.g., being a pseudometric that is zero if and only if the two reward functions induce the same ordering of policies or imply upper and lower bounds on value function differences. In particular, these metrics retain the flexibility of DARD (in terms of specifying transition dynamics), while at the same time preserving the theoretical justification of EPIC.

**Visual and Human Inspection** For an evaluation by visual inspection, Jenner & Gleave (2021) propose a method for preprocessing reward functions by transforming them into simpler but equivalent reward functions for better interpretability. Related to this and the rollout method, the quality of the reward function learned can also be evaluated by a human (or expert) by examining the behavior of the agent on the target task. This can even extended to the reinforcement learning from AI feedback (RLAIF) setting, by using another trained model as the evaluator (Li et al., 2024). Besides, in the context of LLMs, datasets have been proposed and specifically designed to evaluate the (in-)consistency of learned reward models with respect to semantic changes of prompts (Shen et al., 2024).

## 5.4 Reward Model Inputs

Besides the feedback type, another factor is the modality of the reward model input data. This usually consists of the agent's observations and actions. Observations can range from true state to high-dimensional inputs (e.g., images), while actions can range from discrete finite actions to continuous actions.

For instance, many typical RL benchmarks are in the continuous control domain (e.g., MuJoCo simulated robotics tasks) with true state representations and simple discrete actions. In such problems, Christiano et al. (2017) train reward models from these inputs.

When no compact state representation is available, raw images are often used in control tasks, which makes the learning of rewards more challenging since the setting becomes partially observable and the reward function is generally not Markov with respect to the observations. In such cases, the conventional trick of approximating a true state with a sequence of frames is often employed. This approach is used, for instance, by Christiano et al. (2017) to train reward models on the Atari benchmark suite. When taking only observations as inputs, one can resort to recurrent models (Abramson et al., 2022) or Transformer-based models (Kim et al., 2023).

More recently, many applications of RLHF are in the natural language processing (NLP) domain. In these settings, the policy takes natural language as both input and output while the reward model takes it as input (see, e.g., the work by Ouyang et al. (2022)). Naturally, more complex scenarios (e.g., with both language and vision inputs (Abramson et al., 2022)) have also been studied.

## 5.5 Increasing Feedback Efficiency

Maximizing feedback efficiency is vital in RLHF due to the high cost of human feedback. This section delves into methods that enhance learning from limited human feedback. We discuss methods that leverage prior offline data, methods that use (partially unlabeled) data more efficiently, and methods that aim to gather more informative data.

### 5.5.1 Using Prior Data

There are often large amounts of prior data available at little or no additional cost. While this data generally was generated for other tasks, many basic human preferences are the same for various tasks and can often even be extracted from completely unsupervised data such as text corpora. By leveraging this prior data, we can greatly reduce the amount of feedback necessary to learn the current task's objective. We explore various methods, including meta- and transfer learning, leveraging foundation models, reward model initialization, preference model pretraining, and supervised representation learning.

**Meta- and Transfer Learning**   Meta- and transfer learning techniques in reward model training exploit the commonalities across different objectives, facilitating quick adaptation to new tasks. Ren et al. (2022) develop a broadly applicable meta-reward model, pre-trained on a diverse set of tasks to capture a wide range of preference patterns, enabling efficient adaptation to new tasks with fewer examples. Xie et al. (2018) use a similar meta-learning approach to build a goal classifier across multiple visuomotor tasks. Closely related to these meta-learning approaches, Hejna & Sadigh (2022) integrate few-shot learning principles, optimizing their approach for scenarios where only a few examples are available for adapting to new tasks. In the domain of transfer learning, Liu et al. (2023a) explore zero-shot transfer of preferences, a method that enables adapting preferences without additional data from the new task. In a different vein, but closely related to meta- and transfer learning, Mendez et al. (2018) tackle the lifelong inverse RL problem, focusing on inferring reward functions for multiple tasks over time, which involves knowledge transfer between tasks. Collectively, these studies underscore the potential of meta- and transfer learning in enhancing the efficiency and applicability of reward models in RLHF.

**Leveraging Foundation Models**   Foundation models, i.e., large models trained on large amounts of often unlabeled data, can acquire significant knowledge about basic human preferences. A language model trained to predict the next token in a text corpus, for example, may learn to complete the sentence 'Frank was mad that his vacuum robot broke the vase', thereby learning that humans prefer non-destructive behavior. These learned preferences can then be leveraged in RLHF approaches. For instance, Kwon et al. (2023) propose to use LLM as a source of rewards. Du et al. (2023) is another example, where a success detector is trained using a pre-trained vision-language model (Flamingo). Their approach uses a dataset of trajectories with binary success labels, employing a non-interactive training method.

**Reward Model Initialization**   It is often beneficial to initialize the reward model with parameters from a model trained on a related task. This strategy is particularly common in language model fine-tuning, where self-supervised pretraining is a common practice. In such scenarios, it becomes logical to use these pre-trained models for initializing not just the policy but also the reward model. This methodology is adopted by Askell et al. (2021) and Ouyang et al. (2022). Specifically, Ouyang et al. (2022) use a pretrained language model for the reward model, opting for a smaller model relative to the policy to mitigate unstable learning. Notably, while they apply supervised fine-tuning to the policy before the RLHF phase, the reward model is initialized directly from the language model without any preliminary fine-tuning. This approach's applicability extends beyond language models to other areas. A notable example is Abramson et al. (2022), who, in the control domain, begin by training a policy using contrastive self-supervised learning and behavioral cloning. They then add an MLP head to the policy for the prediction of cumulative rewards.

**Reward Model Pretraining**   Reward model pretraining (Askell et al., 2021; Bai et al., 2022a) leverages prior offline data to pretrain the preference model before training it on policy samples. Askell et al. (2021) note that in the case of language models, noisy preference data can be readily obtained from sources such as rated Reddit comments, preferred Stack Overflow answers, and reverted Wikipedia edits. They leverage this as a pretraining step to increase data efficiency. This is in addition to regular language model pretraining, as discussed in the previous paragraph. A similar approach could be applied to control in case prior data and some means of inferring preferences, such as human corrections, are available. Even if no inferred preferences are available, Verma & Kambhampati (2023a) show that it can be beneficial to pre-train the preference model to predict close to constant reward on an initial set of trajectories. This avoids excessive fitting of the policy to random initialization differences in the reward function.

**Supervised Representation Learning**   A compact representation that captures all relevant information for human preferences while minimizing noise can greatly enhance preference learning efficiency. It may also generalize better than a representation learned end-to-end as part of the preference learning task, which may contain spurious correlations. Bobu et al. (2022) address this by proposing the learning of features through explicit human feedback using feature traces. Feature traces (see Section 3.2.3) involve human labelers explicitly teaching relevant features one by one by demonstrating behavior in which the feature monotonically increases or decreases. This method directly aligns the learned representation with human-identified features, enhancing preference learning efficiency but requiring detailed human input. However, feature traces require

labelers to be able to identify and articulate relevant features, which can be challenging. Bobu et al. (2023) offer an alternative approach with their Similarity-based Implicit Representation Learning (SIRL) method. SIRL learns representations from similarity queries (see Section 3.2.3), where human labelers provide feedback on whether behaviors are similar or different concerning the features that matter to them. This method captures a broader range of human notions of similarity without needing explicit feature knowledge, thus reducing the cognitive load on human labelers. In summary, while both approaches emphasize human feedback's centrality in representation learning, they differ in their methods of gathering this feedback. The feature traces used by Bobu et al. (2022) require specific feature knowledge, whereas SIRL used by Bobu et al. (2023) uses more intuitive similarity assessments, potentially offering a more user-friendly way to capture human preferences.

These diverse methods of using prior data demonstrate the potential for enhancing data efficiency in RLHF, enabling more effective learning from limited human feedback.

### 5.5.2   Using Data More Efficiently

Beyond the application of prior data, several techniques can enhance the efficiency of data utilization in training processes. This section will discuss a range of such methods, including self-supervised and semi-supervised training, as well as the integration of inductive biases and data augmentation strategies. These approaches are designed to make the most of the available human interactions and improve the final performance of RLHF models.

**Self-Supervised Auxiliary Tasks**   Self-supervised training enhances data efficiency in reward model training by using unannotated data to capture information about the task. This technique extends beyond the scope of pretraining methods, as discussed in the prior section, to incorporating concurrent auxiliary tasks to maximize the utility of available data. A prevalent technique, as applied by Abramson et al. (2022), Brown et al. (2020), and Metcalf et al. (2023), involves adding self-supervised losses to enhance representation learning for rewards. Abramson et al. (2022) implement a contrastive task where the reward network differentiates between observations that are consistent between multiple modalities and those that are not, blending this with preference learning loss and behavioral cloning. Brown et al. (2020) add multiple auxiliary tasks such as inverse and forward dynamics modeling, temporal distance prediction, and variational autoencoder training. Similarly, Metcalf et al. (2023) use the self-predictive representations technique (Schwarzer et al., 2021) to learn state representations that encode environmental dynamics, enabling a linear model to anticipate successor states, thereby forming an efficient basis for preference learning and significantly boosting sample efficiency. However, auxiliary losses for better representation learning are not the only approach to leverage self-supervised training. An alternate approach by Verma & Metcalf (2024) involves identifying important states using attention weights from a world model transformer and state importance estimates based on a preference predicting transformer. These estimates can aid credit assignment for observed preferences, further optimizing the training process.

**Semi-Supervised Training**   Semi-supervised training, blending labeled and unlabeled data, can leverage the unlabeled data to glean information about the task and the environment. This is most commonly done by generating pseudo-labels for the unlabeled data, either by leveraging model predictions or by making assumptions. The first approach is used by Cao et al. (2021b), using supervised learning methods to predict human preferences, and Zhan et al. (2021), employing a GAN-based approach to learn and mimic human preference patterns. Similarly, Park et al. (2022) expand their data set with high-confidence unlabeled samples based on the preference predictor's evaluations. The second strategy, making assumptions to augment data, is showcased by Zhou & Xu (2020). They generate preference data by assuming that (i) human-written examples are better than model-written examples, (ii) human-written and model-written examples are indistinguishable amongst themselves, and (iii) generations of later model iterations are better than those of earlier ones. This is closely related to RLAIF, where a pretrained language model is used to gather preference feedback, effectively leveraging the knowledge about human preferences learned by the pretrained language model through self-supervised learning.

**Data Augmentation** Data augmentation focuses on creating additional examples from existing labeled data. Temporal augmentation is particularly effective in RLHF, involving trajectory data. This is exemplified by Brown et al. (2019) and Park et al. (2022) who base their augmentation on the premise that preferences for complete trajectories can be extrapolated to cropped segments, allowing the generation of multiple derivative pairs from a single labeled trajectory pair. Park et al. (2022) additionally explore state modifications, such as random re-scaling and Gaussian noise addition, finding temporal cropping to be the most effective, with noise sometimes negatively impacting performance. In a similar vein, Verma & Kambhampati (2023b) focus on augmenting trajectories by concentrating on changing elements in observations and perturbing the other parts, based on the premise that movement indicates importance in image-based observations. Complementing these methods, Abramson et al. (2022) employ augmentation by randomly altering instructions and language responses, thus creating artificial examples of non-preferred behavior. These diverse data augmentation methods collectively enhance the training data set, contributing to the increased robustness and efficacy of RLHF models.

Relatedly, Meta-Reward-Net (Liu et al., 2022) optimizes not only for the preference prediction accuracy of the learned reward function but also of the learned $Q$ function in an actor-critic RL algorithm. This is beneficial since it avoids the phenomenon of confirmation bias, where one learned model (in this case the $Q$ function) overfits to targets predicted by another model (the reward model). It is not strictly a data augmentation technique, but closely related in practice.

### 5.5.3 Gathering Better Data

In addition to leveraging unlabeled data and using labels more efficiently, sample efficiency can be further increased by collecting more informative samples in the first place. This can either be achieved by selecting more informative samples from the experience buffer or by generating more informative experiences. While selecting informative samples from the experience buffer is addressed under active learning (see Section 4.1.1), this section focuses on generating more informative experiences.

While we are not aware of many works in this area, one possible approach involves steering the agent's exploration towards regions of the state space where human feedback would be most beneficial. Liang et al. (2022) implement this by employing intrinsic motivation, driven by the estimated uncertainty of the reward model, to guide the agent's exploration. This highlights the potential of not just using data more efficiently but also generating data in a more targeted manner.

## 6 Policy Learning

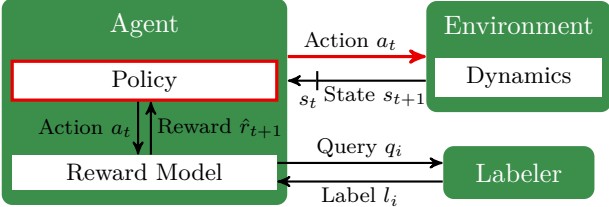

Figure 5: RLHF diagram highlighting components discussed in this section.

After learning a reward model, or, more commonly, interleaved with reward model learning, the next step is to train a policy that maximizes the expected accumulated reward. This section will discuss algorithms for policy learning, which can be categorized into two main techniques: adaptation of conventional RL algorithms and DPO.

### 6.1 Adaptation of RL Algorithms

Using the learned reward model, any standard RL algorithm (e.g., DQN, A3C, PPO, SAC) could potentially be applied to train a policy. However, in the setting of RLHF, this direct application may suffer from two

issues: The non-stationary nature of the learned reward function in RLHF and its inaccuracy, especially in the earlier stages of training. We will discuss these issues and possible adaptations of RL algorithms to address them in the following.

**Non-Stationary Rewards**  RL algorithms are designed to learn a policy that maximizes the expected accumulated reward in an MDP framework, which assumes a stationary reward function. The RLHF setting violates this assumption by periodically updating the reward model, leading to a non-stationary reward function.

Various works have empirically demonstrated that conventional RL algorithms can be applied nonetheless, with little to no modification. Christiano et al. (2017) argue that policy-gradient methods are better suited for non-stationary reward functions compared to value-based methods. They and various follow-up works successfully apply policy-gradient methods without any modifications in this setting. This approach has been picked up for language-model fine-tuning as well (Ouyang et al., 2022).

Later works have shown that value-based methods (possibly in an actor-critic scheme) can also be effective in RLHF (Ibarz et al., 2018; Lee et al., 2021b; Park et al., 2022; Liu et al., 2022; Xue et al., 2023). One trick to make value-based methods work is to use the reward model to relabel the experiences in the replay buffer whenever it is updated (Ibarz et al., 2018; Lee et al., 2021b). Similar to conventional RL, the use of such a replay buffer can greatly decrease the amount of environment interactions necessary for successful learning. As demonstrated by Gulcehre et al. (2023), the sample efficiency can be increased even further by using offline-RL techniques in a growing-batch RL setting, an offline-RL technique that iteratively increases the size of the dataset by policy rollouts while still being more sample-efficient than online RL.

In addition to the basic RL approaches, there are also some policy learning approaches tailored specifically for RLHF. Wu et al. (2023) propose a policy gradient algorithm, called Pairwise Proximal Policy Optimization (P3O), as an alternative to PPO, which avoids estimating the value function and at the same time is provably invariant with respect to equivalent rewards (unlike PPO). In a similar vein, Zhu et al. (2023b) replace the KL-regularization of PPO with a squared error term of the logarithmic probabilities, resulting in a seemingly more stable RL learner.

**Overoptimization of Approximate Rewards**  Since the learned reward model, which is only an approximation of the true reward function, is used to train a policy, overoptimization (Gao et al., 2023) or reward hacking (Skalse et al., 2022) can happen. Section 5.2.5 discusses the interdependence of the reward model and the policy in more detail as well as possible improvements from the reward model side, while here we focus on how to adapt policy training to cope with possibly inaccurate rewards in general.

One approach is to regularize the policy so as not to diverge too much from human-given demonstrations using KL-divergence regularization (Jaques et al., 2017). This is particularly common for language-model fine-tuning (Ouyang et al., 2022; Abramson et al., 2022), but Abramson et al. (2022) explores this for control as well. They found that this was important for some cases, in particular for deciding when to output language, but not for all. Going beyond KL-regularization, Moskovitz et al. (2024) investigate several techniques of constrained RL to only maximize rewards up to a threshold while avoiding excessive deviation from a pre-trained policy.

## 6.2   Framing RLHF for Generative Models as a Bandit Problem

So far, we have assumed that we ultimately want to solve a reinforcement learning problem represented by an MDP. However, especially with regard to the application of RLHF in the area of LLMs, there is now another simplified way of looking at the problem. Namely, as an instantiation of a (contextual) preference-based bandits problem (Bengs et al., 2021), which can of course be modeled by the more general case of a Markov decision process (MDP). In both cases, the focus is on the concept of tokens or rather sequences of tokens. However, in the MDP point of view, the state space $\mathcal{S}$ consists of all previous tokens and the prompt (represented as a sequence of tokens), while the action space $\mathcal{A}$ consists of all potential next tokens. A terminal state is often indicated here by the special token `<eos>` and trajectories are filled with this token until the maximum length $H$ is reached. Moreover, the transition function $P$ is degenerate (or deterministic),

with a value of one only for the state that is the concatenation of the current state and the taken action. A (latent) reward is only received at the end of the trajectory giving rise to a sparse feedback scenario.

The (contextual) preference-based bandits view, on the other hand, naturally considers no state space and no transition function, but an action space consisting of all possible responses to a prompt (both represented as a sequence of tokens). Here, the prompt specifies the context for which at least two actions are executed and for which a qualitative comparison is observed as feedback. In bandit literature, this is also referred to as a "(multi-)duel", coining the term *dueling bandits*. Thus, this point of view takes a trajectory-wise perspective, while the MDP point of view takes a token-wise perspective.

Note that the bandit formulation considers an entire episode (response in the LLM context) as an action with a single associated reward, resulting in sparse feedback from an MDP viewpoint. This is in contrast to the standard RLHF formulation as it is often used in control settings, where it is assumed that the reward of a trajectory is composed of the sum of the rewards of individual steps, which allows the optimizer to distribute rewards densely as best fits the data. On an intuitive level, this leads to state-action pairs that often occur in preferred trajectories to be highly rewarded, without necessarily putting all reward on the terminal actions. In practice, this can lead to nicely-shaped reward functions (Christiano et al., 2017), which cannot directly be achieved in the bandit setting. However, Chan et al. (2024) show how to take advantage of the predominantly used transformer architecture for the reward model in order to obtain a denser reward, even when assuming the bandit setting: More specifically, since the transformer architecture maintains attention weights in the last layer for each token, these can be used to attribute the overall reward signal to individual tokens.

The main appeal of the bandit formulation is that since the environment's dynamics are deterministic, exploration is simplified (although the space of trajectories to be explored is still huge, i.e., exponential with respect to the trajectory length). This setting even allows for policy improvements without any training using best-of-$n$-sampling (also called rejection sampling), a form of search that chooses the highest-rated response from multiple samples (Nakano et al., 2022; Menick et al., 2022). Most notably, however, this formulation notably enables direct policy learning approaches, such as DPO (Rafailov et al., 2023) or $\Psi$PO (Azar et al., 2024), discussed in the following section.

### 6.3 Direct Policy Optimization

The two-phase approach involving utility learning and policy optimization is not the only viable path to learning a policy from human feedback. While we have previously discussed the case in which we learn a reward function from observed preferences by assuming a human feedback model, an emerging branch of the literature circumvents the reward-learning step and uses preferences directly to address the actual RL problem. One important representative algorithm in this direction is DPO (Rafailov et al., 2023). The key trick in this method is to reparameterize the reward model as a function of its optimal policy, which allows the likelihood of the observed feedback (still using the Bradley-Terry model) to be rewritten as a function of the policy parameter. Thus, a policy can be directly trained from pairwise comparisons by minimizing the following negative log likelihood computed on a dataset $\mathcal{D}$ of triplets $(x, y_w, y_l)$ with $x$ being a context (e.g., prompt to LLM) and $y_w, y_l$ being two outputs (e.g., generated answers) such that $y_w \succ y_l$:

$$\mathcal{L}_{DPO}(\pi_\theta, \pi_{\text{ref}}) = -\mathbb{E}_{(x,y_w,y_l)\in\mathcal{D}} \left[ \log \sigma \left( \beta \log \frac{\pi_\theta(y_w \mid x)}{\pi_{\text{ref}}(y_w \mid x)} - \beta \log \frac{\pi_\theta(y_l \mid x)}{\pi_{\text{ref}}(y_l \mid x)} \right) \right] \tag{14}$$

where $\sigma$ is the logistic function and $\beta$ is a hyperparameter controlling the deviation with respect to a reference policy $\pi_{\text{ref}}$ (e.g., pretrained LLM). This loss amounts to increasing the probability of generating better outputs while decreasing that of generating worse ones with respect to the reference policy.

Many improvements, variations, or alternatives to DPO have been proposed. Since those methods are not the main focus of our survey, we refer the interested readers to their respective papers for more details, e.g., SLiC-HF (Zhao et al., 2023), OPPO (Kang et al., 2023), DPPO (An et al., 2023), PRO (Song et al., 2024), RSO (Liu et al., 2024b), SRPO (Choi et al., 2025), or by formulating policy search as a zeroth-order optimization (Tang et al., 2024). Azar et al. (2024) introduce an objective called $\Psi$-preference optimization ($\Psi$PO) that unifies the objective functions in DPO and RLHF. More specifically, for a specific instantiation

of $\Psi$, the objective in $\Psi$PO recovers DPO and SLiC-HF. In addition, DPO has been further generalized to include diverse divergence constraints (Wang et al., 2024a). Besides, Hejna et al. (2024) propose contrastive preference learning based on a regret preference model instead of the usual one in RLHF. It is also possible to learn a $Q$ function from human preferences directly, which implies a policy without the need for separate policy- and reward-model training (Myers et al., 2023).

It is worth noting that approaches for directly learning the policy from preferences have been considered in the past as well (Wilson et al., 2012; Fürnkranz et al., 2012; Wirth & Fürnkranz, 2013b;a; Busa-Fekete et al., 2014). In Sections 3.2.1 and 3.2.2 in the survey by Wirth et al. (2017), these methods are explained in more detail.

Another recent trend in fine-tuning models with human feedback is to even manage it without the usage of RL. An alternative is based on supervised reward learning with new types of loss functions (Lee et al., 2023; Yuan et al., 2023) or a specific learning process (Dong et al., 2023; Korbak et al., 2023). There are also RL-free approaches that do not use a reward model to train a policy to execute natural-language instructions using a transformer architecture (Brohan et al., 2023; Yu et al., 2023). On a related note, Liu et al. (2024a) suggest a way how to convert human feedback to natural language sentences for the task of fine-tuning language models.

# 7 Theory

The field of RLHF has recently made some progress in terms of theoretical results, which we will discuss in this section. First, we consider the contributions where the goal is to learn a provably (near) optimal policy both in an online and offline fashion or even in a way that falls in between. Then, we discuss and highlight recent contributions related to different theoretical aspects of RLHF, such as its relation to the standard reward-based RL. Tables 4 and 5 provide a concise overview of the results for the online or offline policy learning setting. Here, $\mathcal{N}_\mathcal{F}(\epsilon, d)$ denotes the $\epsilon$-covering number of a set $\mathcal{F}$ under some metric $d$[5]. It is worth mentioning that (almost) all works have two standard assumptions, namely that the reward function is bounded and that the ground-truth reward, human feedback model, or transition dynamics are elements of the considered model space, respectively.

## 7.1 Policy Learning

In the literature focusing on theoretical results, there is a distinction (similar to the distinction made in standard RL) between an offline and online setting. In the former, learning is based on a given fixed data set, usually previously collected through an interaction with the environment. In contrast, in the online environment, one interacts directly with the environment to learn from real-time feedback and continuously updates one's strategies based on the feedback received, allowing the agent to learn and adapt as it engages with the environment. Accordingly, an important component of the online variant is the sampling procedure, i.e., how the labels are selected. This is usually accomplished using an acquisition function that is based on uncertainty (see Section 4.1.1).

**Online Learning**  The first work dealing with the question of theoretical guarantees for learning an optimal policy from trajectory comparison feedback (see Section 3.2) in an online manner is by Novoseller et al. (2020). It laid the foundation for a paradigm embraced by many subsequent research endeavors: Adapting learning algorithms from the dueling or preference-based bandit literature (Yue & Joachims, 2009; Sui et al., 2018; Bengs et al., 2021) to the underlying situation with additional states. The preference-based bandit problem can be viewed as a preference-based RL problem with one state, so state transition dynamics must be considered accordingly for a fruitful adaptation. It is worth mentioning that Jain et al. (2015) used a quite similar idea before for feedback in the form of corrections (see Section 3.2) by resorting to the coactive learning setting (Shivaswamy & Joachims, 2012). Assuming the existence of a ground-truth context-trajectory scoring function and that the user's feedback is informative, the Preference Perceptron algorithm by Shivaswamy & Joachims (2012) is used and analyzed in terms of its cumulative regret.

---

[5]The $\epsilon$-covering number is the minimum integer $N$ such that there exists a subset $\mathcal{F}' \subset \mathcal{F}$ with $|\mathcal{F}'| = N$, and for any $f \in \mathcal{F}$, there exists some $f' \in \mathcal{F}'$ satisfying $d(f, f') \leq \epsilon$.

Table 4: An overview of approaches, their assumptions, goals, and properties for online policy learning. $\widetilde{\mathcal{O}}$ is used to hide log-factors. $T$ is the number of iterations of the respective algorithm.

| Algorithm (Reference) | Algorithmic approach | Assumptions | Target(s) and goal(s) of learner | Theoretical guarantee(s) |
|---|---|---|---|---|
| Dueling Posterior Sampling (DPS) (Novoseller et al., 2020) | Leveraging Posterior Sampling from dueling bandits | Linear link function, tabular MDP | Bayes regret minimization w.r.t. optimal policy based on trajectory comparison feedback | Asymptotic regret rate: $\mathcal{O}\left(\lvert\mathcal{S}\rvert\sqrt{\lvert\mathcal{A}\rvert T\log(\lvert\mathcal{A}\rvert)}\right)$ |
| Logistic Preference Reinforcement Learning (LPbRL) (Saha et al., 2023) | Leveraging MaxInP from contextual dueling bandits | Logistic link function, tabular MDP, linear rewards & $d$-dimensional feature embedding of trajectories | Expected regret minimization w.r.t. optimal policy based on trajectory comparison feedback | Transition dynamics: 1. Known $\widetilde{\mathcal{O}}\left(\lvert\mathcal{S}\rvert Hd\sqrt{T\log(T)}\right)$ 2. Unknown $\widetilde{\mathcal{O}}((\sqrt{d}+H^2+\lvert\mathcal{S}\rvert)\sqrt{dT} + \sqrt{\lvert\mathcal{S}\rvert\lvert\mathcal{A}\rvert TH})$ |
| Preference-based Optimistic Planning (PbOP) (Chen et al., 2022) | Leveraging MaxInP; general function approximation | General model classes $\mathcal{F}_{\mathbb{T}}$ for feedback and $\mathcal{F}_{\mathbb{P}}$ for transition dynamics with finite $l_2$-norm $\rho$-Eluder dimension $d_{\mathbb{T}}^{(2)}(\rho)$ and $d_{\mathbb{P}}^{(2)}(\rho)$. | High probability regret minimization w.r.t. optimal policy based on trajectory comparison feedback | $\widetilde{\mathcal{O}}\left(\sqrt{d_{\mathbb{P}}(\frac{1}{T})HT\log\left(\mathcal{N}_{\mathcal{F}_{\mathbb{P}}}\left(\frac{1}{T},d\right)\right)}\right)$ $+\widetilde{\mathcal{O}}\left(\sqrt{d_{\mathbb{T}}(\frac{1}{T})T\log\left(\mathcal{N}_{\mathcal{F}_{\mathbb{T}}}\left(\frac{1}{T},d\right)\right)}\right)$ $d$ being the $\ell$-infinity norm $\|\cdot\|_{\infty}$ |
| Preference-based Policy Search (PPS) (Xu et al., 2020) | Dynamic programming, policy search, $(\epsilon,\delta)$-PAC black-box dueling bandit algorithm and simulator | Uniform dependence of policy preference probabilities on value function differences, tabular MDP, $(\epsilon,\delta)$-PAC dueling bandit algorithm with $\Psi(K,\varepsilon,\delta)\varepsilon^{-\alpha}$ sample complexity for $K$ arms | $(\epsilon,\delta)$-PAC for optimal policy based on trajectory comparison feedback | Simulator step bound $\mathcal{O}\left(\frac{H^{\alpha+1}\lvert\mathcal{S}\rvert\Psi(\lvert\mathcal{A}\rvert,\varepsilon/H,\delta/\lvert\mathcal{S}\rvert)}{\varepsilon^\alpha}\right)$ Sample complexity bound $\mathcal{O}\left(\frac{H^\alpha\lvert\mathcal{S}\rvert\Psi(\lvert\mathcal{A}\rvert,\varepsilon/H,\delta/\lvert\mathcal{S}\rvert)}{\varepsilon^\alpha}\right)$ |
| Preference-based Exploration & Policy Search (PEPS) (Xu et al., 2020) | Similar to PPS, instead of simulator using an auxiliary synthetic reward function | Same as PPS and stochastic triangle inequality of trajectory comparisons preferences | $(\epsilon,\delta)$-PAC for optimal policy based on trajectory comparison feedback | Step complexity bound $\widetilde{\mathcal{O}}\left(\frac{H^{\alpha+1}\lvert\mathcal{S}\rvert^2\Psi(\lvert\mathcal{A}\rvert,\varepsilon/H,\delta/\lvert\mathcal{S}\rvert)}{\varepsilon^{\alpha+1}}\right)$ Comparison complexity bound $\mathcal{O}\left(\frac{H^\alpha\lvert\mathcal{S}\rvert\Psi(\lvert\mathcal{A}\rvert,\varepsilon/H,\delta/\lvert\mathcal{S}\rvert)}{\varepsilon^\alpha}\right)$ |
| UCBVI-Planning (Kong & Yang, 2022) | Optimistic least-squares value iteration, maximum information gain, value iteration based on pessimistic expected value function estimation | Binary rewards for state-action pairs based on human response model $f\in\mathcal{F}_H$ with bounded noise $\Delta>0$, compliant and tabular/linear MDP with dimension $d$ | $(\epsilon,\delta)$-PAC for optimal policy based on binary state-action reward feedback | Tabular MDP: $\mathcal{O}\left(\frac{H^4\lvert\mathcal{S}\rvert\lvert\mathcal{A}\rvert\log\left(\frac{H\lvert\mathcal{S}\rvert\lvert\mathcal{A}\rvert}{\epsilon\delta}\right)}{\epsilon^2}\right.$ $\left.+\frac{H^3\lvert\mathcal{S}\rvert^2\lvert\mathcal{A}\rvert\log\left(\frac{H\lvert\mathcal{S}\rvert\lvert\mathcal{A}\rvert}{\epsilon\delta}\right)}{\epsilon}\right)$ Linear MDP: $\mathcal{O}\left(\frac{\lvert\mathcal{A}\rvert^2d^5d_{\mathcal{F}_H}H^4\log\left(\frac{H\lvert\mathcal{S}\rvert\lvert\mathcal{A}\rvert}{\epsilon\delta\Delta}\right)}{\epsilon^2}\right)$ |
| Preference-based & Randomized Least-Squares Value Iteration (PR-LSVI) (Wu & Sun, 2024) | Least-squares value iteration with perturbed state-action-wise reward model | General differentiable link function $\Phi$, linear MDP, linear rewards with $d$-dimensional feature embedding of trajectories | Expected regret minimization w.r.t. optimal policy and/or low trajectory comparison feedback complexity steered by $\epsilon\in[0,1]$ | Expected regret bound: $\widetilde{\mathcal{O}}\left(\epsilon Td^{1/2}+\sqrt{T}\cdot d^3H^{5/2}\gamma\right.$ $\left.+d^{17/2}H^{11/2}\gamma^3\right)$ Comparison complexity bound: $\widetilde{\mathcal{O}}\left(d^4(\kappa+R_{\max})^2/\epsilon^2\right)$ $\kappa=\inf_{x\in[-R_{\max},R_{\max}]}\Phi'(x)$ |
| Algorithm for Policy Alignment in Reinforcement Learning (A-PARL) (Chakraborty et al., 2024a) | Iterative bilevel optimization via gradient descent based on an estimated policy gradient | Lipschitz assumptions on the objective function, the reward function, the parametric policy class, and convexity assumptions on the value function | Solving the bilevel optimization problem | Convergence rate: $\mathcal{O}(1/T)$ |

Novoseller et al. (2020) suggest the Dueling Posterior Sampling (DPS), which is an adaptation of the self-sparring algorithm (Sui et al., 2017). It takes a Bayesian perspective on the problem and defines a Dirichlet prior on the dynamics and a Gaussian prior on the rewards that are subsequently updated, while the trajectories to be compared by the human labeler are chosen based on their (posterior) probability of being optimal[6]. Assuming a linear link function (see Section 5.1.4) as well as a tabular MDP, it is shown that DPS is (i) consistent, i.e., converges in distribution to the optimal policy, and (ii) achieves an asymptotic expected regret bound (see Table 4).

Xu et al. (2020) combine dynamic programming and policy search with a black-box preference-based bandit algorithm for each state to design routines that return an almost optimal (a.k.a. $\varepsilon$-optimal) policy with high probability[7]. The first routine, called Preference-based Policy Search (PPS), requires access to a simulator, while the second routine, called Preference-based Exploration and Policy Search (PEPS), gets rid of this requirement by exploring the state space by means of an auxiliary synthetic reward function. By assuming that the probability of one policy dominating another policy is bounded uniformly over all states from below by a multiplicative of their value function, they show generic upper bounds for both routines on the number of pairwise trajectory comparisons (see Table 4). If these dominance probabilities have even more structural properties, such as fulfilling stochastic transitivity or stochastic triangle inequality (see Haddenhorst et al. (2020); Bengs et al. (2021)), then these upper bounds can be further refined.

A follow-up work by Saha et al. (2023) assumes a feature embedding of trajectories that gives rise to a feature embedding of policies and adapts the MaxInP algorithm (Saha, 2021) for contextual dueling bandits by essentially viewing the policy embeddings as the contexts. More precisely, assuming a logistic link function (see Section 5.1.4), confidence sets for the expected scores of the policies are constructed based on the maximum likelihood estimate (MLE), and the two policies with the highest uncertainty in terms of maximal variance are used to sample a trajectory, respectively, to be compared. In this way, the logistic preference-based reinforcement learning (LPbRL) is derived and also extended to the case of unknown dynamics by taking the uncertainty regarding the dynamics into account when constructing the confidence sets. For both cases, i.e., known or unknown dynamics, upper bounds on the regret of LPbRL are shown (see Table 4).

In contrast to previous work that all considers tabular MDPs, Chen et al. (2022) consider the case of a general unknown human feedback model and unknown dynamics each from function classes with a finite Eluder dimension[8] (Russo & Van Roy, 2013). They propose and analyze the Preference-based Optimistic Planning (PbOP) algorithm, which essentially follows a similar design as LPbRL but uses least-square estimates for the human feedback model and transitions dynamics along with confidence sets based on them. Moreover, Chen et al. (2022) derive lower bounds for the regret of any learning algorithm by reducing the once-per-episode-feedback RL problem (Chatterji et al., 2021) to the PbRL problem. Finally, they extend their analysis to the case of $K$-wise comparisons, where one obtains all $\binom{K}{2}$ pairwise comparisons for $K$ many queried trajectories. In essence, the regret term coming from the human feedback model class improves by a factor of $\sqrt{K}$.

Wu & Sun (2024) consider a similar learning scenario as Saha et al. (2023) but with the additional objective to keep the number of queries of trajectory comparisons low, which is a combination of two competing objectives also studied in the bandit literature (Degenne et al., 2019). For this purpose, they suggest the Preference-based and Randomized Least-Squares Value Iteration (PR-LSVI) algorithm, which combines least-squares value iteration with a perturbed state-action-based reward model with Gaussian noise for regret minimization; a similar idea to CoLSTIM suggested for contextual dueling bandits (Bengs et al., 2022). More specifically, in each time step, the policy maximizing the value function of the perturbed state-action-based reward model and the policy maximizing the latter in the previous time steps are "played". By sampling trajectories of these two policies and computing their expected absolute reward difference (based on the perturbed state-action-based reward model) as a measure of uncertainty, preference feedback for these two trajectories is queried if the uncertainty exceeds a certain threshold. Moreover, they also suggest a posterior

---

[6]The latter probability is assessed by posterior sampling; a commonly used technique in the bandit literature used by so-called Thompson Sampling strategies, see Lattimore & Szepesvári (2020) for more details.

[7]This is a so-called PAC learning setting (Valiant, 1984) in which the goal of finding the/an optimal object is relaxed to finding a "good enough" object, usually specified by some distance measure on the object domain.

[8]Roughly speaking, the Eluder dimension of a function class refers to the number of worst-case errors one must make to identify an unknown function from that class.

sampling counterpart of this algorithm, the Preference-based Thompson Sampling (PbTS) algorithm, and analyze it in terms of Bayesian quantities.

Recently, Chakraborty et al. (2024a) proposed a bilevel optimization problem that generalizes the standard optimization problem for RLHF with trajectory feedback and the negative log-likelihood as a loss function (i.e., (2)). This problem, which they call PARL (Policy Alignment in Reinforcement Learning), is characterized by explicitly taking into account the dependence on the data-collecting process at one level for the optimal policy parameters at the other level. For this problem, A-PARL is proposed, which is shown to have an $O(1/T)$ convergence rate under specific assumptions, where $T$ is the number of iterations.

Finally, for the LLM training scenario, there are two perspectives on the problem (MDP vs. contextual dueling bandits) as mentioned in Section 6. Accordingly, work from the field of contextual preference-based bandits (Dudík et al., 2015; Saha, 2021; Saha & Krishnamurthy, 2022; Bengs et al., 2022; Sekhari et al., 2023) can also be viewed as theoretical contributions for the RLHF setting for the LLM application. This view is advocated in particular by Xiong et al. (2024), who view the LLM fine-tuning task as a reverse-KL regularized contextual bandit problem. Typically the context is chosen externally, but Mehta et al. (2023) consider the learning variant in which the learning agent chooses the context as well. This variant is referred to as active contextual dueling bandits.

**Offline Learning**  Zhu et al. (2023a) study the performance of a greedy policy trained from a data set consisting of trajectory pairs along with the observed preference that is assumed to be generated by means of a Bradley-Terry model with linear rewards. For this purpose, different results with respect to the MLE of the Bradley-Terry model for different feedback scenarios are derived that are quite of independent interest. In particular, they show concentration inequalities of the MLE for trajectory-based comparison feedback and additionally its asymptotic normality for action-based comparison feedback that also holds for $K$-wise comparisons. Based on these, it is shown that the greedy policy using the MLE in the case of action-based feedback might fail while using a pessimistic MLE leads to minimax-rates with respect to the performance gap[9]. The latter is also shown to be true in the case of trajectory-based feedback. Technically, the pessimistic MLE is realized by taking the policy that has the largest pessimistic expected value function, i.e., the lowest realization of the value function within a hyperparameter-dependent confidence region around the MLE. Further results of independent interest are the inferred theoretical guarantees for maximum entropy inverse RL (Ziebart et al., 2008) and action-based inverse RL algorithms (Neu & Szepesvári, 2009).

The simple model assumptions underlying (Zhu et al., 2023a) were then replaced by more sophisticated assumptions in some subsequent work. The linear reward assumption has been replaced by more general reward function classes by Zhan et al. (2024a) and Li et al. (2023). In addition, Zhan et al. (2024a) also consider more general unknown human feedback models and construct the confidence regions for the pessimistic approach directly from the log-likelihood function. The resulting approach, called FREEHAND, is analyzed in terms of its performance gap, for which some problem-dependent coefficients, the per-step, per-trajectory, and transition concentrability coefficient, are introduced. On the basis of a lower bound, it is shown that the per-trajectory concentrability coefficient should naturally appear in the bound on the performance gap. Moreover, the concentrability coefficient is shown to be upper bounded by the constant appearing in the special case of linear rewards considered by Zhu et al. (2023a). Finally, it is worth mentioning that both trajectory-based and action-based comparison feedback are considered.

In follow-up work, Zhu et al. (2024) found overfitting as well as overoptimization issues of the MLE in the Boltzmann model for pairwise comparison feedback. This can arise in particular if the observations of labels are strongly unbalanced and thus the utilities can become infinite. To overcome this problem, they propose the Iterative Data Smoothing (IDS) algorithm, which implicitly weights observed labels appropriately by their frequency and their current likelihood. Note that these issues do not contradict the results shown by Zhu et al. (2023a) as these are based on the assumption of bounded utilities (or rewards).

Assuming a dynamic discrete choice model (Rust, 1987) underlying the given data set of observed trajectories (without explicitly observed preferences), Li et al. (2023) suggest the Dynamic-Choice-Pessimistic-Policy-Optimization (DCPPO) algorithm. It first estimates the reward model using this assumption and then learns

---

[9]The expected difference between the optimal value function and the value function of the used policy.

Table 5: Overview of approaches, their assumptions, goals, and properties for offline policy learning with a data set of size $n$. $\widetilde{\mathcal{O}}$ is used to hide log-factors. $R_{\max}$ is a bound on the reward.

| Algorithm (Reference) | Algorithmic approach | Assumptions | Target(s) and goal(s) of learner | Theoretical guarantee(s) |
|---|---|---|---|---|
| Pessimistic MLE (Zhu et al., 2023a) | Greedy policy for pessimistic expected value function estimation | Logistic link function, linear reward function for a state-pair feature embedding with some regularity assumptions on weights, known transition dynamics | High probability bound for the performance gap based on trajectory-based (and action-based) feedback | $\mathcal{O}\left(e^{2HR_{\max}}\sqrt{\frac{d+\log(1/\delta)}{n}}\right)$ |
| oFfline ReinforcemEnt lEarning with HumAN feeDback (FREEHAND) (Zhan et al., 2024a) | Greedy policy for pessimistic expected value function estimation | General differentiable link function $\Phi$, general bounded reward function class $\mathcal{F}_r$ and general transition dynamic class $\mathcal{F}_{\mathbb{P}}$ | High probability bound for the performance gap based on trajectory-based (and action-based) feedback | Transition dynamics: 
 1. Known 
 $\mathcal{O}\left(\sqrt{\frac{C_r^2\kappa^2\log(\mathcal{N}_{\mathcal{F}_r}(1/N,|\cdot|)/\delta)}{n}}\right)$ 
 2. Unknown 
 $\mathcal{O}\left(\sqrt{\frac{C_r^2\kappa^2\log(\mathcal{N}_{\mathcal{F}_r}(1/N,|\cdot|)/\delta)}{n}}\right)$ + 
 $\mathcal{O}\left(R_{\max}\sqrt{\frac{C_P^2\kappa^2\log(\mathcal{N}_{\mathcal{F}_{\mathbb{P}}}(1/N,|\cdot|)/\delta)}{n}}\right)$ 
 $\kappa = \inf_{x\in[-R_{\max},R_{\max}]}\Phi'(x)$ 
 $C_r$, $C_P$ reward and transition concentrability coefficient |
| Dynamic-Choice-Pessimistic-Policy-Optimization (DCPPO) (Li et al., 2023) | Value iteration based on pessimistic expected value function estimation | Dynamic discrete choice model, linear MDP, linear reward function for a state-pair feature embedding with some regularity assumptions on weights, known model class entailing the value and reward function of the dynamic discrete choice model | High probability bound for the performance gap based on action-based feedback | Linear model class: 
 $\mathcal{O}\left(|\mathcal{A}|d^{3/2}H^2e^H\sqrt{\frac{\log(dHn/\delta)}{n}}\right)$ 
 RKHS model class with different eigenvalue decay: 
 $\mathcal{O}\left(\tilde{d}He^H|\mathcal{A}|\sqrt{\mu\log(nR_{\max}H/\delta)}\right)$ 
 $\mu$-finite spectrum, 
 $\mathcal{O}\left(\tilde{d}He^H|\mathcal{A}|\sqrt{(\log(nR_{\max}H)/\delta)^{1+1/\mu}}\right)$ 
 $\mu$-exponential decay, 
 $\mathcal{O}\left(\tilde{d}He^H|\mathcal{A}|(nR_{\max})^{\kappa^*}\sqrt{\log(nR_{\max}H/\delta)}\right)$ 
 $\mu$-polynomial decay, 
 $\kappa^* = \frac{d+1}{2(\mu+d)} + \frac{1}{\mu(1-2\tau)-1}$, 
 $\tilde{d}$ =population effective · sampling effective dimension |
| LCBVI-Tabular-Offline (Kong & Yang, 2022) | Maximum information gain for reward querying, value iteration based on pessimistic expected value function estimation for policy learning | Binary rewards for state-action pairs based on human response model with bounded noise, compliant and tabular MDP | High probability bound for the performance gap based on binary state-action reward feedback | Linear model class: 
 $\mathcal{O}\left(H\sqrt{|\mathcal{S}|\log(|\mathcal{S}||\mathcal{A}|Hn/\delta)}\right.$ 
 $\left. \cdot \mathbb{E}_{\pi^*}\left[\sum_{h=1}^{H}(N_h(s_h,a_h)+1)^{-1/2}\right]\right)$ 
 $N_h$ are numbers of visit time |

a policy in a (pessimistic) value iteration manner from the estimated reward model. In the case of a linear MDP and a known model class that entails both the value and the reward function of the dynamic discrete choice model, DCPPO is analyzed with respect to its performance gap. This is done for the case of a linear function model class as well as a subset of a reproducing kernel Hilbert space (RKHS) as the model class.

Focusing on the estimation of the weight parameter in the Bradley-Terry model for the action-based feedback under label differential privacy conditions (Dwork, 2008), Chowdhury & Zhou (2023) analyze two estimation procedures, MLE and stochastic gradient descent (SGD), under similar assumptions as in Zhu et al. (2023a). In both cases, the cost of ensuring label differential privacy is a multiplicative factor.

Reward collapse, a term introduced by Song et al. (2023), describes the issue when rank-based training methods for LLMs lead to the same reward distribution regardless of the prompts used in the final training steps. The authors show that this occurs because the rank-based approach does not adequately account for prompt-related information. To address this problem, the authors propose a family of utility functions as well as an optimization method that successfully creates prompt-dependent reward distributions, effectively mitigating the collapse of rewards during training.

**Blending Online and Offline Learning** Kong & Yang (2022) study the problem of optimal policy learning from critique feedback (see Section 3.2), i.e., binary rewards for state-action pairs, with as few queries to the human as possible. They assume an underlying ground-truth human feedback model that leads to a positive evaluation for a state-action pair if it exceeds a specific threshold evaluated at that pair. In addition, the learning process consists of two phases: First, exploring the environment in an unsupervised manner, and then querying user feedback in an active reward learning phase to learn the human feedback model. This learning process is again analyzed in two variants: Either the exploration phase was performed externally, and a data set consisting of trajectories is provided (offline), or this data set is actively collected itself (online). For both variants, an active learning algorithm is proposed that essentially selects query points (state-action pairs) that provide the most information gain given the points already designated to be queried. For the online variant, an exploration strategy based on optimistic least-squares value iteration (Jin et al., 2020) is also introduced for tabular or linear MDPs. In both variants, policy learning is carried out by a pessimistic value iteration with the empirical transitions and the estimated reward function, resulting in UCBVI-Planning (online) and LCBVI-Tabular-Offline (offline). Under the assumption of bounded noise (Massart & Nédélec, 2006) or low-noise assumption (Korba et al., 2017; Haddenhorst et al., 2021), bounds on the performance gap of both algorithms are derived.

The question of the ideal experimental design for RLHF is addressed by Zhan et al. (2024b), in particular, how to separate the process of data acquisition (e.g., trajectories to be evaluated) from the process of retrieving human feedback to avoid constantly involving humans in the training loop. Assuming linear rewards, the Bradley-Terry model and either a transition oracle (e.g., available for tabular or low-rank MDPs) or a linear MDP they suggest the expeRimental dEsiGn for queryIng huMan prEference (REGIME) algorithm that first samples exploratory trajectories indented to be as informative as possible for learning the reward via MLE and then applies a greedy policy based on the reward learned by the latter. They explicitly show that REGIME requires less human feedback to be queried in order to output an $\epsilon$-optimal policy at the end than the approach by Saha et al. (2023).

## 7.2 Preference-Based vs. Reward-Based Learning

There have been some theoretical analyses regarding the question of how far, or if at all, preference-based feedback in the form of trajectory comparisons is more suitable compared to numerical feedback. Ji et al. (2023c) suggest a human rating model for this purpose in the numerical feedback case and analyze the LCB algorithm (Jin et al., 2021) in order to compare it with the pessimistic MLE (Zhu et al., 2023a). It is shown that under specific assumptions, LCB has a constant performance gap, while the preference-based pessimistic MLE under similar assumptions has a similar bound as in Table 5.

Wang et al. (2023b) provide reduction-based algorithms that can directly use state-of-the-art results in reward-based RL for RLHF with utility-based and general state-action and trajectory-based comparison feedback. They show, in general, how theoretical results of the underlying standard RL algorithm can be

translated to theoretical results for the resulting preference-based RL algorithm. For some special cases, such as MDPs with finite Eluder dimension and utility-based preference feedback, the theoretical guarantees are explicitly derived using state-of-the-art RL algorithms that are qualitatively similar to explicit preference-based RL algorithms.

### 7.3 Nash Learning from Human Feedback

The majority of theoretical works use the modeling of (pairwise comparison) feedback by means of a link function (see Section 5.1.4). Even if this often leads to simpler derivations, this modeling has the decisive disadvantage that it imposes transitivity of the human feedback that does not necessarily prevail in reality. In other words, it is quite possible that preference cycles can occur. For this reason, there is a new direction in theoretical work that dispenses with parametric modeling of the preference probability similar to Chen et al. (2022) but uses it to formulate a new learning objective. Specifically, the problem is considered from a game theory perspective, where two policies each propose a trajectory that should be highly preferred by the human user. Thus, the goal is to find a policy that suggests trajectories that are preferred to the trajectories of any other policy, i.e., a Nash equilibrium or a von Neumann winner.

This learning variant was first considered by Wang et al. (2023b), who showed that the problem can be reduced to finding restricted Nash equilibria in a multi-agent RL problem (based on numerical rewards). For special situations of the latter problem, wrapper algorithms are proposed that have been shown to find the von Neumann winner with high probability. The learning problem was recently taken up and analyzed by Munos et al. (2024) and Ye et al. (2024) in a KL-regularization variant. While the former considers the online-learning setting assuming a known preference model, the latter considers both online as well as offline learning settings and the preference model belonging to a finite function class.

## 8 Applications and Benchmarks

The field of RLHF has advanced significantly in the last few years, with increasing interest driven by prominent applications. First and foremost are applications to large language models, exemplified by Chat-GPT (OpenAI, 2022). This section starts by providing a sample of such applications, showcasing how this technology is being used in fields as varied as robotics, language processing, image generation, and more. We will also delve into libraries that provide foundational support for RLHF research, enabling researchers and practitioners to experiment with and refine a range of approaches. We then explore a spectrum of benchmarks that have been developed to standardize and simplify the evaluation of new approaches, offering insights into their performance in different settings. Finally, and closely related to those benchmarks, we will discuss common evaluation practices.

### 8.1 Applications

RLHF finds applications across various domains, showcasing its versatility in addressing complex and nuanced tasks. The most prominent application is ChatGPT (OpenAI, 2022), which is an example of an application in the domain of language models. Beyond that, however, applications extend across diverse domains such as control tasks, generative models, and recommender systems. This section provides an overview of notable works applying RLHF in different areas.

**Control and Interactive Environments** There is a long history of using control environments as benchmark tasks for RL. In addition to the breadth of available environments, control applications are of particular interest because tasks are often hard to specify. Christiano et al. (2017) demonstrated the effectiveness of RLHF in games as well as simulated continuous control tasks, matching the performance of RL agents trained on ground-truth rewards with a fraction of the feedback. Extending to robotics, Ding et al. (2023) trained a reward model for diverse tasks with a single robot, achieving human-like behavior. Kupcsik et al. (2018) applied RLHF for precise robot-to-human handovers. Similarly, Abramson et al. (2022) used RLHF in the Playhouse simulator, a platform for sensorimotor task training, and Milani et al. (2022) showcase

an application in the context of the MineRL Basalt competition for Minecraft tasks. Recently, Dong et al. (2024b) use RLHF to guide a diffusion-based planning model.

**Generative Models in Language and Imaging**  Generative models, i.e., models that generate new data instead of just predicting labels, can be framed as an RL setting in which a policy assembles the output through its actions. In the context of language models, this means that the language model is interpreted as a policy with tokens as actions. Using this reframing, we can use RLHF approaches to fine-tune generative models to produce preferred outputs. ChatGPT (OpenAI, 2022) and GPT-4 (OpenAI, 2023) are prime examples of language models fine-tuned using RLHF. These applications build on earlier work, such as by Ouyang et al. (2022), Ziegler et al. (2020) and Glaese et al. (2022). This method extends to text summarization (Gao et al., 2018; 2020; Stiennon et al., 2020), dialogue summarization (Chen et al., 2023), and question answering (Nakano et al., 2022; Menick et al., 2022). In image generation, Lee et al. (2023) and Xu et al. (2023) demonstrate the use of reward modeling for text-to-image tasks, while Pinto et al. (2023) and Kazemi et al. (2020) explore RLHF applications in broader computer vision tasks. Interestingly, in the context of LLMs, reward learning has also been expressed as density estimation (Dumoulin et al., 2024) instead of the supervised approach described in Section 5.

**Recommender Systems**  In the context of recommender systems, Xue et al. (2023) have shown the potential of RLHF in optimizing for long-term engagement. Although it is, in principle, possible to algorithmically evaluate policies in this domain, these rewards are sparse. To combat this, Xue et al. (2023) use RLHF to distill sparse, global feedback into a dense reward model.

These diverse applications underscore the adaptability of RLHF and its growing importance in various technological domains, paving the way for innovative solutions and enhanced human-computer interactions.

## 8.2  Supporting Libraries

Several libraries have emerged that aim to provide a toolset for implementing and experimenting with RLHF and reward learning algorithms, contributing to the ease and efficiency of research and development. One notable example is the `imitation` library (Gleave et al., 2022b). It encompasses a collection of imitation and reward learning algorithms, including those introduced in the seminal work by Christiano et al. (2017). In the offline realm, `Clean-Offline-RLHF` (Yuan et al., 2024) provides implementations for offline RL algorithms with human feedback. Two other libraries, `APReL` (Bıyık et al., 2022b) and `POLAR` (Tucker et al., 2022), focus on the Bayesian setting. Bıyık et al. (2022b) provide a specialized framework for preference-based reward learning with a focus on Bayesian methods. Meanwhile, Tucker et al. (2022) introduce a framework designed for Bayesian reward learning from multiple feedback types, including pairwise preferences, in MATLAB. Finally, in the domain of language model fine-tuning, the `trlX` library (Castricato et al., 2023) offers a toolkit specifically designed for language model training. It specializes in the fine-tuning of transformer-based language models, treating the language model as the policy in an RLHF setup.

Due to the many interacting components and the human element in RLHF research, implementing new ideas and running experiments can be quite challenging. The discussed libraries reduce this challenge and make RLHF research more approachable to many researchers.

## 8.3  Benchmarks

Due to the difficulty of reproducible evaluations without a ground-truth objective and with humans in the loop, benchmarks play an important role in advancing and evaluating RLHF approaches. Several benchmarks have been proposed, each focusing on different applications and challenges.

One such benchmark is B-Pref (Lee et al., 2021a), which focuses on control tasks with synthetic feedback. B-Pref aims to provide simulated human feedback that captures some irrationalities, thereby coming closer to evaluation with real human feedback than other approaches. At the same time, by relying entirely on synthetic feedback, the results are reproducible and cost-effective to generate. In a similar vein, Freire et al. (2020) propose a set of environments designed to diagnose common problems in reward learning. These environments help in identifying and addressing the typical challenges that arise in RLHF scenarios.

The offline RLHF setting is particularly well-suited for benchmarks, as it allows for the use of static datasets. Shin et al. (2023) evaluate pre-existing offline RL benchmarks for their suitability for RLHF evaluation, and find that many are ill-suited due to the simplicity of the required reward function. They do, however, identify a subset of these benchmarks together with their own addition for evaluation. While Shin et al. (2023) leverage synthetic rewards, Yuan et al. (2024) propose a dataset and benchmark for offline RLHF, including preference data. This helps to circumvent the challenges of synthetic feedback and benchmark reproducibility with real feedback.

The MineRL BASALT competition (Shah et al., 2021b; Milani et al., 2022) gives a more application-driven benchmark with a complex environment. The competition proposes the challenge of solving tasks defined by natural language descriptions in Minecraft based on human feedback. Writing hand-engineered reward functions is very challenging in that setting, which makes it a good benchmark for methods based on human feedback. The competition is method-agnostic in principle, and non-RL approaches such as behavioral cloning are also considered. While the initial dataset consists of human demonstrations, the competition is agnostic for the feedback type, which may include demonstrations, comparisons, and others. The final evaluation is performed by humans through pairwise comparisons.

In the domain of language modeling, Truthful QA (Lin et al., 2022) serves as a benchmark that measures the truthfulness of models. Also, in the context of language models, Ramamurthy et al. (2023) introduce a set of pretrained reward models, learned from human feedback, as benchmarks. These models serve as reference points for evaluating new RLHF techniques against established standards.

For evaluating language reward models specifically, Lambert et al. (2024) introduce RewardBench, a benchmark designed to test reward model performance across chat, safety, reasoning, and preference learning tasks. The benchmark includes challenging test cases such as adversarial prompts and edge cases where reward models commonly fail, providing standardized evaluation for both classifier-based and DPO-trained reward models. RewardBench has become a widely-used evaluation framework with a public leaderboard that tracks performance of submitted reward models.

Together, these benchmarks provide a diverse and comprehensive suite of tests that drive the development and refinement of RLHF methods, ensuring they are robust, effective, and capable of handling a wide range of real-world scenarios.

## 8.4 Datasets

Due to its interactive and online nature, RLHF research often does not rely on static datasets. This is because the feedback is generally collected interactively and depends on the current policy. When the reward model is not refined iteratively, however, as is common practice for the related settings of LLM fine-tuning and offline RLHF, static datasets can be used. Such a static dataset can significantly simplify the development and evaluation of RLHF methods.

Since language model fine-tuning is a popular application of RLHF and generally does not iteratively refine the reward model, many datasets have been developed for this purpose. Particularly notable are `hh-rlhf` (Bai et al., 2022a) and `PKU-Safe-RLHF` (Ji et al., 2023a), two datasets focusing on harmless and helpful responses, the OpenAssistant datasets (`oasst1`, `oasst2`) (Köpf et al., 2023), containing not only response rankings but also ratings on various dimensions, the `summarize_from_feedback` dataset (Stiennon et al., 2020) focusing on preferences over text summaries, the Stanford Human Preferences Dataset (`SHP`) (Ethayarajh et al., 2022),[10] which is based on Reddit responses, the WebGPT dataset (`webgpt_comparisons`) (Nakano et al., 2022), focused on long-form question answering and the `HelpSteer` (Wang et al., 2024d) dataset, which is not based on preferences but instead gives ratings on for 4 attributes (helpfulness, correctness, coherence, complexity) for each response.

Although static datasets are used more rarely in the control setting, some datasets have been developed for offline RLHF in this domain. Concretely, Yuan et al. (2024) propose the `Uni-RLHF` dataset and a benchmark

---

[10]This paper introduces the V-usable information framework for measuring dataset difficulty. The SHP dataset was created after the paper's publication, using its techniques to identify learnable preference pairs from Reddit data.

for offline RLHF while Kim et al. (2023) publish a dataset of real human preferences for typical offline RL tasks (D4RL, Robosuite).

## 8.5 Evaluation

Evaluating RLHF poses unique challenges, particularly in scenarios without precise ground-truth task specifications. Evaluations generally focus on either the learned policy or the reward model, each shedding light on different aspects of system performance.

**Policy Evaluation**  Assessing learned behavior is crucial for the evaluation of an RLHF system. In domains with ground-truth rewards, these can be used for policy evaluation (Christiano et al., 2017). However, many RLHF applications lack this clarity. Ouyang et al. (2022), for instance, evaluate the quality of language model responses by having labelers rate the output quality on a test set of prompts, highlighting the significance of human judgement in assessing model outputs. Jain et al. (2015) use direct scores on a Likert scale for evaluations, including self-assessments by trainers and cross-evaluations by others. Losey et al. (2022) extend this with a survey based on scores and free-form participant comments, comparing evaluations based on known true rewards with subjective experiences. Moreover, Abramson et al. (2022) employ a multi-stage evaluation scheme that includes scripted probe tasks, a standardized test suite evaluated by humans, and full interactive assessments, demonstrating the need for diverse and thorough evaluation methodologies in RLHF.

**Reward Model Evaluation**  Direct reward model evaluation complements policy assessment. While reward model accuracy is a more direct measure of preference-learning success, the ultimate goal is inducing effective policies. A perfectly accurate reward model is often not necessary to induce a good policy, which is the actual goal of RLHF. Therefore, both evaluation methods are ideally used in combination.

Jain et al. (2015) also use a ranking loss method for test sets of trajectories, compared against expert evaluations with known scores. This approach provides quantitative measures of the reward model's fidelity. In addition, Wilde & Alonso-Mora (2022) compare parameter-based and reward-based evaluation measures for learned reward functions, identifying strengths and weaknesses in both methods and contributing to a more nuanced understanding of reward model assessment in RLHF. These approaches provide a quantitative measure of the reward model's accuracy in reflecting human preferences and expert judgements. For a detailed discussion of reward model evaluation, also refer to Section 5.3.

Policy- and reward model evaluation both offer insights into the performance of an RLHF approach. Ideally, both measures should be combined to enable quick iteration and give insights into both the preference learning performance as well as the quality of the learned behavior.

## 9 Discussion and Conclusion

In this survey, we have provided an overview of the current state of RLHF, tracing its evolution from PbRL and examining its broad applications across various domains like control, natural language processing, and computer vision. Given the rapid expansion of this field, our coverage necessarily has limitations in addressing every extension and application in full depth. In particular, given the rapidly evolving nature of LLM research, our coverage of techniques specific to this domain is less exhaustive than our treatment of control and robotics applications. In this concluding section, we discuss key extensions and alternative approaches, identify open questions, and highlight future research directions.

### 9.1 Extensions and Related Approaches

While our survey focuses on RLHF methods that learn reward functions online from human feedback, several promising approaches extend beyond this scope while addressing the same core challenge of learning human-aligned objectives.

An increasingly important alternative is **RL from AI feedback** (RLAIF), which replaces human feedback with evaluations from pre-trained AI systems. This approach leverages foundation models as preference sources and has demonstrated success across diverse applications: language model fine-tuning (Bai et al., 2022b; Sun et al., 2024), generating intrinsic motivation for text-based games (Klissarov et al., 2024), and learning rewards (Wang et al., 2024c) or coding reward functions (Ma et al., 2024b; Xie et al., 2024) for control tasks. A hybrid approach (*assisted evaluation*) involves AI systems assisting rather than replacing human evaluators, such as generating critiques of model outputs (Saunders et al., 2022) or enhancing responses with metadata such as citations (Nakano et al., 2022; Menick et al., 2022).

While most work on RLHF implicitly assumes that tasks can be specified by maximization of expected accumulated scalar rewards, this *reward hypothesis* (Silver et al., 2021) is under active debate in the RL community (Lambert, 2021; Vamplew et al., 2022; Bowling et al., 2023; Skalse & Abate, 2022). Skalse & Abate (2024) investigate the sensitivity of inverse RL to reward misspecification, and recent RLHF approaches are beginning to move **beyond scalar rewards** through more complex objective functions, such as multi-objective frameworks with non-linear aggregation of vector rewards (Qian et al., 2023).

Finally, the RLHF framework has inspired numerous **extensions across RL domains** that revisit classic RL topics through the lens of human feedback. These extensions span three main categories: fundamental algorithmic concerns such as exploration (Liang et al., 2022), reward feature learning (Katz et al., 2021), hindsight experience replay (Zhang et al., 2023a), and reward shaping (Xiao et al., 2020); advanced learning paradigms including multi-task learning (Ouyang et al., 2022; Abramson et al., 2022; Myers et al., 2023), continual learning (Zhang et al., 2024), and hierarchical RL (Pinsler et al., 2018); and specialized domains such as risk-sensitive (Chen et al., 2024), safe (Dai et al., 2024; Cosner et al., 2022), and fair RL (Siddique et al., 2023). Additionally, as discussed in Section 4.2, the intersection of RLHF with human-computer interaction (HCI) offers rich opportunities for advancing feedback collection mechanisms. It is crucial to keep human psychology in mind when designing these systems and to learn from other related fields that already studied such issues extensively.

## 9.2 Current Limitations

Despite its successes, RLHF faces several limitations that constrain its broader applicability. Casper et al. (2023) offer a thorough analysis of open problems and fundamental limitations of RLHF; beyond that, we highlight several key limitations in the following.

A fundamental limitation concerns **feedback quality and interpretation**. Without assistance during feedback, RLHF is limited by the tasks humans can reliably judge (Leike, 2022; Leike et al., 2018; Wu et al., 2021; Christiano et al., 2018) – a problem exemplified in language model fine-tuning, where humans often prefer assertive but incorrect responses (Hosking et al., 2024). Furthermore, current approaches often fail to learn the actual causes of human feedback, learning spurious correlations instead (Tien et al., 2022).

Current approaches also struggle with **awareness of gaps in preference understanding**. The common separation between the policy and the reward model limits how the agent can reason about its knowledge of human preferences. This limitation may be addressed by tighter integration such as in the cooperative inverse RL setting (also called assistance games) (Hadfield-Menell et al., 2016; Shah et al., 2021a), where the agent simultaneously learns about human preferences and acts to maximize them, enabling behaviors like conditional planning while awaiting feedback and relevance-aware active learning. More broadly, the limited ability to reason about preference knowledge represents a fundamental constraint of current RLHF approaches.

From a **theoretical perspective**, a primary challenge lies in relaxing underlying assumptions. This requires balancing assumptions that are not overly restrictive for practical use cases while maintaining feasibility of theoretical guarantees for computationally efficient algorithms. Key questions include whether algorithms can avoid actively maintaining a policy space and eliminate sub-optimal policies, or relying on computation oracles. Recent work, such as by Wang et al. (2023b) or Wu & Sun (2024), gives hope that this may be possible.

Beyond technical limitations, RLHF raises important **ethical and social considerations** that require careful attention. These include fundamental questions about whose preferences should guide alignment, ensuring ethical data collection practices that respect privacy, mitigating the amplification of human biases, and addressing incentives for labeler manipulation (Armstrong et al., 2020; Carroll et al., 2023) inherent in current training methodologies. While some technical solutions are emerging, such as **privacy-preserving alignment** through differential privacy techniques (Wu et al., 2024), addressing these concerns comprehensively requires coordinated progress across technical research, ethical frameworks, and governance structures.

### 9.3 Conclusion

Although RLHF has significantly contributed to advancements in LLMs and other areas of ML, it remains a domain in its infancy with many unanswered questions and inherent limitations. These challenges present opportunities for further advancements in theory and practice, potentially resulting in more robust algorithms that make more efficient use of human feedback. It remains intriguing to what extent RLHF will continue to shape natural language processing, RL, robotics, AI alignment, and beyond.

**Acknowledgements**  We thank Tom Bewley, Andreea Bobu, Adam Gleave, Erdem Bıyık, Yannick Metz, Peter Stone, and Banghua Zhu for their feedback on earlier versions of this survey. This publication was supported by LMUexcellent, funded by the Federal Ministry of Education and Research (BMBF) and the Free State of Bavaria under the Excellence Strategy of the Federal Government and the Länder as well as by the Hightech Agenda Bavaria. This work has also been supported in part by the program of National Natural Science Foundation of China (No. 62176154) and a collaborative project funded by NetEase. EH has received funding from the European Union's Horizon Europe research and innovation programme under the Marie Sklodowska-Curie grant agreement No 101073307.

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

# A    RLHF for Language Models

Generative models, particularly LLMs, have significantly driven recent interest and advances in RLHF. These models are generally trained on vast amounts of data using self-supervised pretraining objectives that are not aligned with their intended usage. RLHF can be used for preference-based fine-tuning of these models, enabling them to generate more helpful, honest, and harmless outputs (Askell et al., 2021). This has highlighted the potential of RLHF to enhance the performance and adaptability of intelligent systems across a wide range of applications.

The application of RLHF to LLMs can be traced back to early works focused on summarization (Böhm et al., 2019; Ziegler et al., 2020; Stiennon et al., 2020). This line of work considers a setting where a human provides feedback on language model outputs, such as text summaries, which is then used to train a reward model that guides the language model towards generating better outputs. In order to apply RLHF techniques to this setting, the language model is treated as an RL agent, with the reward model providing the feedback that guides the agent's learning.

The corresponding RL problem is characterized by an MDP where the agent's actions correspond to generating text tokens, the state is the generated text so far, and the reward is provided by the reward model based on the human feedback. This is most commonly framed as a single-turn interaction with sparse reward, i.e., the episode consists of generating a single output (e.g., summary or dialogue response) with a single reward at the end evaluating the quality of the output.

The process typically starts with supervised fine-tuning on a small dataset labeled with the desired behavior (behavioral cloning). Empirical evidence shows that supervised fine-tuning alone is often insufficient, however, motivating preference-based fine-tuning (Stiennon et al., 2020). This may be due to the limited size and diversity of the supervised fine-tuning data, leading to overfitting and poor generalization. Preference data is often more diverse and can be collected more efficiently, helping the model learn a more robust and generalizable objective. This is further supported by Ramamurthy et al. (2023) who find that training a reward model from preference data which can then be used for policy optimization can be much more data efficient than supervised fine-tuning. Using a less capable model for the reward model may further improve generalization, as it is less likely to overfit the training data. Additionally, RLHF can train the model to give responses or refuse the response based on its own knowledge boundary, which is tricky to achieve with supervised fine-tuning since the label needs to depend on the model's state of knowledge (Schulman, 2023; Zhang et al., 2023b; Kadavath et al., 2022).

Regular RL tasks generally require two types of learning: Learning which states are rewarding and learning how to reach these states. This is in many ways a simplified setting compared to traditional RL problems, as state transitions are entirely deterministic, effectively requiring no exploration of environment dynamics. An equivalent formulation, possible due to the deterministic nature of the environment, is to consider the model's entire response as a single action, effectively treating the problem as a bandit problem (further discussed in Section 6.2). Token-level and response-level interpretations can also be mixed (Nakano et al., 2022), as they lend themselves to different parts of the training process. Ramamurthy et al. (2023) find, for example, that token-level discounting can be beneficial.

**Practical challenges** Note that in practice this assumption of single-turn interactions is frequently violated, with the LLM interacting with the human in a dialogue setting, occasionally calling out to external systems, or interacting with the environment in other ways. This is not in conflict with single-turn training, as the dialogue setting can be seen as a sequence of single-turn interactions. As a consequence, however, the LLM has limited planning capabilities and must rely on the human to provide feedback on each generated output. It generally cannot ask follow-up questions or engage in multi-turn interactions to clarify the human's feedback, except if explicitly trained or prompted to do so (Zhou et al., 2024). In practice, there are many language-model specific implementation details (Ramamurthy et al., 2023; Huang et al., 2024; Dong et al., 2024a; Xu et al., 2024) (see, e.g., Appendix E in the paper by Nakano et al. (2022) for a concrete example) that are not all covered in this work.

**Exploration** While we previously argued that exploration of *environment dynamics* is not necessary, exploration of the reward landscape is crucial. To reinforce behavior, it is necessary to explore the space of possible responses. This is particularly challenging due to the large action space and long time horizon, prohibiting random exploration. Exploration is commonly encouraged by a KL-divergence regularization term in the policy optimization objective, encouraging the agent to stay close to the pretrained model's behavior, which is generally stochastic due to the pretraining objective (Stiennon et al., 2020; Nakano et al., 2022). This avoids a phenomenon commonly referred to as entropy collapse, where the LLM overfits to the reward model and starts to give low-entropy next-token predictions (Nakano et al., 2022). Nonetheless, the pretrained model may not explore the space of possible responses sufficiently if the reward model encourages responses that are rarely or never generated by the pretrained model, such as abstaining from answering.

**Prompt distribution** RLHF for LLMs can be seen as a form of language-conditioned RL (Luketina et al., 2019), where the user's initial prompt specifies the task. Training a general-purpose model therefore requires a diverse set of prompts for training. These prompts are sometimes the same as used for preference learning (Rafailov et al., 2023), sometimes come from the same pool and may overlap partially (Ziegler et al., 2020; Stiennon et al., 2020) and sometimes are chosen to be distinct from that dataset (Nakano et al., 2022; Ouyang et al., 2022).

**Off-policy training** When studied in small-scale research contexts, RLHF is generally applied in an interleaved fashion, iterating or even parallelizing reward model training and reinforcement learning (Christiano et al., 2017). This results in a setting where the trajectories used to train the reward model are generated by a recent policy (i.e., semi-on-policy) and updated during the training process, although often not continuously (i.e., semi-online). For control applications, changes to the policy can quickly accumulate over longer time-horizons, leading to rapid shifts of the distribution of the observed trajectories and requiring on-policy updates to retain a useful reward model. In contrast, LLMs pose a slightly different setting, as the distribution shift is generally less pronounced due to the single-turn interactions and the diversity of the initial training data. Additionally, the size of the required dataset and the cost of human feedback often prohibit fully online training, necessitating at least some degree of offline or off-policy training (e.g., by using scraped preferences or prior data) (Stiennon et al., 2020; Askell et al., 2021). In practice, RLHF for LLMs is often implemented in a semi-online manner as well, introducing both on-policy and off-policy preferences infrequently (Stiennon et al., 2020). While most approaches in the literature include some online or on-policy training (Ziegler et al., 2020; Wu et al., 2021; Ouyang et al., 2022), the complexity of the task often leads to ad-hoc data collection methods (Stiennon et al., 2020; Nakano et al., 2022) compared to the more principled approach possible for small-scale research efforts (Christiano et al., 2017). Although it may be of less importance than for control applications, many works emphasize that at least some degree of online or on-policy data is important for final performance of LLM fine-tuning as well (Ziegler et al., 2020; Dong et al., 2024a; Xu et al., 2024). Note that direct methods such as DPO often lack online data, causing issues due to distributional shift (Xu et al., 2024).

**Extensions** Note, however, that RLHF for LLMs does not *always* operate in this simplified setting. Zhou et al. (2024) extend the RLHF setting to multi-turn interactions with LLMs, where the agent can ask follow-up questions to clarify the human's feedback. The LLM may additionally be given access to tools such as a web-browser (Nakano et al., 2022), leading to further sources of non-determinism and complexity.

Studied under the names of multi-turn RLHF or agentic LLMs fine-tuning (with the latter setting generally incorporating actions beyond text output), this setting re-introduces many of the complexities of traditional RL settings, such as nondeterministic state transitions, partial observability and delayed rewards (Ma et al., 2024a). Another extension is to introduce dense reward signals (Chan et al., 2024), which can help to guide the agent more effectively.

**Pretraining**  The language model, i.e., the policy, is generally pretrained with a self-supervised objective on a large corpus of text. This is followed by a step of supervised fine-tuning on a small dataset labeled with the desired behavior. This can be seen as a form of behavioral cloning, and a similar approach is sometimes used in RLHF for control. Similarly, since judging the quality of a response requires a certain level of natural language understanding that is difficult to acquire from the limited training data available for reward model training, the reward model is generally initialized from a language model pretrained on a large corpus of text. As a side-effect of this initialization, as the pretraining data likely contains examples of the desired behavior as well as natural language descriptions of human values, the reward model likely already has some knowledge of the desired behavior which then needs to be elicited by fine-tuning with the new objective (Yang, 2024). It is then additionally often pre-trained from offline preferences, e.g., scraped from the web or mined from implicit expressions of preferences (Askell et al., 2021). Remaining preferences are then often gathered in a somewhat ad-hoc manner, with the reward model trained on a combination of preferences on responses various intermediate models (Nakano et al., 2022).

**Inference-time selection**  The single-turn setting allows for a simple inference-time selection strategy, where the model generates several completions and the best one is selected based on the reward model (*best-of-n-sampling*) (Nakano et al., 2022; Menick et al., 2022). This is beneficial in some settings, as it reduces the risk of forgetting during fine-tuning and avoids the cost and time of policy optimization. It can be viewed as a single-step lookahead search, where the reward model acts as the value function.

**Direct methods**  While the separation into reward learning and policy training phases is common and useful in the context of RLHF for control, since it enables the RL agent to collect more samples for policy training than for reward model training, effectively learning about the environment dynamics without excessive human supervision, this separation is less useful in the context of LLM fine-tuning. Section 6.3 discusses direct methods for RLHF in more detail. Xu et al. (2024) finds that PPO, when tuned for the purpose of fine-tuning LLMs can often outperform direct methods, a phenomenon that can be partially explained by the distributional shift incurred when optimizing for human preferences, which reward-model based methods account for by using online data and a reward model that may partially generalize the human preferences (Lambert et al., 2024).

**Capabilities vs. alignment**  Parts of the vast training data will generally capture this type of interaction, so the model likely already has the capability to perform the task (Zhou et al., 2023; Gudibande et al., 2024), but it needs to be elicited by fine-tuning with the new objective (Yang, 2024). Note that this even extends to capabilities that could be considered part of the aligning process, such as helpfulness (Wang et al., 2023a), harmlessness (Ganguli et al., 2023), and honesty (Kadavath et al., 2022).

**Regularization**  In addition to the reward from the reward model, the policy is often regularized with a KL-divergence term (Jaques et al., 2017; Stiennon et al., 2020; Nakano et al., 2022) to the pretrained policy (Ramamurthy et al., 2023) or early-stopping (Nakano et al., 2022) to encourage it to stay close to the behavior of the pretrained model. This has been found to be crucial for preventing the policy from diverging from the behavior of the pretrained model, which can lead to catastrophic forgetting of the pretrained knowledge (Nakano et al., 2022; Ramamurthy et al., 2023).

# B   Prior Surveys

Based on the criteria discussed in Section 1.3, here we will first differentiate our survey from other surveys in marginally related subject areas sharing the common theme of human-in-the-loop RL. Then, we will describe

Table 6: An overview of prior surveys of human-in-the-loop RL. ✓ indicates that the criterion is a main focus of the survey, (✓) indicates that the criterion is partially addressed, while ✗ indicates that the criterion is not covered.

| Reference | Topic | Reward Modelling | Human Defined | Interactive and Online | Scalable and Async. |
|---|---|---|---|---|---|
| Wu et al. (2022) | Human-in-the-loop ML | ✗ | ✗ | ✗ | ✗ |
| Retzlaff et al. (2024) | Human-in-the-loop RL | (✓) | (✓) | ✓ | (✓) |
| Najar & Chetouani (2021) | RL with human advice | (✓) | (✓) | ✓ | (✓) |
| Lin et al. (2020a) | Social feedback | ✗ | (✓) | ✓ | ✗ |
| Poole & Lee (2024) | RL from brain signals | ✗ | ✓ | ✓ | ✗ |
| Cruz & Igarashi (2020) | Interactive RL for HCI | ✗ | ✗ | ✓ | ✗ |
| Osa et al. (2018) | Imitation learning | ✗ | ✓ | ✗ | ✗ |
| Arora & Doshi (2021) | Inverse RL | ✓ | ✓ | ✗ | ✓ |
| Bignold et al. (2021) | Assisted RL | ✗ | ✗ | (✓) | ✗ |
| Luketina et al. (2019) | Language-informed RL | ✗ | ✗ | ✗ | ✗ |
| Zhang et al. (2021) | Human guidance | ✗ | ✓ | ✓ | ✗ |
| Ji et al. (2023b) | AI Alignment | ✓ | ✗ | ✓ | ✓ |
| Liu et al. (2023c) | LLM applications | ✗ | ✗ | ✗ | ✗ |
| Ours | RLHF | ✓ | ✓ | ✓ | ✓ |

the differences between our survey and previous surveys or survey-like articles that exist within the RLHF field.

## B.1 Human-in-the-Loop RL

Human participation in ML, particularly in guiding machine learners, is a well-studied scenario. This field, commonly referred to as human-in-the-loop ML, can be further divided into subfields based on various criteria, e.g., the ones detailed in Section 1.3. Prior surveys of these subfields are compiled in Table 6 and briefly summarized in the following.

**Human-in-the-Loop** Learning from human feedback falls into the domain of human-in-the-loop ML. Wu et al. (2022) survey human-in-the-loop ML in general. They also cover some applications of RLHF (for LLMs in particular) but do not provide a detailed overview. Retzlaff et al. (2024) provide a similar overview over human-in-the-loop RL in particular, focusing on human involvement in RL on a more abstract level than our work and not covering RLHF in detail. Similarly broad in scope, Najar & Chetouani (2021) study the setting of RL with human advice, which they define as 'teaching signals that can be communicated by the teacher to the learning system without executing the task'. While this setting subsumes RLHF, the broad generality limits the depth to which their survey can cover RLHF approaches.

**Interactive RL** RLHF can be considered a sub-field of interactive RL, which studies RL algorithms that learn in interaction with humans. This interaction can take the form of feedback defining an objective, resulting in the RLHF setting, but can also, e.g., be used to drive exploration or speed up the agent's learning process.

Cruz & Igarashi (2020) survey interactive RL from an HCI viewpoint, exploring various ways humans can influence RL agents, with a particular focus on reward definition based on human feedback, without a predefined environmental reward function. Due to the breadth of their survey, they do not cover many works in the area. The survey by Lin et al. (2020a) centers on interactive RL using human social cues, like gestures and spoken language, but does not cover the reward modeling

aspect. Similarly, the study by Poole & Lee (2024) examines RL with direct feedback from human brain signals, such as through brain-computer interfaces, also not focusing on reward modeling.

**Demonstrations** Learning from demonstrations, in the form of behavior cloning (Osa et al., 2018) and inverse RL (Arora & Doshi, 2021), shares the goal of RLHF to learn behavior from human input. In contrast to RLHF, however, it requires demonstrations of the desired behavior instead of feedback, and these demonstrations are usually not provided interactively and online. This limits their applications and final performance due to the need for near-optimal demonstrations. Nonetheless, imitation and demonstration can be a useful component of an RLHF system but are not the main focus of this survey. However, we will discuss the intersection between these fields in some parts whenever necessary.

**Assisted RL** Bignold et al. (2021) review the field of assisted RL, where an agent may receive external information (for example, from a human) that aids it in action selection. While updates to the reward function are one of the possible effects of advice in this setting (in addition to action selection or modifications of the agent's internal state), it is usually assumed that an initial reward function is given and the extent of the updates is limited to reward shaping or supplementary reward signals. In contrast to RLHF, the external information does not define the task but only helps the agent achieve it. Closely related to this, Luketina et al. (2019) survey RL assisted by natural language. In addition to this assistance setting, they also discuss approaches that infer a language-conditioned reward function. However, they discuss this setting rather briefly and use techniques from inverse RL and not RLHF.

**Guidance** In their survey on human guidance, Zhang et al. (2021) explore various aspects related to RLHF. Although they touch on aspects such as reward learning, it is not the primary emphasis of their work. Instead, their main focus lies on exploring more immediate approaches that do not involve the learning of a reward model.

**AI Alignment** Ji et al. (2023b) provide a general overview of AI alignment, i.e., the challenge of aligning the objectives of an intelligent system with those of its human operators. This survey covers RLHF in some detail. As AI alignment is a very broad field, however, the article nevertheless does not go into as much depth on the topic of RLHF as we do here.

**Applications** Liu et al. (2023c) give an overview of current applications of RLHF methods for LLMs such as ChatGPT and GPT-4. Even though it currently enjoys a lot of attention, it is only one specific application area for RLHF. Our survey adopts a broader perspective, examining the diverse applications and impact of RLHF encompassing application areas beyond LLMs.

## B.2   PbRL and RLHF

Several previous surveys or survey-like articles are closely related to RLHF. Table 7 gives a brief overview of how these articles differ from ours, which we will explain in more detail below.

**Preference-Based RL** Previous surveys in the domain of RLHF often focus on PbRL, where feedback is limited to binary preferences (see Section 1.2). An illustrative example of this is the survey by Wirth et al. (2017), which is a direct precursor to our work. In contrast to our work, they concentrate on binary preferences for trajectories and primarily survey methods that learn policies without deriving a reward model. Since then, the reward-modeling approach has become dominant, and other approaches have extended RLHF to new feedback types. Abdelkareem et al. (2022) give another more recent literature review of PbRL. While this review focuses on reward modeling and includes some recent work, it is far less comprehensive than our review, as many aspects are only touched upon and partly overlap with those of Wirth et al. (2017).

**Feedback Types** Although not a survey per se, Jeon et al. (2020) propose reward-rational implicit choice as a unifying framework to comprehend many previous studies in PbRL and RLHF. To illustrate its generality, they overview different feedback types used in previous work and explain how they fit

Table 7: An overview of prior RLHF-specific surveys and articles with substantial review components. The symbols indicate coverage: ✓ indicates that the aspect is addressed, (✓) indicates that the aspect is partially addressed, while ✗ indicates that the aspect is not covered.

| Reference (Focus) | Beyond Comparisons | Label Collection | RM Training | Theory | App. and Benchmarks |
|---|---|---|---|---|---|
| Wirth et al. (2017) (preference-based RL) | ✗ | (✓) | (✓) | ✗ | (✓) |
| Abdelkareem et al. (2022) (recent advances of PbRL) | ✗ | ✗ | (✓) | (✓) | (✓) |
| Jeon et al. (2020) (feedback modelling) | ✓ | ✗ | (✓) | ✗ | ✗ |
| Casper et al. (2023) (open issues in RLHF) | ✓ | ✗ | ✓ | ✗ | (✓) |
| Fernandes et al. (2023) (language generation) | ✓ | ✓ | ✗ | ✗ | (✓) |
| Metz et al. (2023) (feedback types) | ✓ | ✓ | ✗ | ✗ | ✗ |
| Yuan et al. (2024) (feedback types) | ✓ | ✓ | ✗ | ✗ | ✓ |
| Ours (fundamentals, recent advances, and trends) | ✓ | ✓ | ✓ | ✓ | ✓ |

into their framework. The concurrent works by Metz et al. (2023) and Yuan et al. (2024), which are also not strictly surveys, propose frameworks for studying user interaction and interface design for multiple feedback types. As part of their work, they provide a classification of feedback types and a brief overview of RLHF approaches. Metz et al. (2023) have a stronger focus on the feedback interface and on learning from multiple feedback types simultaneously, discussing properties of feedback types and proposing a standard encoding for them. On the other hand, Yuan et al. (2024) also include an offline RLHF benchmark and have a stronger focus on the reward learning aspect, focusing on the entire learning pipeline. Nevertheless, many facets of RLHF are not addressed in those studies, as they are not primarily survey articles. Our survey has a broader scope and therefore provides more extensive coverage, going beyond the study of feedback types and discussing more recent work.

**Domain-Specific** Fernandes et al. (2023) focuses on human feedback for language generation. As a result of their focus, their survey is less comprehensive than this work but discusses some language-specific aspects that do not fall into our scope, such as using feedback models at generation time.

**Open Problems** Casper et al. (2023) provide a detailed overview of the open questions and limitations of RLHF with a particular focus on aspects of security, governance, and transparency. In their article, reward modeling is also covered, as is human feedback, which goes beyond preference comparisons, but other aspects, such as theoretical approaches or an overview of existing benchmarks, are not included. Thus, it can be seen as a supplementary article that is ideal for further reading once being familiarized with the topic through our survey.

