# OpenReview forum: "A Survey of Reinforcement Learning from Human Feedback"
_TMLR — Accepted by TMLR_

### Review · Reviewer_VZY8 · 2024-07-01

**Summary Of Contributions:**

The authors present a new survey on reinforcement learning from human feedback, focusing on a general frame of view across each of the steps in the RL process. The “novelty” in this area is to give the newest possible map of papers for new entrants in the field.

**Audience:**

Yes

**Claims And Evidence:**

Yes

**Requested Changes:**

1. I mentioned incomplete coverage of core papers in the flourishing of RLHF, here’s a list of papers I grabbed from a blog post from early 2023 that should be considered (coverage was pretty good in checking this):
    - GopherCite: [**Teaching language models to support answers with verified quotes**](https://www.deepmind.com/publications/gophercite-teaching-language-models-to-support-answers-with-verified-quotes) (Menick et al. 2022): Train a LM with RLHF to return answers with specific citations.
    - [**Dynamic Planning in Open-Ended Dialogue using Reinforcement Learning**](https://arxiv.org/abs/2208.02294) (Cohen at al. 2022): Using RL to enhance the conversational skill of an open-ended dialogue agent.
    - [**Llama 2**](https://arxiv.org/abs/2307.09288) (Touvron et al. 2023): Impactful open-access model with substantial RLHF details.
    - The first paper to do KL penalties for sequences https://arxiv.org/abs/1611.02796 (section 5.2.1 addition)
2. All of the content on previous surveys should probably be moved to the end or the appendix
3. It is probably worth adding a diagram that works for how the RLHF used for language models can work (offline labels, no state feedback)
4. There can be citations added to the paper that support arguments, but they probably are more niche. A section that I felt like could use populating was 5.1.6 (and nearby).
    1. Toward pluralistic alignment https://arxiv.org/abs/2402.05070
    2. Social choice for AI alignment https://arxiv.org/abs/2404.10271v1
    3. The History and Risks of Reinforcement Learning and Human Feedback, https://arxiv.org/abs/2310.13595
5. I found the active learning and query-learning framing to be a bit forced, as I have never seen it before in the literature, which makes the entry costs to the paper a bit higher. I don’t really know how to fix this.
6. In section 3.2.3 in discussing improvements, the concept of revisions and critiques should be included (e.g. constitutional AI and https://arxiv.org/abs/2402.07896)
7. I thought it was weird that section 4.2.2 had no mention of dataset size per domain, which is likely a crucial detail.
8. Section 5 seemed to have a bit higher variance than the core sections preceding it (e.g. 3 and 4). Some feedback:
    1. 5.1.X is likely to vary a lot by domain. This links with my language model comment, but it just didn’t seem as clear how to use it as some of the other sections.
    2. 5.2 seemed a little tangential, not sure how necessary it is in fullness
    3. 5.2 is missing references to a bunch of recent reward model ensembling and averaging papers: https://arxiv.org/abs/2312.09244 https://arxiv.org/abs/2401.12187
    4. 5.3 and the later evaluation section can include recent work on evaluating reward models (and generative reward models): https://arxiv.org/abs/2403.13787 https://arxiv.org/abs/2310.05470  https://arxiv.org/abs/2309.16155 (consistency eval is small)
    5. 5.5.1 the analysis of reward model pretraining is not accurate with the method described in the paper. It was additional pretraining, which is really more of fine-tuning of a language model.
    6. 5.5.2 should probably mention RLAIF and recent distillation of preferences work. And Meta Llama 2’s margin RM loss
9. Section 6 should mention best of n sampling and rejection sampling
10. Nits:
    1. 5.1.6 has a broken bibtex ref
    2. 4.2.1 particularly was a bit long and could use some of the paragraph breakdowns you did
    3. Section 6 has a clear typo in the intro to the section
    4. 3.2.2 has a author note that was supposed to be deleted

**Strengths And Weaknesses:**

This paper is clearly a useful document, especially once into the central portions of the paper (sections 3 to 5), but there is some thematic confusion and balance challenges that are to be expected as a subfield goes through such substantial shifts in interest. My primary concern with the paper is trying to reconcile how most of the readers in the paper will show up due to an interest in the application of RLHF to language models, but the inclusion of language-modeling related content seems largely an afterthought. Throughout, most of the figures and references either miss core recent works in that area of don’t indicate what is different about this blossoming area of research.

To be clear, as a reviewer I don’t think this is my decision to make, but given I’ll be giving a positive score either way, I think some discourse on the matter is important. There are plenty of sentences that aren’t really true for RLHF on LMs, but it’s not easy to fix:

- “Learning the agent’s objective from human feedback circumvents reward engineering challenges and fosters robust training” —> the only way to get a reasonable reward
- Anything “human-in-the-loop” doesn’t apply because it’s a contrived offline bandits approach
- reward model “should be learning in an interactive, online, …”

There are also other sources of confusion similar to this:

- Limiting the scope to papers within 5 years, which leads to selective inclusion or not of core papers outside of that window. The seminal papers of the field, which is pretty recent, should really be included.
- There could potentially be more forward pointers in the text. Like my language model comment, in section 5.2.7 I was really wondering when DPO would be mentioned (and if), which I also wondered during the intro. There were other examples where I was confused reading it, only to find it later, which forward links could avoid.

I focus on this because a lot of the content in the paper is great. There are plenty of sections that I starred as well done, such as feedback types and the origins of the Bradley Terry model. So, I think this is a good paper, but should revisit some of the framing and organization.

---

> ### Author Response · Authors · 2024-08-05
>
> Thank you for taking the time to review our paper and provide such extensive feedback! Your suggestions resulted in many improvements to the paper, and we truly believe that the paper is better for it, especially from the perspective of someone primarily interested in RLHF for language models.
>
> We have submitted an update to the paper that hopefully addresses your concerns and feedback, highlighting important changes in purple and additionally noting change identifiers in the margins. If you search for your reviewer ID (VZY8) in the document, you will find the changes we made in response to your feedback. We will not discuss the self-explanatory changes here, but rather use the space to discuss points that might require further clarification or discussion.
> We want to place a special emphasis on the new section `RLHF for Language Models' in the appendix (referenced in the introduction) that is intended to address the thematic confusion you pointed out.
>
> A. **Active Learning and Query-Learning Framing (5):** Thank you for your feedback regarding the active learning section. We are unsure how best to react to it. Do you think any changes are necessary and, if so, where do you think we could improve?
>
> B. **Reward Model Pretraining (8.6):** Could you clarify your concern regarding reward model pretraining in section 5.5.1? The way we currently phrase it, the reward model (after being initialized from a pretrained language model) is trained on preference data scraped from the web, as a pretraining phase to the "proper" reward model training. Is our understanding wrong, or could you clarify how we could improve the statement to avoid misleading the reader?
>
> C. **Domain-Specificity of Section 5.1 (8.1):** Could you clarify your concern regarding the domain-specificity of section 5.1? We believe that the right choice of feedback model is similarly important for many domains, as demonstrated by KTO in the LLM context.
>
> Thank you again for your detailed review and valuable suggestions. We look forward to further clarifying the points above and to your continued feedback.

---

> > ### Comment · Reviewer_VZY8 · 2024-08-06
> > **Comments**
> >
> > A. I think my comments may be well interpretted through the lens the other reviewer mentioned on making it more "tutorially." This type of thing, as long as simple enough to be used by the reader, seems fine.
> > B. This sentence overstates the importance:
> > > g Reward model pretraining (Askell et al., 2021; Bai et al., 2022a) leverages prior offline data to pretrain the preference model before training it on policy samples.
> > They were the only paper to do that, and they phased it out in future papers.
> > C. I think my point was mostly about how the different models of feedback will end up with a lot of caveats depending on the domain data is collected. At this point, I don't really remember more than feeling like the section wouldn't hold up as well when translate to how people are using RLHF today.

---

> > > ### Author Response · Authors · 2024-08-09
> > >
> > > Thank you for the clarifications!
> > >
> > > (A) We have strived to make our survey more tutorial-like in response to the reviews and hope that this addresses your feedback, please let us know if you have further improvements in mind.
> > >
> > > (B) Thanks you for that insight, we were not aware that this has been phased out in future papers! We are having difficulties to incorporate this information into the paper, however, since we cannot find any discussion of this in the literature. Given the closed-off nature of many recent works, absence of evidence is not evidence of absence and I'd like to avoid making an unsubstantiated claim. Do you have any publicly available sources that discuss this, or is it more your personal experience? Do you have any suggestions on how to best address this in the paper?
> > >
> > > (C) Regarding the domain-specificity of section 5.1, we believe that this section serves several purposes: it discusses how RLHF is applied (at least when a reward network is learned), how this model can be justified, and some of its main alternatives. We think that these insights are valuable across different domains. We hope this clarifies your concerns, but we are open to further suggestions on how to improve this section.

---

### Review · Reviewer_HME9 · 2024-07-10

**Summary Of Contributions:**

This is a very long paper (56 pages without the referneces) covering the RLHF-related literature, from basic components and definitions to variants in the form of feedback and label collection, reward model training, policy learning, regret theory, applications and benchmarks.

**Audience:**

Yes

**Broader Impact Concerns:**

No discussion of broader impact in the paper. Yet, there are discussions of the impact of the RL component of RLHF creating an incentive for deception in the resulting agents, which clearly has ethical considerations in terms of safety.

**Claims And Evidence:**

No

**Requested Changes:**

See the above weakness, please try to move towards a more tutorial-like presentation. Adding examples throughout would help (e.g., in sec. 3.1 for each of the attributes, in sec. 3.2.1, examples of actual instances and feedback).

p. 4: "accurately specifying reward is notoriously difficult ... mitigated by utilizing human feedback... helps avoid safety issues arising from misaligned reward"
Unfortunately, the last part if a misleading statement. We are far (as a community) from having found how to avoid the safety issues due to misaligned reward. It is basically a giant and open problem in AI safety (see some review in the International Scientific Report on the Safety of Advanced AI). Please change the text appropriately. In fact, RLHF has been shown to INCREASE some of the safety issues, e.g., the LLM learns deception in order to please the human labelers (mentioned in the last paragraph of the section).

p. 26: I don't see how pure on-policy data get provide guarantees. In fact it is almost certaint to miss the non-visited high-reward regions and correct OOD behavior so the sentence that ends with "to guarantee the relevance of the generated queries for the current policy" seems wrong.

p. 26: about query diversity, there is work on entropy regularized RL and related approaches like generative flow networks (including for automated red-teaming LLMs) to sample from high reward trajectories rather than maximize (and end up with a single high-reward solution which may be brittle).

p. 44: I did not understand what "intermediate" reward models referred to (top).

p. 45: "resulting in sparse feedback": I don't understand what you call that sparse, since every trajectory is assigned a (generally non-zero) reward, and we are in the formulation, as written at the beginning of the paragraph, where we consider a whole trajectory (episode) as an action. What is sparse of course is the coverage of episodes (among all the possible ones). For this very reason, I am not convinced that the statement "does not require exploration" in the last paragraph of that section is correct. There is going to be an exponentially large number of possible episodes, so exploration seems quite important to find more useful ones.

p. 45: Lots of DPO papers are cited but there is not one once of explanation about what it is. This is an example of poor writing for a survey.

p. 50: example of lack of tutorial-like writing: the term pessimistic MLE is not really explained. It is described in reference to pssimistic value function, which is also not explained or defined.

Typos:

p. 20, form --> from

p. 21, leftover logistical comment, top of sec 3.2.2

p. 36, been presented --> presented

p. 41, pretaining

p. 44, degenerated

p. 45, with being --> being

p. 46, subsequently embraced by many subsequent -->  embraced by many subsequent

**Strengths And Weaknesses:**

This topic is clearly timely and reading the paper allowed to size how large the literature already was. The list of cited papers is impressive.
The division in sections makes sense. The introductory sections are fairly clear.

However, at some point in the reading, maybe because the paper is too long, I found the paper's style becoming less engaging and too abstract.
A good survey should also serve to explain the concepts that are covered, as well as provide examples to help the reader create mental pictures of the relevant ideas. A survey is not necessarily like a tutorial, but still, it should not require the reader to have read most of the cited papers already. The paper tries to explain, but it quickly gets into unexplained jargon and terms that a ML researcher who is not deep into RLHF would find very difficult to follow.

---

> ### Author Response · Authors · 2024-08-05
>
> Thank you for taking the time to review our paper and for your valuable feedback. Your suggestions have significantly helped us improve our work. We agree that the paper can be more engaging and tutorial-like, and we hope we have made a step in that direction.
>
> We have submitted an update to the paper that hopefully addresses your concerns and feedback, highlighting important changes in purple and additionally noting change identifiers in the margins. If you search for your reviewer ID (HME9) in the document, you will find the changes we made in response to your feedback. We will not discuss the self-explanatory changes here, but rather use the space to discuss points that might require further clarification or discussion.
>
> 1. **Tutorial-Like Presentation:**
>    We have added examples throughout the paper, particularly in sections 3.1 and 3.2.1, to make it more engaging and easier to understand.
>
> 2. **Misleading Statement on Safety Issues (p. 4):**
>    We revised the statement to: "represents a step towards tackling the open problem posed by safety issues due to misaligned rewards," to better reflect the current state of research.
>
> 3. **Query Diversity (p. 26):**
>    We are unsure how best to integrate this comment. Could expand on those works on entropy regularized RL and generative flow networks?
>
> Thank you again for your insightful comments. We look forward to your further feedback.

---

> > ### Comment · Reviewer_HME9 · 2024-08-05
> > **Reply after author updates to the paper**
> >
> > Thank you for the extensive changes.
> >
> > Regarding query diversity, I suggest considering the following papers:
> >
> > * original work on generating a more diverse set of actions using entropy-regularized RL, e.g.,
> >      Ziebart, Brian D., Andrew L. Maas, J. Andrew Bagnell, and Anind K. Dey. "Maximum entropy inverse reinforcement learning." In Aaai, vol. 8, pp. 1433-1438. 2008.
> >      Haarnoja, Tuomas, Haoran Tang, Pieter Abbeel, and Sergey Levine. "Reinforcement learning with deep energy-based policies." In International conference on machine learning, pp. 1352-1361. PMLR, 2017.
> >
> > * more recent work which addresses the limitations of maxent RL when the same state can be reached in many ways:
> >      Bengio, E., Jain, M., Korablyov, M., Precup, D., & Bengio, Y. (2021). Flow network based generative models for non-iterative diverse candidate generation. Advances in Neural Information Processing Systems, 34, 27381-27394.
> >
> >     specifically with applications to diverse generation with LLMs:
> >
> >      Hu, Edward J., Moksh Jain, Eric Elmoznino, Younesse Kaddar, Guillaume Lajoie, Yoshua Bengio, and Nikolay Malkin. "Amortizing intractable inference in large language models." arXiv preprint arXiv:2310.04363 (2023).
> >
> >      Lee, Seanie, Minsu Kim, Lynn Cherif, David Dobre, Juho Lee, Sung Ju Hwang, Kenji Kawaguchi et al. "Learning diverse attacks on large language models for robust red-teaming and safety tuning." arXiv preprint arXiv:2405.18540 (2024).

---

> > > ### Author Response · Authors · 2024-08-09
> > >
> > > Thanks a lot for clarifying and providing the references! What you suggest looks like a very promising approach to generate diverse trajectories, and therefore more diverse queries. Since we are not aware of any existing work that explores this direction in the context of RLHF, we added a note in the paper to highlight this as a potential future research direction. We hope this addresses your concern. If you have any further suggestions or comments, please let us know.

---

### Review · Reviewer_9bJj · 2024-07-15

**Summary Of Contributions:**

The manuscript presents a comprehensive survey of the field of Reinforcement Learning from Human Feedback (RLHF), an emerging area that integrates human input directly into the reinforcement learning process rather than relying solely on predefined reward functions. The paper builds upon earlier work in preference-based reinforcement learning and explores the broader application of RLHF in training large language models (LLMs) and other intelligent systems.

**Audience:**

Yes

**Broader Impact Concerns:**

~

**Claims And Evidence:**

Yes

**Requested Changes:**

- Include one or two detailed case studies showcasing the application of RLHF in real-world scenarios, which would help illustrate the practical benefits and challenges discussed.
- Enhance the discussion on ethical considerations by including a section that explores potential risks, such as privacy concerns and biases in human feedback, and suggests mitigation strategies.
- Develop a comparative framework or table that contrasts RLHF with conventional RL and other human-in-the-loop learning methods, providing clearer insights into the unique advantages and limitations of each approach.

**Strengths And Weaknesses:**

Strengths:
- The paper is well-structured and articulately written, providing clear explanations that make complex concepts accessible to both new researchers and seasoned practitioners in the field.
- Given the increasing importance of human-in-the-loop systems in AI, the survey addresses a timely and significant topic that has implications for the development of ethical and effective AI systems.
- The survey covers a broad spectrum of literature, offering a balanced view on both theoretical advancements and practical implementations of RLHF.

Weaknesses:
- While the paper discusses theoretical aspects and potential applications, it lacks detailed case studies that demonstrate real-world implementations and outcomes of RLHF systems.
- The paper touches on the alignment of machine objectives with human values but does not delve deeply into the ethical challenges and societal impacts that might arise from RLHF.
- The survey could benefit from a more detailed comparative analysis between RLHF and other learning paradigms, highlighting specific scenarios where RLHF is particularly beneficial or detrimental.

---

> ### Author Response · Authors · 2024-08-05
>
> Thank you for your constructive feedback and suggestions. We appreciate your insights and have made several changes to the paper based on your comments.
>
> We have submitted an update to the paper that hopefully addresses your concerns and feedback, highlighting important changes in purple and additionally noting change identifiers in the margins. If you search for your reviewer ID (9bJj) in the document, you will find the changes we made in response to your feedback. We will not discuss the self-explanatory changes here, but rather use the space to discuss points that might require further clarification or discussion.
>
> 1. **Case Studies:**
>    Our applications section (8.1) includes examples that demonstrate the practical benefits and challenges of RLHF. We believe these address your request for detailed case studies. Could you clarify if you had something else in mind?
>
> 2. **Ethical Considerations:**
>    We have added a section discussing ethical considerations, including potential risks such as privacy concerns and biases in human feedback, along with mitigation strategies.
>
> 3. **Comparative Analysis:**
>    We address the comparative analysis with Table 1 and the discussion of related surveys in section B.0.1. These sections contrast RLHF with conventional RL and other human-in-the-loop learning methods. A direct comparison in a table is challenging due to the fundamentally different assumptions about the existence of a ground-truth reward function. Does this clarify your concern, or did you have a different comparison in mind?
>
> Thank you again for your valuable feedback. We look forward to your further comments.

---

### Comment · Editors_In_Chief · 2025-03-14

Hello authors, and thank you for your patience on this paper. It recently came to our attention that there was a technical issue on our side that prevented this paper from moving to the next stage, which is why there has been radio silence for several months. In essence, this paper had "fallen through the cracks," and so are working to get it completed ASAP (hopefully within the next couple weeks).

Thanks again for your patience,
Gautam

---

### Decision · Action_Editor_2tHY · 2025-05-08

**Recommendation:** Accept as is

**Comment:**

This is a comprehensive overview of the field of reinforcement learning from human feedback (RLHF), that is valuable to the machine learning community at large.

It provides an extensive survey, touching on the historical origins and fundamental principles of RLHF, applications spanning several domains, and broader research trends. It successfully synthesizes a wide range of literature, making it a valuable resource for both new researchers and experienced practitioners for this timely topic.

The survey is well-organized with a nice and gentle introduction that lays a strong foundation for the rest of the material, like a mini-textbook.

The authors have made several revisions based on reviewer comments, improving tutorial-like explanations, adding examples, and enhancing discussions on ethical considerations.

One minor deficiency relates to the coverage of LLM-Specific RLHF: as a reviewer has pointed out, the paper does not always draw the connections to what is overwhelmingly the most popular application domain for RLHFs today, namely, for training LLMs.

However, overall, this is still easily a worthy contribution to TMLR.

**Audience:**

Yes

**Claims And Evidence:**

Yes